# Harmful Overfitting in Sobolev Spaces

**Kedar Karhadkar** [* 1]   **Alexander Sietsema** [* 1]   **Deanna Needell** [1]   **Guido Montufar** [1 2]

## Abstract

Motivated by recent work on benign overfitting in overparameterized machine learning, we study the generalization behavior of functions in Sobolev spaces $W^{k,p}(\mathbb{R}^d)$ that perfectly fit a noisy training data set. Under assumptions of label noise and sufficient regularity in the data distribution, we show that approximately norm-minimizing interpolators, which are canonical solutions selected by smoothness bias, exhibit harmful overfitting: even as the training sample size $n \to \infty$, the generalization error remains bounded below by a positive constant with high probability. Our results hold for arbitrary values of $p \in [1, \infty)$, in contrast to prior results studying the Hilbert space case ($p = 2$) using kernel methods. Our proof uses a geometric argument which identifies harmful neighborhoods of the training data using Sobolev inequalities.

## 1. Introduction

A classical tenet of statistical learning theory is that exact interpolation of noisy data should lead to poor generalization. Surprisingly, a growing body of recent work has shown that this intuition can fail in modern overparameterized regimes: in certain settings, a statistical model can perfectly fit a noisy training dataset, potentially with several or even many incorrect labels, while still generalizing well to unseen test data, a phenomenon now referred to as benign overfitting.

The ability of a model to exhibit benign overfitting is particularly relevant because standard learning algorithms fit noisy training data via empirical risk minimization, and its occurrence indicates a surprising robustness of the learned model to label noise. Consequently, there is significant interest in understanding when benign overfitting can arise and

---
*Equal contribution [1]Department of Mathematics, University of California, Los Angeles, Los Angeles, CA, USA [2]Department of Statistics & Data Science, University of California, Los Angeles, Los Angeles, CA, USA. Correspondence to: Kedar Karhadkar <kedar@math.ucla.edu>.

*Proceedings of the $43^{rd}$ International Conference on Machine Learning*, Seoul, South Korea. PMLR 306, 2026. Copyright 2026 by the author(s).

when it necessarily fails. There has been some work on benign overfitting in neural networks but almost all of it is for extremely high-dimensional data, where the dimension must significantly exceed the number of data points. Here, we instead aim to understand the behavior of fixed-dimension data which is often more realistic.

**Our contributions**

Our main result (Theorem 3.7) shows approximately norm-minimizing interpolators in Sobolev spaces *cannot* benignly overfit. Under standard assumptions on label noise and mild regularity of the data distribution, we prove that any interpolating solution whose Sobolev norm is within a constant factor of the minimum necessarily suffers from persistent excess risk. In particular, even as the number of training samples tends to infinity, the population error of such solutions remains bounded away from the Bayes-optimal risk with high probability. This is a significant generalization of prior work, which works with either specific model classes within Sobolev spaces or with general Sobolev spaces but more restricted parameter bounds.

More precisely, we establish a uniform lower bound on the expected excess risk of all approximately norm-minimizing interpolators in $W^{k,p}(\mathbb{R}^d)$ for a broad range of smoothness parameters, including values beyond the Hilbert space case $p = 2$. This bound depends only on the Sobolev parameters, the data distribution, and the noise level, but is independent of the sample size, as long as the sample size is large enough. This demonstrates that smoothness bias alone does not guarantee benign overfitting in Sobolev spaces, and that norm minimization can in fact lead to *harmful overfitting*.

Our results improve upon existing work on interpolation in Sobolev spaces and kernel regression by establishing that harmful overfitting occurs in a more general setting than previously studied. In particular, we loosen the following requirements:

- We consider interpolators in spaces $W^{k,p}(\mathbb{R}^d)$ for $k \in \mathbb{N}$ and $p \in [1, \infty)$, which goes beyond the commonly studied case $p = 2$. For $p \neq 2$, $W^{k,p}(\mathbb{R}^d)$ is not a Hilbert space, and minimum-norm interpolation is no longer a linear problem. As a result, our proof techniques use the local oscillations of interpolators

rather the linear structure of the function class.

- We study *approximately* norm-minimizing (ANM) interpolators rather than just norm-minimizing interpolators. ANM functions are defined by having sufficient smoothness as determined by the Sobolev norm and do not take a particular functional form.

- We consider a general class of data distributions with label noise which in particular includes regression with broad classes of additive heteroskedastic noise.

- We assume that the loss function satisfies a mild growth condition, which includes all $\ell^q$ losses for $q \in [1, \infty)$.

**Organization:** This paper is organized as follows: after reviewing related work in Section 2, in Section 3, we outline our assumptions and state the main theorem. In Section 4, we provide an overview of the proof of the main theorem including the statements of several intermediate results. In Section 5, we provide a complete proof of the main theorem. Finally, in Section 6, we explore a different scaling regime from the main theorem and contrast it with our other results. All proofs from Sections 3 and 4 are provided in the appendices.

## 2. Related work

The phenomenon of *benign overfitting*, where models that interpolate noisy training data nonetheless achieve strong generalization, has attracted significant attention in recent years. Early theoretical work established that exact interpolation does not necessarily preclude consistency, challenging classical statistical intuitions (Bartlett et al., 2020; Zhang et al., 2021). Building on this line of research, Mallinar et al. (2022) proposed a taxonomy distinguishing *benign*, *tempered*, and *catastrophic* overfitting, providing a unified framework for comparing different interpolation regimes.

Benign overfitting has been most thoroughly understood in linear settings. In particular, Bartlett et al. (2020) and subsequently Muthukumar et al. (2020); Zou et al. (2021); Hastie et al. (2022); Koehler et al. (2021); Chatterji & Long (2022); Shamir (2022); Wang & Thrampoulidis (2022) analyzed minimum-norm interpolation in overparameterized linear regression, identifying precise conditions under which interpolators generalize despite fitting noisy labels. These results highlight the central role of the inductive bias induced by norm minimization.

More recently, a growing body of work has investigated benign overfitting in nonlinear models, including neural networks and transformer architectures. Several studies provide theoretical or empirical evidence that overparameterized neural networks trained by gradient-based methods

on high-dimensional data can exhibit benign overfitting under suitable conditions (Cao et al., 2022; Frei et al., 2023; George et al., 2023; Xu et al., 2024; Karhadkar et al., 2024; Magen et al., 2024; Kou et al., 2023). Together, these works suggest that benign overfitting is not restricted to linear or kernelized models, but may arise more broadly in modern deep learning architectures.

Related to this, a number of papers have explored *tempered overfitting*, an intermediate regime in which the test error does not vanish but remains controlled. In the setting of threshold networks with binary weights, Harel et al. (2024) establish provable guarantees for tempered overfitting. The role of ambient dimension is further investigated by Kornowski et al. (2023), who show that two-layer neural networks trained on constant data with label-flipping noise exhibit benign overfitting in very high dimensions ($d \gtrsim n^2 \log n$), but only tempered overfitting in one dimension. The impact of the loss function has also been highlighted: Joshi et al. (2023) demonstrate that, for two-layer minimum-norm univariate networks interpolating noisy data, the $L^1$ loss leads to tempered overfitting, whereas the $L^2$ loss results in catastrophic overfitting.

Beyond gradient-based learning in parametric models, other learning rules have also been analyzed through the lens of interpolation. Barzilai et al. (2025) show that Nadaraya-Watson interpolators can exhibit benign, tempered, or catastrophic overfitting depending on the choice of the locality parameter. Classical results on nearest-neighbor methods provide an early example of interpolation with controlled risk: Cover & Hart (1967) showed that the 1-nearest neighbor classifier has risk at most twice that of the Bayes-optimal predictor. Kur et al. (2024) prove an upper bound for the generalization error for minimum-norm interpolators in Banach spaces based on Rademacher complexity and the geometry of the space which is sharp for $\ell^p$ linear regression ($p \in [1, 2]$).

Several works have studied harmful overfitting in kernel regression. Of particular relevance to the present work, Rakhlin & Zhai (2019) show that minimum-norm kernel regression exhibits harmful overfitting when the associated reproducing kernel Hilbert space (RKHS) is a Sobolev space $W^{k,2}(\mathbb{R}^d)$ with $d$ odd and $k = (d+1)/2$. Buchholz (2022) generalize upon this work by showing that kernel regression is exhibits harmful overfitting when the RKHS is a Sobolev space $W^{k,2}(\mathbb{R}^d)$ with $d/2 < k < 3d/4$. Beaglehole et al. (2023) establish that minimum-norm kernel regression is inconsistent for a broader set of kernels, and investigate the effect of the bandwidth of the kernel on generalization error. Cheng et al. (2024) identify regimes in which kernel regression is benign, tempered, and catastrophic in terms of the spectrum of the kernel. Li et al. (2024) show that kernel regression generalizes poorly for a class of kernels

satisfying a particular eigenvalue decay condition.

In contrast, Haas et al. (2023) demonstrate that interpolation can be consistent for kernel regression when the kernel bandwidth is chosen appropriately. These results underscore that interpolation behavior in Sobolev-type function classes is delicate and is sensitive to both regularity and inductive bias.

Sobolev norms and other norms measuring smoothness over a domain are closely related to the implicit bias and representational cost of neural networks. Ongie et al. (2019) show that the representational cost of an infinite-width two-layer neural network is given by an integral norm of the Radon transform of a fractional Laplacian of the network in function space. This norm is upper bounded by the $W^{d+1,1}$ norm. Parhi & Nowak (2021) characterize two-layer neural networks as minimizers of an inverse problem regularized by a higher order integral norm. Ma & Ying (2021) upper bound the Sobolev seminorm of a neural network trained with SGD, showing that neural networks have an implicit bias towards minimizing higher order variation.

# 3. Setup and main result

In this work, we consider a purely probabilistic data model, which includes a variety of regression problems as special cases. We let $\Omega$ be a bounded open subset of $\mathbb{R}^d$ with $C^1$ boundary, and let $\mu$ be a Borel probability measure on $\overline{\Omega} \times \mathbb{R}$. We denote the marginal distributions of $\mu$ in the first and second coordinates by $\mu_{\mathbf{x}}$ and $\mu_y$, respectively. For $i \in [n]$, we sample training pairs $(\mathbf{x}_i, y_i)$ i.i.d. from $\mu$, where $x_i \in \mathbb{R}^d$ represents an input data point and $y_i \in \mathbb{R}$ represents the corresponding label. We also let $\ell : \mathbb{R} \times \mathbb{R} \to [0, \infty)$ be a continuous loss function with $\ell(\hat{y}, y) = 0$ if and only if $\hat{y} = y$.

When studying overfitting, a natural type of predictor to consider is one which perfectly fits a training dataset while being "as simple as possible" according to some notion of complexity. We will interpret "simple" to mean a function which is smooth in the sense that its derivatives take small values. We can formalize this through the notion of Sobolev spaces. For $k \in \mathbb{N}$ and $p \in [1, \infty)$, the Sobolev space $W^{k,p}(\mathbb{R}^d)$ consists of functions $g : \mathbb{R}^d \to \mathbb{R}$ whose derivatives up to order $k$ have finite $L^p$-norm. We define the norm in $W^{k,p}(\mathbb{R}^d)$ by

$$\|g\|_{W^{k,p}(\mathbb{R}^d)} = \sum_{|\alpha| \le k} \left( \int_{\mathbb{R}^d} |D^\alpha g(\mathbf{x})|^p d\mathbf{x} \right)^{1/p},$$

where $D^\alpha g$ denotes a higher order partial derivative of $g$ multi-indexed by $\alpha = (\alpha_1, \ldots, \alpha_d)$. We will use this norm as our measure of the complexity of a function.

We consider functions that are elements of a Sobolev space

$W^{k,p}(\mathbb{R}^d)$ where $kp > d$, so pointwise evaluation is well-defined and continuous by the Sobolev embedding theorem (see Theorem A.1). We say that $f^*$ is a *minimum-norm solution* for the dataset if it solves the optimization problem

$$\min_{f \in W^{k,p}(\mathbb{R}^d)} \|f\|_{W^{k,p}(\mathbb{R}^d)} \tag{1}$$
$$\text{subject to } \ell(f(\mathbf{x}_i), y_i) = 0 \text{ for all } i \in [n].$$

Minimum norm interpolation is a common framework for studying benign overfitting; see, e.g., Bartlett et al. (2020). We consider a more general class of solutions. We say that $f \in W^{k,p}(\mathbb{R}^d)$ is an *approximately norm minimizing solution with factor* $\gamma \ge 1$ ($\gamma$-ANM solution) if $\ell(f(\mathbf{x}_i), y_i) = 0$ for all $i \in [n]$ and $\|f\|_{W^{k,p}} \le \gamma \|f^*\|_{W^{k,p}}$, where $f^*$ is a minimum-norm solution.

**Notation:** We fix the following notation: if $X$ is a metric space, and $x \in X$, we write $B(x, r)$ to denote the open ball of radius $r$ about $x$. If $E \subset \mathbb{R}^d$ is measurable, we denote its Lebesgue measure by $|E|$. If $X$ is a metric space, we let $\mathcal{B}(X)$ denote the set of Borel subsets of $X$. We use $\text{Poly}_S(r)$ to denote an arbitrary polynomial in $r$ whose coefficients and degree may depend on the elements of some set $S$, and use $\gtrsim_S, \lesssim_S$ to denote inequality up to some constant dependent only on the elements of $S$.

## 3.1. Assumptions on loss function and data distribution

For our main results, we impose some additional assumptions on the loss function and data distribution.

**Assumption 3.1** (Growth rate of loss function). There exist constants $C_\ell, \tau_\ell > 0$ such that for all $\hat{y}, y \in \mathbb{R}$,

$$\ell(\hat{y}, y) \le C_\ell \exp(\tau_\ell(1 + |\hat{y}|)(1 + |y|)).$$

The above assumption is satisfied by most losses in regression settings, including $\ell^q$ losses for $q \in [1, \infty)$. We impose this mild growth condition to prevent outliers in the output dataset from having an extremely large effect on the population loss. One could relax this assumption by imposing stricter regularity on the range of values possible for the output distribution (such as boundedness).

Next, we assume that the conditional distribution of $y$ given $\mathbf{x}$ exists and is sufficiently regular (see B for a review of definitions).

**Assumption 3.2** (Regularity of conditional distribution). For $(\mathbf{x}, y) \sim \mu$, there exists a regular conditional probability $(\mathbf{x}_0, A) \mapsto \mathbb{P}(y \in A \mid \mathbf{x} = \mathbf{x}_0)$. There also exists a version of this regular conditional probability $\nu : \overline{\Omega} \times \mathcal{B}(\mathbb{R}) \to [0, 1]$ which is weakly continuous in the following sense. If $\mathbf{x}_m \to \mathbf{x}_0$ and $g : \mathbb{R} \to \mathbb{R}$ is continuous and bounded, then

$$\lim_{m \to \infty} \int_{\mathbb{R}} g d\nu(\mathbf{x}_m, \bullet) = \int_{\mathbb{R}} g d\nu(\mathbf{x}_0, \bullet).$$

We often write $\nu_{\mathbf{x}_0}$ to denote the probability measure $\nu(\mathbf{x}_0, \bullet)$.

While the above assumption includes continuous joint distributions, it also includes some distributions which have discrete marginal distributions. One such example is the case where $y$ is a Bernoulli random variable with $\mathbb{P}(y = 1 \mid \mathbf{x}) = p(\mathbf{x})$ for a continuous function $p : \overline{\Omega} \to [0, 1]$.

We define the *conditional loss* $\mathcal{L} : \mathbb{R} \times \overline{\Omega} \to [0, \infty]$ by

$$\mathcal{L}(\hat{y}; \mathbf{x}) = \int_{\mathbb{R}} \ell(\hat{y}, y) d\nu_{\mathbf{x}}(y).$$

The conditional loss measures the expected loss given that the input data point is $\mathbf{x}$ and we predict the value $\hat{y}$. Assumption 3.2 is the minimal one needed to ensure that the conditional loss exists and is well-behaved. It is in particular satisfied whenever the distribution of $(\mathbf{x}, y)$ has a continuous bounded density function.

Next, we will assume that the marginal distribution of $\mathbf{x}$ is sufficiently regular and the density does not attain arbitrarily large values.

**Assumption 3.3** (Regularity of marginal distribution). There exist constants $c_\mu, C_\mu > 0$ such that for all $\mathbf{x}, \mathbf{x}_0 \in \overline{\Omega}$, the marginal distribution $\mu_{\mathbf{x}}$ has a density $p_{\mathbf{x}} : \overline{\Omega} \to [0, \infty)$ satisfying
$$c_\mu \leq p_{\mathbf{x}}(\mathbf{x}_0) \leq C_\mu.$$

We impose this condition, following Buchholz (2022); Rakhlin & Zhai (2019), because our proof is fundamentally geometric. For each training point which has high conditional loss, we find a neighborhood of points which also has high conditional loss. This region has high Euclidean volume, and if the density function is well-behaved, it also has high probability.

We also assume that our output label distribution is sufficiently weak-tailed.

**Assumption 3.4** (Conditionally sub-Gaussian output). There exists a constant $C_y > 0$ such that for $(\mathbf{x}, y) \sim \mu$ and all $t \geq 0$,

$$\mathbb{P}(|y| \geq t \mid \mathbf{x}) \leq 2 \exp\left(-\frac{t^2}{C_y^2}\right)$$

almost surely.

This assumption complements Assumption 3.1 in ensuring that outliers do not affect the population loss pathologically.

The following assumption encodes that predicting using only a single (constant) output label will always be suboptimal.

**Assumption 3.5** (Label noise). Suppose that $(\mathbf{x}, y) \sim \mu$. Then there exist universal constants $\sigma > 0$, $\rho \in (0, 1]$ such

that

$$\mathbb{P}\left(\mathcal{L}(y; \mathbf{x}) \geq \sigma + \inf_{\hat{y} \in \mathbb{R}} \mathcal{L}(\hat{y}; \mathbf{x}) \,\Big|\, \mathbf{x}\right) \geq \rho$$

almost surely.

This condition is analogous in purpose to the assumption that $\mathrm{Var}(y \mid \mathbf{x}) \geq \sigma^2$ in that it is requiring there to be some probability of a point being "mislabeled".

We say that a Borel-measurable function $f : \overline{\Omega} \to \mathbb{R}$ is *Bayes-optimal* if for $(\mathbf{x}, y) \sim \mu$,

$$\mathbb{E}[\ell(f(\mathbf{x}), y) \mid \mathbf{x}] = \inf_{\hat{y} \in \mathbb{R}} \mathbb{E}[\ell(\hat{y}, y) \mid \mathbf{x}]$$

almost surely.

**Assumption 3.6** (Regularity of Bayes-optimal function). There exists a Bayes-optimal function $f_{\mathrm{Bayes}} \in W^{k,p}(\mathbb{R}^d)$.

We can interpret the above assumption as saying that although the data distribution is noisy, there is some de-noised ground truth which is sufficiently smooth.

Our assumptions are analogous to and generalize the assumptions of Buchholz (2022) and Rakhlin & Zhai (2019). Assumption 3.3 is the same assumption on the density of the input distribution used by Buchholz (2022) and Rakhlin & Zhai (2019). Assumptions 3.4 and 3.5 include the case of regression with the square loss and additive Gaussian or uniform noise. Assumption 3.6 states that there exists a sufficiently regular truth function generating the data, and generalizes the assumption of a smooth ground truth function with compact support.

### 3.2. Main theorem

With these assumptions in place, we now state our main result. This theorem establishes that, with high probability, the expected regret of any $\gamma$-ANM interpolant compared to the Bayes optimizer is bounded below by a constant independent of $n$. In other words, we show that any approximately norm-minimizing interpolant has at least constant generalization error even as $n \to \infty$, so we cannot benignly overfit.

**Theorem 3.7.** *Let $\epsilon \in (0, 1)$, let $k \in (d/p, 1.5d/p)$, and let $n \gtrsim \rho^{-2} \log(\epsilon^{-1}) + \mathrm{Poly}_{k,p,d}(\epsilon^{-1})$. Let $(\mathbf{x}, y) \sim \mu$ be a test point chosen independently from the training set $(\mathbf{X}, \mathbf{y})$. Then with probability at least $1 - \epsilon$ over the training set, the following holds. For all $\gamma$-ANM solutions $f_\gamma$:*

$$\mathbb{E}[\ell(f_\gamma(\mathbf{x}), y) - \ell(f_{\mathrm{Bayes}}(\mathbf{x}), y) \mid \mathbf{X}, \mathbf{y}] \geq C\gamma^{-pd/(kp-d)},$$

*where $C \in (0, \infty)$ is a constant depending on $k, d, p, \mu$, and $\ell$.*

To demonstrate this theorem in a concrete setting, we apply it to the case of Gaussian heteroskedastic noise using the squared loss $\ell(\hat{y}, y) = (\hat{y} - y)^2$. Given a ground truth function $g \in W^{k,p}(\Omega)$, suppose that

$$y = g(\mathbf{x}) + \epsilon,$$

where $\epsilon$ is drawn from a Gaussian distribution $\mathcal{N}(0, \sigma^2(\mathbf{x}))$ conditional on $\mathbf{x}$, with $0 < \sigma_{\min} \leq \sigma(\mathbf{x}) \leq \sigma_{\max}$ for some constants $\sigma_{\min}, \sigma_{\max} > 0$, and $\mathbf{x} \mapsto \sigma(\mathbf{x})$ is continuous. Additionally, suppose that the distribution $\mu_{\mathbf{x}}$ has a density function $p_{\mathbf{x}}$ satisfying Assumption 3.3. By checking that the rest of the assumptions of Theorem 3.7 hold under these conditions, we again conclude that the generalization error of any $\gamma$-ANM interpolator is bounded below; in this case, the generalization error simplifies to the $L^2(\mu)$ error between the learned solution and $g$.

**Corollary 3.8.** *Let $\epsilon \in (0, 1)$, let $k \in (d/p, 1.5d/p)$, and let $n \gtrsim \log(\epsilon^{-1}) + \mathrm{Poly}_{k,p,d}(\epsilon^{-1})$. With squared loss and a Gaussian noise distribution, with probability at least $1 - \epsilon$ over the training set, the following holds. For all $\gamma$-ANM solutions $f_\gamma$:*

$$\|f_\gamma - g\|_{L^2(\mu)}^2 \geq C \gamma^{-pd/(kp-d)},$$

*where $C \in (0, \infty)$ is a constant depending on $k, d, p$, and $\mu$.*

## 4. Proof overview and supporting results

Our proof follows three main steps:

1. Explicitly identify an interpolating solution and give an upper bound on its norm. This bounds the norm of any minimum-norm solution.

2. Show that there are a large number of points in the dataset which are sufficiently noisy and sufficiently separated from the other data points.

3. Show that around these points, any $\gamma$-ANM solution must be smooth enough not to violate the minimum-norm bound, and thus accumulates generalization error around these points.

In this section, we will expand on each of these steps.

### 4.1. Explicit interpolating solution

Our explicit interpolating function is based on *bump functions*, which are continuous, radially symmetric functions with compact support. Bump functions and radial basis functions are common tools for analyzing the the local behavior of Sobolev functions (Evans, 2022; Adams & Fournier, 2003). We first show the existence of bump functions supported on balls of arbitrary radius with bounded norm.

**Lemma 4.1.** *Let $\mathbf{x}_0 \in \mathbb{R}^d$ and let $\delta > 0$. There exists $\psi \in W^{k,p}(\mathbb{R}^d)$ satisfying the following properties:*

1. *For all $\mathbf{x} \in \mathbb{R}^d$, $\psi(\mathbf{x}) \in [0, 1]$.*

2. *For all $\mathbf{x} \in B(\mathbf{x}_0, \delta/2)$, $\psi(\mathbf{x}) = 1$.*

3. *For all $\mathbf{x} \notin B(\mathbf{x}_0, \delta)$, $\psi(\mathbf{x}) = 0$.*

4. *$\|\psi\|_{W^{k,p}(\mathbb{R}^d)} \lesssim_{k,d,p} 1 + \delta^{(d-kp)/p}$.*

We construct an interpolant by placing bump functions around each data point in the dataset with magnitudes corresponding to the associated label. To ensure that these functions have non-overlapping supports, we show that the corresponding balls are disjoint. For a dataset $\mathbf{X} \in \mathbb{R}^{n \times d}$, we define the values $\delta_1(\mathbf{X}), \cdots, \delta_n(\mathbf{X}) \in (0, \infty)$ as follows. For $i \in [n]$, let $\delta_i(\mathbf{X})$ be the radius of the largest ball about $\mathbf{x}_i$ not containing any other data points. That is,

$$\delta_i(\mathbf{X}) = \min_{\ell \neq i} \|\mathbf{x}_i - \mathbf{x}_\ell\|.$$

We write $\delta_i$ in place of $\delta_i(\mathbf{X})$, except in situations where it is necessary to specify the dataset. With this definition, we establish that the radius-$\delta_i/2$ balls are disjoint.

**Lemma 4.2.** *The sets $B(\mathbf{x}_i, \delta_i/2)$ for $i \in [n]$ are disjoint.*

We then define our interpolant for the dataset $(\mathbf{X}, \mathbf{y})$ as

$$f = \sum_{i=1}^{n} y_i \psi_i,$$

where the $\psi_i$ are defined as in Lemma 4.1 with $\mathbf{x}_0 = \mathbf{x}_i$ and $\delta = \delta_i/2$.

By combining the norm bound from Lemma 4.1 (4.) with a concentration argument for the $\delta_i$s, we can bound the norm of the above interpolant, and consequently the norm of minimum-norm functions in terms of $n$.

**Corollary 4.3.** *Let $\epsilon \in (0, 1)$ and $k \in (d/p, 1.5d/p)$. If $n \geq \mathrm{Poly}_{\beta,d}(\epsilon^{-1})$, then with probability at least $1 - \epsilon$, a minimum-norm solution satisfies*

$$\|f^*\|_{W^{k,p}(\mathbb{R}^d)}^p \lesssim_{k,d,p} n^{kp/d}.$$

### 4.2. Existence of noisy, separated subset

To show high generalization error, we first must identify a subset of data points which are both *noisy*, i.e., have sufficiently high conditional loss compared to the Bayes optimum, and are *separated*, i.e., whose nearest-neighbor radii are sufficiently large. We will also need to show that the associated labels for these data points are not too large, so that individual data points do not have a disproportionate influence on the norm of interpolators. This result is summarized in the following lemma.

**Lemma 4.4.** *Let $\epsilon \in (0,1)$ and let $n \gtrsim \rho^{-2} \log \frac{1}{\epsilon}$. Then with probability at least $1 - \epsilon$, there exists a subset $\mathcal{B} \subset [n]$ satisfying the following properties:*

1. *$|\mathcal{B}| \gtrsim \rho n$.*

2. *For all $i \in \mathcal{B}$, $\delta_i \gtrsim_d n^{-1/d}$.*

3. *For all $i \in \mathcal{B}$,*
$$|y_i| \lesssim \sqrt{1 + \log(\rho^{-1})}.$$

4. *For all $i \in \mathcal{B}$,*
$$\mathcal{L}(y_i; \mathbf{x}_i) \geq \sigma + \mathcal{L}(f_{\text{Bayes}}(\mathbf{x}_i); \mathbf{x}_i).$$

With these properties, we can show that the norm bound on the interpolator forces it to accumulate generalization error around these points.

### 4.3. Smoothness of the $\gamma$-ANM solution

The following lemma characterizes the local deviation of a function in terms of its Sobolev norm.

**Corollary 4.5.** *Let $\delta > 0$ and $d < kp$, and suppose that $u \in W^{k,p}(\mathbb{R}^d)$. Then for all $\mathbf{x}_0 \in \mathbb{R}^d$ and all $\mathbf{x}_1 \in B(\mathbf{x}_0, \delta)$,*
$$|u(\mathbf{x}_1) - u(\mathbf{x}_0)|^p \lesssim_{k,d,p} \delta^{kp-d} \|u\|_{W^{k,p}(B(\mathbf{x}_0, 2\delta))}^p.$$

This lemma allows us to show that any interpolant must remain bounded away from the Bayes optimal solution in some neighborhood around the noisy training points. To convert this into information about the loss function, we show that the conditional loss is continuous under our previous assumptions.

**Lemma 4.6.** *The conditional loss $\mathcal{L} : \mathbb{R} \times \overline{\Omega} \to [0, \infty)$ is continuous.*

To conclude, we note here that the size of the radii in Lemma 4.4 (2.) is critical, as it exactly offsets the scaling of the volumes of the nearest-neighbor balls $B(\mathbf{x}_i, \delta_i)$, which scale proportional to $\delta_i^d$. By aggregating over a number of data points proportional to $n$, we recover a bound on the generalization error independent of $n$.

## 5. Proof of the main theorem

*Proof of Theorem 3.7.* By Corollary 4.3, if $n \geq \text{Poly}_{k,p,d}(\epsilon^{-1})$, then with probability at least $1 - \frac{\epsilon}{2}$,
$$\|f^*\|_{W^{k,p}(\mathbb{R}^d)}^p \leq C_1 n^{kp/d}, \tag{2}$$

where $C12$ depends on $k, d$, and $p$. We denote the event that this occurs by $\omega_1$.

By Lemma 4.4, if $n \gtrsim \rho^{-2} \log \frac{1}{\epsilon}$, then with probability at least $1 - \frac{\epsilon}{2}$, there exists a subset $\mathcal{B} \subset [n]$ satisfying the following properties:

1. $|\mathcal{B}| \geq C_2 \rho n$.

2. For all $i \in \mathcal{B}$, $\delta_i \geq C_3 n^{-1/d}$.

3. For all $i \in \mathcal{B}$,
$$|y_i| \leq C_4 \sqrt{1 + \log(\rho^{-1})}.$$

4. For all $i \in \mathcal{B}$,
$$\mathcal{L}(y_i; \mathbf{x}_i) \geq \sigma + \mathcal{L}(f_{\text{Bayes}}(\mathbf{x}_i); \mathbf{x}_i).$$

Here $C_2, C_4 \in [1, \infty)$ are universal constants, and $C_3 \in (0, \infty)$ is a constant depending only on $d$. We denote the event that this occurs by $\omega_2$.

Suppose that both $\omega_1$ and $\omega_2$ occur. Applying Corollary 4.5, we see that for all $i \in \mathcal{B}$ and $\mathbf{x} \in B(\mathbf{x}_i, \delta_i/4)$,
$$|f_\gamma(\mathbf{x}) - f_\gamma(\mathbf{x}_i)|^p \leq C_5 \|\mathbf{x} - \mathbf{x}_i\|^{kp-d} \|f_\gamma\|_{W^{k,p}(B(\mathbf{x}_i, \delta_i/2))}^p, \tag{3}$$

where $C_5 > 0$ is a constant depending on $k, d, p$.

Consider the regret function $\mathcal{R} : \mathbb{R} \times \overline{\Omega} \to [0, \infty)$, defined by
$$\mathcal{R}(y; \mathbf{x}) = \mathcal{L}(y; \mathbf{x}) - \mathcal{L}(f_{\text{Bayes}}(\mathbf{x}); \mathbf{x}).$$

We show that $f_\gamma$ attains high regret around many of the training data points. By Theorem A.1, $f_{\text{Bayes}}$ is continuous, and by Lemma 4.6, $\mathcal{L}$ is continuous, so $\mathcal{R}$ is continuous. Let $K = 1 + C_4\sqrt{1 + \log(\rho^{-1})}$. Since $\mathcal{R}$ is uniformly continuous on $[-K, K] \times \overline{\Omega}$, there exists $\delta \in (0, 1)$ depending on $\mu, \ell$ such that for $(y, \mathbf{x}), (y', \mathbf{x}') \in \overline{\Omega}$ with $|y - y'|, \|\mathbf{x} - \mathbf{x}'\| \leq \delta$, we have
$$|\mathcal{R}(y; \mathbf{x}) - \mathcal{R}(y'; \mathbf{x}')| \leq \frac{\sigma}{2}. \tag{4}$$

Let us define
$$C_6 = \min\left( \frac{C_3}{2}, \left( \frac{C_2 \delta^p \rho}{2 C_1 C_5} \right)^{1/(kp-d)}, \delta \right),$$

so $C_6$ is a constant depending on $k, p, d, \mu$, and $\ell$. Let $\mathcal{B}' \subset \mathcal{B}$ consist of all of the indices $i \in \mathcal{B}$ such that for all $\mathbf{x} \in B(\mathbf{x}_i, C_6 \gamma^{-p/(kp-d)} n^{-1/d})$,
$$|f_\gamma(\mathbf{x}) - f_\gamma(\mathbf{x}_i)| \leq \delta.$$

Suppose that $i \in \mathcal{B} \setminus \mathcal{B}'$. By the definition of $C_6$ and $C_3$,
$$C_6 \gamma^{-p/(kp-d)} n^{-1/d} \leq C_6 n^{-1/d} \leq \frac{C_3 n^{-1/d}}{2} \leq \frac{\delta_i}{2},$$

so $B(\mathbf{x}_i, C_6\gamma^{-p/(kp-d)}n^{-1/d}) \subset B(\mathbf{x}_i, \delta_i/2)$, and therefore for some $\mathbf{x} \in B(\mathbf{x}_i, C_6\gamma^{-p/(kp-d)}n^{-1/d})$,

$$
\begin{aligned}
\delta^p &\leq |f_\gamma(\mathbf{x}) - f_\gamma(\mathbf{x}_i)|^p \\
&\leq C_5 \|\mathbf{x} - \mathbf{x}_i\|^{kp-d} \|f_\gamma\|_{W^{k,p}(B(\mathbf{x}_i,\delta_i/2))}^p \quad \text{(By (3))} \\
&\leq C_5 \left(C_6\gamma^{-p/(kp-d)}n^{-1/d}\right)^{kp-d} \|f_\gamma\|_{W^{k,p}(B(\mathbf{x}_i,\delta_i/2))}^p \\
&\leq C_5 \left(\left(\frac{C_2\delta^p\rho}{2C_1C_5\gamma^p}\right)^{1/(kp-d)} n^{-1/d}\right)^{kp-d} \|f_\gamma\|_{W^{k,p}(B(\mathbf{x}_i,\delta_i/2))}^p \\
&= \frac{1}{2}\frac{C_2}{C_1}\delta^p\gamma^{-p}\rho n^{1-kp/d} \|f_\gamma\|_{W^{k,p}(B(\mathbf{x}_i,\delta_i/2))}^p.
\end{aligned}
$$

Summing the above over $\mathcal{B} \setminus \mathcal{B}'$, we get

$$
\begin{aligned}
\delta^p|\mathcal{B} \setminus \mathcal{B}'| &\leq \frac{1}{2}\frac{C_2}{C_1}\delta^p\gamma^{-p}\rho n^{1-kp/d} \\
&\quad \sum_{i \in \mathcal{B}\setminus\mathcal{B}'} \|f_\gamma\|_{W^{k,p}(B(\mathbf{x}_i,\delta_i/2))}^p.
\end{aligned}
$$

By Lemma 4.2, the sets $B(\mathbf{x}_i, \delta_i/2)$ are disjoint, so

$$
\begin{aligned}
&\sum_{i \in \mathcal{B}\setminus\mathcal{B}'} \|f_\gamma\|_{W^{k,p}(B(\mathbf{x}_i,\delta_i/2))}^p \\
&= \sum_{i \in \mathcal{B}\setminus\mathcal{B}'} \sum_{j=0}^k \int_{B(\mathbf{x}_i,\delta_i/2)} \|D^j f_\gamma(\mathbf{x})\|^p d\mathbf{x} \\
&\leq \sum_{j=0}^k \int_{\cup_{i\in\mathcal{B}\setminus\mathcal{B}'} B(\mathbf{x}_i,\delta_i/2)} \|D^j f_\gamma(\mathbf{x})\|^p d\mathbf{x} \\
&\leq \sum_{j=0}^k \int_{\mathbb{R}^d} \|D^j f_\gamma(\mathbf{x})\|^p d\mathbf{x} \\
&\leq \|f_\gamma\|_{W^{k,p}(\mathbb{R}^d)}^p
\end{aligned}
$$

and we can write

$$
\begin{aligned}
\delta^p|\mathcal{B} \setminus \mathcal{B}'| &\leq \frac{1}{2}\frac{C_2}{C_1}\delta^p\gamma^{-p}\rho n^{1-kp/d}\|f_\gamma\|_{W^{k,p}(\mathbb{R}^d)}^p \\
&\leq \frac{1}{2}\frac{C_2}{C_1}\delta^p\rho n^{1-kp/d}\|f^*\|_{W^{k,p}(\mathbb{R}^d)}^p \\
&\leq \frac{1}{2}C_2\delta^p\rho n \quad \text{(By (2))}.
\end{aligned}
$$

Rearranging, we get

$$
|\mathcal{B} \setminus \mathcal{B}'| \leq \frac{1}{2}C_2\rho n,
$$

so by property 1 of $\mathcal{B}$,

$$
|\mathcal{B}'| = |\mathcal{B}| - |\mathcal{B} \setminus \mathcal{B}'| \geq C_2\rho n - \frac{1}{2}C_2\rho n = \frac{1}{2}C_2\rho n. \quad (5)
$$

Now suppose that $i \in \mathcal{B}'$ and $\mathbf{x} \in B(\mathbf{x}_i, C_6\gamma^{-p/(kp-d)}n^{-1/d})$. By Property 3 of $\mathcal{B}$,

$|f_\gamma(\mathbf{x}_i)| \leq K - 1$. By the construction of $\mathcal{B}'$, $|f_\gamma(\mathbf{x}) - f_\gamma(\mathbf{x}_i)| \leq \delta$, and in particular

$$
|f_\gamma(\mathbf{x})| \leq |f_\gamma(\mathbf{x}_i)| + \delta \leq K.
$$

So $(\mathbf{x}, f_\gamma(\mathbf{x})), (\mathbf{x}_i, f_\gamma(\mathbf{x}_i)) \in [-K, K] \times \overline{\Omega}$, and so

$$
\begin{aligned}
\mathcal{R}(f_\gamma(\mathbf{x}); \mathbf{x}) &\geq \mathcal{R}(f_\gamma(\mathbf{x}_i); \mathbf{x}_i) \\
&\quad - |\mathcal{R}(f_\gamma(\mathbf{x}); \mathbf{x}) - \mathcal{R}(f_\gamma(\mathbf{x}_i); \mathbf{x}_i)| \\
&\geq \mathcal{R}(f_\gamma(\mathbf{x}_i); \mathbf{x}_i) - \frac{\sigma}{2} \quad \text{(By (4))} \\
&\geq \sigma - \frac{\sigma}{2} \quad \text{(Property 4 of } \mathcal{B}) \\
&= \frac{\sigma}{2}.
\end{aligned}
\qquad (6)
$$

We have shown that for all $i \in \mathcal{B}'$, there is a large enough neighborhood around $\mathbf{x}_i$ in which $f_\gamma$ attains high regret. Aggregating over these regions, we get

$$
\begin{aligned}
&\int_{\mathbb{R}^d} \mathcal{R}(f_\gamma(\mathbf{x}); \mathbf{x}) d\mu_\mathbf{x}(\mathbf{x}) \\
&= \int_{\mathbb{R}^d} \mathcal{R}(f_\gamma(\mathbf{x}); \mathbf{x}) p_\mathbf{x}(\mathbf{x}) d\mathbf{x} \\
&\geq c_\mu \int_{\mathbb{R}^d} \mathcal{R}(f_\gamma(\mathbf{x}); \mathbf{x}) d\mathbf{x} \quad \text{(Assumption 3.3)} \\
&\geq c_\mu \sum_{i\in\mathcal{B}'} \int_{\Omega\cap B(\mathbf{x}_i, C_6\gamma^{-p/(kp-d)}n^{-1/d})} \mathcal{R}(f_\gamma(\mathbf{x}); \mathbf{x}) d\mathbf{x} \\
&\geq c_\mu \sum_{i\in\mathcal{B}'} \int_{\Omega\cap B(\mathbf{x}_i, C_6\gamma^{-p/(kp-d)}n^{-1/d})} \frac{\sigma}{2} d\mathbf{x} \quad \text{(By (6))} \\
&= \frac{1}{2}c_\mu\sigma \sum_{i\in\mathcal{B}'} |\Omega \cap B(\mathbf{x}_i, C_6\gamma^{-p/(kp-d)}n^{-1/d})| \\
&\gtrsim_{k,p,d,\mu,\ell} \sum_{i\in\mathcal{B}'} \gamma^{-pd/(kp-d)}n^{-1} \quad \text{(By Theorem D.14)} \\
&= \gamma^{-pd/(kp-d)}n^{-1}|\mathcal{B}'| \\
&\gtrsim_\mu \gamma^{-pd/(kp-d)} \quad \text{(By (5))}.
\end{aligned}
$$

So there exists a constant $C_7$ depending on $k, p, d, \mu, \ell$ such that

$$
\int_{\mathbb{R}^d} \mathcal{R}(f_\gamma(\mathbf{x}); \mathbf{x}) d\mu_\mathbf{x}(\mathbf{x}) \geq C_7\gamma^{-pd/(kp-d)}.
$$

To conclude, we rewrite

$$
\begin{aligned}
&\mathbb{E}[\ell(f_\gamma(\mathbf{x}), y)] - \mathbb{E}[\ell(f_{\text{Bayes}}(\mathbf{x}); y)] \\
&= \int_{\mathbb{R}^d} \left(\mathcal{L}(f_\gamma(\mathbf{x}); \mathbf{x}) - \mathcal{L}(f_{\text{Bayes}}(\mathbf{x}); \mathbf{x})\right) d\mu_\mathbf{x}(\mathbf{x}) \\
&= \int_{\mathbb{R}^d} \mathcal{R}(f_\gamma(\mathbf{x}); \mathbf{x}) d\mu_\mathbf{x}(\mathbf{x}) \\
&\geq C_7\gamma^{-pd/(kp-d)},
\end{aligned}
$$

where the first line follows from total expectation. $\qquad\square$

# 6. Higher order regime

Theorem 3.7 establishes that for $k \in (d/p, 1.5d/p)$, ANM interpolators harmfully overfit. For $k < d/p$, the interpolation problem is ill-posed, as Sobolev embedding fails and the function class consists of discontinuous functions. For the case $k > 2d/p$, we will show that the ANM class is rich enough to contain some interpolators which overfit benignly and other interpolators which overfit harmfully. In order to prove this result, we place the following additional assumption that the loss grows at most polynomially in the labels.

**Assumption 6.1.** There exist constants $\kappa_\ell, q > 0$ such that for all $\hat{y}, y \in \mathbb{R}$,

$$\ell(\hat{y}, y) \leq \kappa_\ell (1 + |\hat{y}|^q + |y|^q).$$

**Theorem 6.2** (Existence of benign interpolators). *Let $kp > 2d$, $\alpha \in (1/(kp - d), 1/d)$, and $\epsilon \in (0, 1)$. There exist constants $c_1, c_2, c_3 > 0$ depending on $k, p, d, \Omega, \mu, \ell, \alpha$ such that the following holds. If $n \geq c_1 \epsilon^{-1}$, and $\gamma \geq c_2 \epsilon^{-2(kp-d)/(dp)}$, then with probability at least $1 - \epsilon$, there exists a $\gamma$-ANM interpolator $f_{\text{good}} \in W^{k,p}(\mathbb{R}^d)$ such that for a test point $(\mathbf{x}, y)$ drawn i.i.d. from $\mu$,*

$$\mathbb{E}[\ell(f_{\text{good}}(\mathbf{x}), y) \mid \mathbf{X}, \mathbf{y}] - \mathbb{E}[\ell(f_{\text{bayes}}(\mathbf{x}), y) \mid \mathbf{X}, \mathbf{y}]$$
$$\leq c_3 n^{-1+\alpha d}.$$

In the above theorem, $\alpha$ can be taken to be arbitrarily close to $1/(kp - d)$, so the generalization error can attain the rate $O(n^{-1+d/(kp-d)+c})$ for arbitrarily small $c > 0$.

**Theorem 6.3** (Existence of harmful interpolators). *Let $kp > 2d$ and let $\epsilon \in (0, 1)$. There exist constants $c_1, c_2, c_3 > 0$ depending on $k, p, d, \Omega, \mu, \ell$ such that the following holds. If $n \geq c_1 \epsilon^{-1}$, and $\gamma \geq c_2 \epsilon^{-2(kp-d)/(dp)}$, then with probability at least $1 - \epsilon$, there exists a $\gamma$-ANM interpolator $f_{\text{bad}} \in W^{k,p}(\mathbb{R}^d)$ such that for a test point $(\mathbf{x}, y)$ drawn i.i.d. from $\mu$,*

$$\mathbb{E}[\ell(f_{\text{bad}}(\mathbf{x}), y) \mid \mathbf{X}, \mathbf{y}] - \mathbb{E}[\ell(f_{\text{bayes}}(\mathbf{x}), y) \mid \mathbf{X}, \mathbf{y}] \geq c_3.$$

The above two theorems follow a similar proof structure. Both of them use the observation that for large values of $p$, the representational cost of a function $f \in W^{k,p}(\mathbb{R}^d)$ is dominated by its values and derivatives on a small outlier subset. In the limiting case $p = \infty$, the representational cost of a function is entirely determined by the maximum value of its derivatives. While most points in the training dataset have nearest-neighbor distance $\delta_i \sim n^{-1/d}$, there are a small number of outlier points with $\delta_i \sim n^{-2/d}$. These outlier points are able to absorb a first-order amount of representational cost, giving us additional flexibility to construct bump functions around the other training points which can be either benign or harmful. We provide complete proofs of these theorems in Appendix E.

# 7. Conclusion

In this paper, we study approximately norm-minimizing interpolators of noisy datasets in Sobolev spaces. We show that under certain mild assumptions, any such functions must have positive constant generalization error, even as the number of training samples approaches infinity, that is, any such functions must harmfully overfit to the training data in fixed dimension (in contrast to the more typically studied high-dimensional $d \gg n$ regime, in which one sees benign overfitting in many settings). We also demonstrate this result in a common setting with squared loss and Gaussian noise. Our results imply that norm minimization and smoothness are not sufficient for interpolation to generalize well, and suggest that improved generalization could require undertraining models when the dataset has high sample size.

There are a number of possible future directions which could expand upon our results. While we study norm minimization in Sobolev spaces, our arguments depend on local control of the oscillation of functions and could potentially be extended to other function spaces which satisfy similar inequalities. We consider regression-type loss functions $\ell$ taking value $0$ if and only if we predict the output label, and future work could investigate classification-type loss functions. We study spaces with integer derivatives $k$ which could potentially be extended to fractional Sobolev spaces. We characterize overfitting when $k \in (d/p, 1.5d/p)$ and when $k \in (2d/p, \infty)$, and future work could address the intermediate scaling $k \in [1.5d/p, 2d/p]$. Finally, the intermediate regime between overfitting in fixed dimension and overfitting in very high dimensions remains an open area of investigation.

## Impact Statement

This paper presents work whose goal is to advance the field of machine learning. There are many potential societal consequences of our work, none of which we feel must be specifically highlighted here.

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

## A. Background on Sobolev spaces

In this section, we briefly review some of the tools from the theory of Sobolev spaces that we employ. For a more complete introduction, see Evans (2022) and Adams & Fournier (2003).

First, recall that for an open set $U \subset \mathbb{R}^d$, $W^{k,p}(U)$ consists of functions $u \in L^p(U)$ which admit weak derivatives $D^\alpha u \in L^p(U)$ for all multi-indices $\alpha$ with $|\alpha| \leq k$. Here the weak derivative $D^\alpha u$ is defined to be a function in $L^p(U)$ such that for all smooth functions $v : U \to \mathbb{R}$ with compact support,

$$\int_U (D^\alpha u) v d\mathbf{x} = (-1)^{|\alpha|} \int_U u (D^\alpha v) d\mathbf{x}.$$

That is, the weak derivative is defined to be the function such that integration by parts holds.

A main tool in the analysis of Sobolev spaces is the Sobolev embedding theorem, which allows us to understand Sobolev spaces as spaces of smooth functions for the right scaling of $k$, $d$, and $p$.

**Theorem A.1** (Sobolev embedding). *Let $U$ be a bounded open subset of $\mathbb{R}^d$ with $C^1$ boundary. Suppose that $k > \frac{d}{p}$, and let*

$$\gamma = \begin{cases} \left\lfloor \frac{d}{p} \right\rfloor + 1 - \frac{d}{p} & \text{if } \frac{d}{p} \text{ is not an integer,} \\ \text{any positive number} < 1, & \text{if } \frac{d}{p} \text{ is an integer.} \end{cases}$$

*Then there exists $u^* \in C^{k-\lfloor d/p \rfloor - 1, \gamma}(\overline{U})$ such that $u^* = u$ almost everywhere, and*

$$\|u^*\|_{C^{k-\lfloor d/p \rfloor - 1, \gamma}(\overline{U})} \lesssim_{k,p,\gamma,U} \|u\|_{W^{k,p}(U)}.$$

This also lets us define pointwise evaluation for functions $u \in W^{k,p}(U)$ for open sets $U \subset \mathbb{R}^d$ when $k > \frac{d}{p}$. By the Sobolev embedding theorem, there exists a unique continuous function $u^* \in C^0(\mathbb{R}^d)$ which is equal to $u$ almost everywhere. For a point $\mathbf{x}_0 \in \mathbb{R}^d$, we define $u(\mathbf{x}_0) = u^*(\mathbf{x}_0)$. This evaluation defines a linear functional on $W^{k,p}(U)$. If $U$ is bounded with $C^1$ boundary, this functional is continuous, since

$$|u(\mathbf{x}_0)| = |u^*(\mathbf{x}_0)| \leq \|u^*\|_{C^0(\overline{U})} \lesssim_{k,p,U} \|u\|_{W^{k,p}(U)}.$$

In particular, this allows us to define pointwise evaluation for functions $u \in W^{k,p}(\mathbb{R}^d)$, since any such function is also an element of $W^{k,p}(U)$ for any open ball $U$ containing the point.

## B. Background on conditional probabilities

Here we review the notion of regular conditional probabilities which we use in Assumption 3.2. For a more complete introduction, see Durrett (2019).

Recall that for an integrable random variable $X$ taking values in a measurable space $(\Omega, \mathcal{F})$ and a sub-$\sigma$-algebra $\mathcal{G} \subset \mathcal{F}$, the *conditional expectation* $\mathbb{E}[X \mid \mathcal{G}]$ is defined to be the unique $\mathcal{G}$-measurable random variable (up to a set of probability zero) such that for all $\mathcal{G}$-measurable sets $A$,

$$\int_A \mathbb{E}[X \mid \mathcal{G}] d\mathbb{P} = \int_A X d\mathbb{P}.$$

We say that $g$ is a *version* of the conditional expectation $\mathbb{E}[X \mid \mathcal{G}]$ if $g = \mathbb{E}[X \mid \mathcal{G}]$ almost surely.

If $X$ is a random variable taking values in $(\mathcal{X}, \mathcal{F})$ and $Y$ is a random variable taking values in $(\mathcal{Y}, \mathcal{G})$, a *regular conditional probability* for $Y$ given $X$ is a function $\nu : \mathcal{X} \times \mathcal{G} \to [0, 1]$ such that

1. For all $x \in \mathcal{X}$, $\nu(x, \bullet)$ is a probability measure on $(\mathcal{Y}, \mathcal{G})$.

2. For all $A \in \mathcal{G}$, $\nu(\bullet, A)$ is a measurable function on $(\mathcal{X}, \mathcal{F})$.

3. For all $A \in \mathcal{G}$, $\nu(X(\bullet), A)$ is a version of $\mathbb{P}(Y \in A \mid X)$.

This formalizes the notion of the distribution of $Y$ given $X$.

# C. Proof of Corollary 3.8

*Proof of Corollary 3.8.* We check each of the assumptions of Theorem 3.7.

*Assumption 3.1 (Growth rate of loss function):* With $C_\ell, \tau_\ell = 1$, we have

$$
\begin{aligned}
\ell(\hat{y}, y) &= (\hat{y} - y)^2 \\
&\leq (|\hat{y}| + |y|)^2 \\
&\leq \exp(|\hat{y}| + |y|) \\
&\leq \exp((1 + |\hat{y}|)(1 + |y|)).
\end{aligned}
$$

*Assumption 3.2 (Regularity of conditional distribution):* By construction, the distribution has a regular conditional probability given by $(\mathbf{x}, A) \mapsto \mu_{\mathbf{x}}(A)$, where $\mu_{\mathbf{x}_0}$ is a Gaussian measure with mean 0 and variance $\sigma(\mathbf{x})^2$. If $h : \mathbb{R} \to \mathbb{R}$ is continuous and bounded, then

$$
\lim_{m \to \infty} \int_{\mathbb{R}} g \, d\nu_{\mathbf{x}_m} = \lim_{m \to \infty} \int_{\mathbb{R}} \frac{1}{\sqrt{2\pi\sigma(\mathbf{x}_m)^2}} g(\mathbf{x}) \exp\left(-\frac{\mathbf{x}^2}{\sigma(\mathbf{x}_m)^2}\right) d\mathbf{x}.
$$

The integrand on the right-hand side is less than

$$
\frac{1}{\sqrt{2\pi\sigma_{\min}^2}} g(\mathbf{x}) \exp\left(-\frac{\mathbf{x}^2}{\sigma_{\max}^2}\right),
$$

which is integrable, so by the dominated convergence theorem, the integral is equal to

$$
\int_{\mathbb{R}} \frac{1}{\sqrt{2\pi\sigma(\mathbf{x}_0)^2}} g(\mathbf{x}) \exp\left(-\frac{\mathbf{x}^2}{\sigma^2(\mathbf{x}_0)}\right) d\mathbf{x} = \int_{\mathbb{R}} g \, d\nu_{\mathbf{x}_0},
$$

and the assumption holds.

As the conditional probability has a continuous and bounded density, this follows from the dominated convergence theorem.

*Assumption 3.3 (Regularity of marginal distribution):* We assume this condition to hold.

*Assumption 3.4 (Output distribution is conditionally sub-Gaussian):* The output distribution is conditionally Gaussian by assumption, with sub-Gaussian parameter given by $\sigma_{\max}$.

*Assumption 3.5 (Label noise):* The conditional loss is

$$
\begin{aligned}
\mathcal{L}(\hat{y}; \mathbf{x}) &= \int_{\mathbb{R}} \ell(\hat{y}, y) d\nu_{\mathbf{x}}(y) \\
&= \int_{\mathbb{R}} (\hat{y} - y)^2 d\nu_{\mathbf{x}}(y) \\
&= \mathbb{E}[(\hat{y} - Z)^2],
\end{aligned}
$$

where $Z$ is a Gaussian random variable with mean $g(\mathbf{x})$ and variance $\sigma(\mathbf{x})^2$. Then

$$
\begin{aligned}
\mathbb{E}[(\hat{y} - Z)^2] &= \text{Var}(\hat{y} - z) + \mathbb{E}[\hat{y} - Z]^2 & (7) \\
&= \sigma(\mathbf{x})^2 + (\hat{y} - g(\mathbf{x}))^2. & (8)
\end{aligned}
$$

So

$$
\mathcal{L}(\hat{y}; \mathbf{x}) = \sigma(\mathbf{x})^2 + (\hat{y} - g(\mathbf{x}))^2.
$$

Taking the infimum of both sides in $\hat{y}$, we get

$$
\inf_{\hat{y} \in \mathbb{R}} \mathcal{L}(\hat{y}; \mathbf{x}) = \sigma(\mathbf{x})^2
$$

with equality attained if and only if $\hat{y} = g(\mathbf{x})$. In particular, $g$ is Bayes-optimal. Then for $(\mathbf{x}, y) \sim \mu$,

$$
\begin{aligned}
\mathbb{P}\left(\mathcal{L}(y; \mathbf{x}) \geq \sigma_{\min}^2 + \inf_{\hat{y} \in \mathbb{R}} \mathcal{L}(\hat{y}; \mathbf{x}) \Big| \mathbf{x}\right) &= \mathbb{P}\left((y - g(\mathbf{x}))^2 \geq \sigma_{\min}^2 \big| \mathbf{x}\right) \\
&= \mathbb{P}(\epsilon^2 \geq \sigma_{\min}^2 \mid \mathbf{x}) \\
&\geq \mathbb{P}(\epsilon^2 \geq \sigma(\mathbf{x})^2 \mid \mathbf{x}) \\
&\geq 0.1,
\end{aligned}
$$

where $\epsilon$ is a Gaussian random variable with mean $0$ and standard deviation $\sigma(\mathbf{x})$. So the assumption is satisfied with constants $\sigma_{\min}^2$ and $0.1$.

*Assumption 3.6*: We have shown that $g$ is Bayes-optimal, and $g \in W^{k,p}(\mathbb{R}^d)$ by definition.

Therefore, the assumptions of Theorem 3.7 are satisfied. Thus, we are guaranteed that there exists a constant $C$ depending on $k, d, p$, and $\mu$ such that $\mathbb{E}[\ell(f_\gamma(\mathbf{x}), y) - \ell(g(\mathbf{x}), y)] \geq C$. We will rewrite the left-hand side of this inequality. By the law of total expectation,

$$
\begin{aligned}
\mathbb{E}[\ell(f_\gamma(\mathbf{x}), y) - \ell(g(\mathbf{x}), y)] &= \mathbb{E}[\mathbb{E}[\mathcal{L}(f_\gamma(\mathbf{x}); \mathbf{x}) - \mathcal{L}(g(\mathbf{x}); \mathbf{x}) \mid \mathbf{x}]] \\
&= \mathbb{E}[\mathbb{E}[(f_\gamma(\mathbf{x}) - g(\mathbf{x}))^2 \mid \mathbf{x}]] \qquad\qquad \text{(By 7)} \\
&= \mathbb{E}[(f_\gamma(\mathbf{x}) - g(\mathbf{x}))^2] \\
&= \|f_\gamma - g\|_{L^2(\mu)}^2.
\end{aligned}
$$

This is bounded below by a constant independent of $n$, so the result follows.

$\square$

# D. Proofs from Section 4

In this section, we present the proofs of the results from Section 4.

## D.1. Properties of the conditional loss

As a first step, we show some basic properties of the conditional loss.

**Lemma D.1.** *For all $\tau \geq 0$,*

$$
\sup_{\mathbf{x} \in \overline{\Omega}} \int_{\mathbb{R}} \exp(\tau |y|) d\nu_{\mathbf{x}}(y) < \infty.
$$

*Proof of Lemma D.1.* Suppose that $(\mathbf{x}, y) \sim \mu$. By Assumption 3.4, $y$ is conditionally sub-Gaussian given $\mathbf{x}$, so there exist $C_1, C_2 \in (0, \infty)$ (depending on $\tau$) such that

$$
\begin{aligned}
2 &\geq \mathbb{E}\left[\exp\left(\frac{y^2}{C_1^2}\right) \Big| \mathbf{x}\right] \\
&\geq C_2 \mathbb{E}\left[\exp(\tau |y|) \mid \mathbf{x}\right]
\end{aligned}
$$

almost surely. Then by the definition of $\nu_{\mathbf{x}}$, there exists a subset $\Omega' \subset \overline{\Omega}$ of full measure such that

$$
\int_{\mathbb{R}} \exp(\tau |y|) d\nu_{\mathbf{x}}(y) \leq \frac{2}{C_2}
$$

for all $\mathbf{x} \in \Omega'$. We show that the above inequality in fact holds for all $\mathbf{x} \in \overline{\Omega}$. Let $\mathbf{x} \in \overline{\Omega}$. Since $\Omega'$ has full measure in $\Omega$ and $\Omega$ is an open subset of $\mathbb{R}^d$, $\Omega'$ is dense in $\overline{\Omega}$. Let $\{\mathbf{x}_m\}_{m \in \mathbb{N}}$ be a sequence in $\Omega'$ converging to $\mathbf{x}$. Let $K > 0$. Then for all $m \in \mathbb{N}$,

$$
\frac{2}{C_2} \geq \int_{\mathbb{R}} \min\left(K, \exp(\tau |y|)\right) d\nu_{\mathbf{x}_m}(y),
$$

and by Assumption 3.2,

$$\lim_{m \to \infty} \int_{\mathbb{R}} \min\left(K, \exp(\tau|y|)\right) d\nu_{\mathbf{x}_m}(y) = \int_{\mathbb{R}} \min\left(K, \exp(\tau|y|)\right) d\nu_{\mathbf{x}}(y).$$

So for all $K > 0$,

$$\int_{\mathbb{R}} \min(K, \exp(\tau|y|)) d\nu_{\mathbf{x}}(y) \leq \frac{2}{C_2}.$$

By the monotone convergence theorem,

$$\int_{\mathbb{R}} \exp(\tau|y|) d\nu_{\mathbf{x}}(y) = \int_{\mathbb{R}} \lim_{K \to \infty} \min(K, \exp(\tau|y|)) d\nu_{\mathbf{x}}(y)$$

$$= \lim_{K \to \infty} \int_{\mathbb{R}} \min(K, \exp(\tau|y|)) d\nu_{\mathbf{x}}(y)$$

$$\leq \frac{2}{C_2}.$$

This holds for all $\mathbf{x} \in \overline{\Omega}$, so the result follows. $\qquad\square$

As defined, the conditional loss could in principle be infinite. We show that in fact it only attains finite values.

**Lemma D.2.** *For all $y_0 \in \mathbb{R}$ and $\mathbf{x}_0 \in \overline{\Omega}$, $\mathcal{L}(y_0; \mathbf{x}_0) < \infty$.*

*Proof of Lemma D.2.* Let $y_0 \in \mathbb{R}$ and $\mathbf{x}_0 \in \overline{\Omega}$. Then

$$\mathcal{L}(y_0; \mathbf{x}_0) = \int_{\mathbb{R}} \ell(y_0, y) d\nu_{\mathbf{x}_0}(y)$$

$$\leq \int_{\mathbb{R}} C_\ell \exp\left(\tau_\ell(1 + |y_0|)(1 + |y|)\right) d\nu_{\mathbf{x}_0}(y) \qquad \text{(Assumption 3.1)}$$

$$< \infty. \qquad \text{(Lemma D.1)}$$

$\qquad\square$

Since $\mathcal{L}(y; \mathbf{x}) < \infty$ for all $y \in \mathbb{R}$ and $\mathbf{x} \in \overline{\Omega}$, we will henceforth view $\mathcal{L}$ as a function $\mathbb{R} \times \overline{\Omega} \to [0, \infty)$. Importantly, we can show that this function is continuous.

Finally, we conclude that the Bayes optimizer minimizes the conditional loss at each point.

**Lemma D.3.** *For all $\mathbf{x}_0 \in \overline{\Omega}$,*
$$\mathcal{L}(f_{\text{Bayes}}(\mathbf{x}_0); \mathbf{x}_0) = \inf_{\hat{y} \in \mathbb{R}} \mathcal{L}(\hat{y}; \mathbf{x}_0).$$

*Proof of Lemma D.3.* Let $(\mathbf{x}, y) \sim \mu$. By Assumption 3.6, there exists a subset $\Omega' \subset \overline{\Omega}$ of full measure such that for all $\mathbf{x}_0 \in \Omega$,

$$\mathcal{L}(f_{\text{Bayes}}(\mathbf{x}_0); \mathbf{x}_0) = \int_{\mathbb{R}} \ell(f_{\text{Bayes}}(\mathbf{x}), y) d\nu_{\mathbf{x}_0}(y)$$

$$= \mathbb{E}[\ell(f_{\text{Bayes}}(\mathbf{x}), y) \mid \mathbf{x} = \mathbf{x}_0]$$

$$= \inf_{\hat{y} \in \mathbb{R}} \mathbb{E}[\ell(\hat{y}, y) \mid \mathbf{x} = \mathbf{x}_0]$$

$$= \inf_{\hat{y} \in \mathbb{R}} \int_{\mathbb{R}} \ell(\hat{y}, y) d\nu_{\mathbf{x}_0}(y)$$

$$= \inf_{\hat{y} \in \mathbb{R}} \mathcal{L}(\hat{y}; \mathbf{x}_0).$$

So the statement holds for all $\mathbf{x}_0 \in \Omega'$. Now suppose that $\mathbf{x}_0 \in \overline{\Omega}$. Since $\Omega'$ has full measure in $\overline{\Omega}$ and $\Omega$ is open, $\Omega'$ is dense in $\overline{\Omega}$. Let $\{\mathbf{x}_m\}_{m\in\mathbb{N}}$ be a sequence in $\Omega'$ converging to $\mathbf{x}_0$. By Sobolev embedding (Theorem A.1), $f_{\text{Bayes}}$ is continuous, and by Lemma 4.6, $\mathcal{L}$ is continuous, so the map $\mathbf{x} \mapsto \mathcal{L}(f_{\text{Bayes}}(\mathbf{x}); \mathbf{x})$ is continuous. Let $\hat{y} \in \mathbb{R}$. Then for all $m \in \mathbb{N}$,

$$\mathcal{L}(f_{\text{Bayes}}(\mathbf{x}_m); \mathbf{x}_m) \leq \mathcal{L}(\hat{y}; \mathbf{x}_m).$$

Taking the limit of both sides as $m \to \infty$ and applying continuity, we get

$$\mathcal{L}(f_{\text{Bayes}}(\mathbf{x}_0); \mathbf{x}_0) \leq \mathcal{L}(\hat{y}; \mathbf{x}_0).$$

Since this holds for all $\hat{y} \in \mathbb{R}$, we have

$$\mathcal{L}(f_{\text{Bayes}}(\mathbf{x}_0); \mathbf{x}_0) = \inf_{\hat{y}\in\mathbb{R}} \mathcal{L}(\hat{y}; \mathbf{x}_0).$$

So the condition holds for all $\mathbf{x}_0 \in \overline{\Omega}$. $\qquad\qquad\square$

With these results in hand, we can prove that the conditional loss is continuous as stated in Lemma 4.6.

*Proof of Lemma 4.6.* Let $\{(y_m, \mathbf{x}_m)\}_{m\in\mathbb{N}}$ be a sequence in $\mathbb{R} \times \overline{\Omega}$ converging to $(y_0, \mathbf{x}_0) \in \mathbb{R} \times \overline{\Omega}$. Let $\epsilon > 0$. Let $\phi : [0, \infty) \to [0, 1]$ be a continuous function such that $\phi(t) = 1$ for $t \leq 1$ and $\phi(t) = 0$ for $t \geq 2$. Let $K > 0$. Then for all $\mathbf{x} \in \overline{\Omega}$ and $\hat{y} \in [y_0 - 1, y_0 + 1]$,

$$
\begin{aligned}
\mathcal{L}(\hat{y}; \mathbf{x}) &= \int_{\mathbb{R}} \ell(\hat{y}, y) d\nu_{\mathbf{x}}(y) \\
&= \int_{\mathbb{R}} \ell(\hat{y}, y)\phi(K|y|) d\nu_{\mathbf{x}}(y) + \int_{\mathbb{R}} \ell(\hat{y}, y)(1 - \phi(K|y|)) d\nu_{\mathbf{x}}(y).
\end{aligned}
\tag{9}
$$

By Assumption 3.1, we can bound the second term as

$$
\begin{aligned}
\int_{\mathbb{R}} \ell(\hat{y}, y)(1 - \phi(K|y|)) d\nu_{\mathbf{x}}(y) &\leq \int_{\mathbb{R}} \mathbf{1}_{|y|\geq K} \ell(\hat{y}, y) d\nu_{\mathbf{x}}(y) \\
&\leq \int_{\mathbb{R}} \mathbf{1}_{|y|\geq K} C_\ell \exp\left(\tau_\ell(1 + |\hat{y}|)(1 + |y|)\right) d\nu_{\mathbf{x}}(y) \\
&\leq \int_{\mathbb{R}} \mathbf{1}_{|y|\geq K} C_\ell \exp\left(\tau_\ell(2 + |y_0|)(1 + |y|)\right) d\nu_{\mathbf{x}}(y).
\end{aligned}
$$

By Lemma D.1,

$$\int_{\mathbb{R}} C_\ell \exp\left(\tau_\ell(2 + |y_0|)(1 + |y|)\right) d\nu_{\mathbf{x}}(y) < \infty,$$

so there exists $K \in (0, \infty)$ such that for all $\mathbf{x} \in \overline{\Omega}$,

$$\int_{\mathbb{R}} \mathbf{1}_{|y|\geq K} C_\ell \exp\left(\tau_\ell(1 + |y_0|)(1 + |y|)\right) d\nu_{\mathbf{x}}(y) \leq \frac{\epsilon}{6}.$$

We fix this value of $K$ for the remainder of the proof. Substituting into (9), we get

$$\left| \mathcal{L}(\hat{y}; \mathbf{x}) - \int_{\mathbb{R}} \ell(\hat{y}, y)\phi(K|y|) d\nu_{\mathbf{x}}(y) \right| \leq \frac{\epsilon}{6}
\tag{10}$$

for all $\mathbf{x} \in \overline{\Omega}$ and $\hat{y} \in [y_0 - 1, y_0 + 1]$. The function $y \mapsto \ell(y_0, y)\phi(K|y|)$ is continuous and bounded, so by Assumption 3.2, there exists $M_1 \in \mathbb{N}$ such that for $m \geq M_1$,

$$\left| \int_{\mathbb{R}} \ell(y_0, y)\phi(K|y|) d\nu_{\mathbf{x}_m}(y) - \int_{\mathbb{R}} \ell(y_0, y)\phi(K|y|) d\nu_{\mathbf{x}}(y) \right| \leq \frac{\epsilon}{6}.
\tag{11}$$

Combining (10) and (11), we obtain

$$
\begin{aligned}
|\mathcal{L}(y_0; \mathbf{x}_m) - \mathcal{L}(y_0; \mathbf{x}_0)| &\leq \left| \mathcal{L}(y_0; \mathbf{x}_m) - \int_{\mathbb{R}} \ell(y_0, y)\phi(K|y|)d\nu_{\mathbf{x}_m}(y) \right| \\
&\quad + \left| \int_{\mathbb{R}} \ell(y_0, y)\phi(K|y|)d\nu_{\mathbf{x}_m}(y) - \int_{\mathbb{R}} \ell(y_0, y)\phi(K|y|)d\nu_{\mathbf{x}}(y) \right| \\
&\quad + \left| \mathcal{L}(y_0; \mathbf{x}_0) - \int_{\mathbb{R}} \ell(y_0, y)\phi(K|y|)d\nu_{\mathbf{x}_0}(y) \right| \\
&\leq \frac{\epsilon}{6} + \frac{\epsilon}{6} + \frac{\epsilon}{6} \\
&= \frac{\epsilon}{2}
\end{aligned}
\tag{12}
$$

for $m \geq M_1$.

There exists $M_2 \in \mathbb{N}$ such that for $m \geq M_2$, $y_m \in [y_0 - 1, y_0 + 1]$. Since $\ell$ is uniformly continuous on the compact set $[y_0 - 1, y_0 + 1] \times [-2K, 2K]$, there exists $M_3 \in \mathbb{N}$ such that for $m \geq M_3$,

$$
\sup_{y \in [-2K, 2K]} |\ell(y_m, y) - \ell(y_0, y)| \leq \frac{\epsilon}{24K}.
\tag{13}
$$

Now suppose that $m \geq \max(M_2, M_3)$. Then by (10) and (13),

$$
\begin{aligned}
|\mathcal{L}(y_m; \mathbf{x}_m) - \mathcal{L}(y_0; \mathbf{x}_m)| &\leq \int_{\mathbb{R}} |\ell(y_m, y) - \ell(y_0, y)| \, d\nu_{\mathbf{x}_m}(y) \\
&= \int_{\mathbb{R}} (1 - \phi(K|y|)) |\ell(y_m, y) - \ell(y_0, y)| \, d\nu_{\mathbf{x}_m}(y) \\
&\quad + \int_{\mathbb{R}} \phi(K|y|) |\ell(y_m, y) - \ell(y_0, y)| \, d\nu_{\mathbf{x}_m}(y)
\end{aligned}
\tag{14}
$$

We bound the two terms on the right-hand side separately. For the first term, we have

$$
\begin{aligned}
\int_{\mathbb{R}} (1 - \phi(K|y|)) |\ell(y_m, y) - \ell(y_0, y)| \, d\nu_{\mathbf{x}_m}(y) &\leq \int_{\mathbb{R}} (1 - \phi(K|y|))\ell(y_m, y)d\nu_{\mathbf{x}_m}(y) \\
&\quad + \int_{\mathbb{R}} (1 - \phi(K|y|))\ell(y_0, y)d\nu_{\mathbf{x}_m}(y) \\
&\leq \frac{\epsilon}{6} + \frac{\epsilon}{6} \qquad\qquad \text{By (10)} \\
&= \frac{\epsilon}{3}.
\end{aligned}
\tag{15}
$$

For the second term, we have

$$
\begin{aligned}
\int_{\mathbb{R}} \phi(K|y|)|\ell(y_m, y) - \ell(y_0, y)|d\nu_{\mathbf{x}_m}(y) &\leq \int_{-2K}^{2K} |\ell(y_m, y) - \ell(y_0, y)|d\nu_{\mathbf{x}_m}(y) \\
&\leq \int_{-2K}^{2K} \frac{\epsilon}{12K}d\nu_{\mathbf{x}_m}(y) \qquad \text{(By (13))} \\
&\leq \frac{\epsilon}{6}.
\end{aligned}
\tag{16}
$$

Substituting (15) and (16) into (14), we get

$$
|\mathcal{L}(y_m; \mathbf{x}_m) - \mathcal{L}(y_0; \mathbf{x}_m)| \leq \frac{\epsilon}{3} + \frac{\epsilon}{6} = \frac{\epsilon}{2}.
\tag{17}
$$

Combining (12) and (17), we see that for $m \geq \max(M_1, M_2, M_3)$,

$$
\begin{aligned}
|\mathcal{L}(y_m; \mathbf{x}_m) - \mathcal{L}(y_0; \mathbf{x}_0)| &\leq |\mathcal{L}(y_m; \mathbf{x}_m) - \mathcal{L}(y_0; \mathbf{x}_m)| + |\mathcal{L}(y_0; \mathbf{x}_m) - \mathcal{L}(y_0; \mathbf{x}_0)| \\
&\leq \frac{\epsilon}{2} + \frac{\epsilon}{2} \\
&= \epsilon.
\end{aligned}
$$

Since $\epsilon$ was arbitrary, it follows that $\mathcal{L}(y_m; \mathbf{x}_m) \to \mathcal{L}(y_0; \mathbf{x}_0)$, and therefore $\mathcal{L}$ is continuous. $\qquad\square$

### D.2. Existence of bump functions

In this section, we will review that bump functions with small enough norm exist in Sobolev spaces. If $V$ is a Banach space, we say that a map $\psi : \mathbb{R}^d \to V$ is *radially symmetric* if for all $\mathbf{x}, \mathbf{x}' \in U$ with $\|\mathbf{x}\| = \|\mathbf{x}'\|$, we have $\psi(\mathbf{x}) = \psi(\mathbf{x}')$.

**Lemma D.4.** *If $V$ is a Banach space and $\psi : \mathbb{R}^d \to V$ is a smooth radially symmetric map, then there exists a smooth function $\varphi : \mathbb{R} \to V$ such that for all $\mathbf{x} \in \mathbb{R}^d$, we have $\psi(\mathbf{x}) = \varphi(\|\mathbf{x}\|)$.*

*Proof of Lemma D.4.* Let $\mathbf{x}_0 \in \mathbb{R}^d$ be such that $\|\mathbf{x}_0\| = 1$. Let us define $\varphi : \mathbb{R} \to V$ by $\varphi(t) = \psi(t\mathbf{x}_0)$. Then $\varphi$ is smooth. If $\|\mathbf{x}\| = t$, then by the radial symmetry of $\psi$,

$$\psi(\mathbf{x}) = \psi(t\mathbf{x}_0) = \varphi(t) = \varphi(\|\mathbf{x}\|).$$

This holds for all $\mathbf{x} \in \mathbb{R}^d$, so $\varphi$ satisfies the desired property. $\qquad\square$

**Lemma D.5.** *If $V$ is a Banach space and $\psi : \mathbb{R}^d \to V$ is a smooth radially symmetric map, then $\|D^k\psi\| : \mathbb{R}^d \to \mathcal{B}((\mathbb{R}^d)^{\otimes k}, V)$ is radially symmetric. Here $\mathcal{B}$ denotes the space of bounded linear maps.*

*Proof of Lemma D.5.* Let $\mathbf{x}_1, \mathbf{x}_2 \in \mathbb{R}^d \setminus \{0\}$ with $\|\mathbf{x}_1\| = \|\mathbf{x}_2\|$. Let $\mathbf{R} : \mathbb{R}^d \to \mathbb{R}^d$ be an orthogonal linear map with $\mathbf{R}\mathbf{x}_1 = \mathbf{x}_2$. Since $\psi$ is radially symmetric, $\psi \circ \mathbf{R} = \psi$. By the chain rule,

$$\begin{aligned}
(D^k\psi)(\mathbf{x})(\xi_1, \cdots, \xi_k) &= (D^k(\psi \circ R))(\mathbf{x})(\xi_1, \cdots, \xi_k) \\
&= (D^k\psi)(\mathbf{R}\mathbf{x})(\mathbf{R}\xi_1, \cdots, \mathbf{R}\xi_k),
\end{aligned}$$

so $\|(D^k\psi(\mathbf{x}))\| = \|(D^k\psi)(\mathbf{R}\mathbf{x})\|$ for all $\mathbf{x} \in \mathbb{R}^d$. Setting $\mathbf{x} = \mathbf{x}_1$ yields $\|(D^k\psi)(\mathbf{x}_1)\| = \|(D^k\psi)(\mathbf{x}_2)\|$. Hence $\|D^k\psi\|$ is radially symmetric. $\qquad\square$

Using the previous two lemmas, we can explicitly construct bump functions with bounded Sobolev norms as shown in Lemma 4.1.

*Proof of Lemma 4.1.* It suffices to prove the statement for $\mathbf{x}_0 = \mathbf{0}_d$; the general case follows from composing $\psi$ with a translation. Let $\varphi : \mathbb{R} \to [0,1]$ be a smooth function supported on $[-1,1]$ with $\varphi(x) = 1$ for $|x| \le 1/4$. Let us define $\psi_\delta : \mathbb{R}^d \to \mathbb{R}$ by $\psi_\delta(\mathbf{x}) = \varphi\left(\frac{\|\mathbf{x}\|^2}{\delta^2}\right)$. Then $\psi_\delta$ is supported on $B(\mathbf{x}_0, \delta)$, and $\psi(\mathbf{x}) = 1$ if $\|\mathbf{x} - \mathbf{x}_0\| \le \frac{\delta}{2}$. The map $\psi_\delta$ is a composition of the maps $\mathbf{x} \mapsto \frac{\mathbf{x}}{\delta}$ and $\psi_1$. So by the chain rule, $(D^j\psi_\delta)(\mathbf{x}) = (\delta^{-j}D^j\psi_1)\left(\frac{\mathbf{x}}{\delta}\right)$ for all $j \le k$ and $\mathbf{x} \in \mathbb{R}^d$. Observe that $\psi_1$ is radially symmetric, so by Lemma D.5, $\|D^j\psi_1\|$ is radially symmetric. By Lemma D.4, there exists a smooth function $g_{j,d} : \mathbb{R} \to \mathbb{R}$ such that $\|D^j\psi_1(\mathbf{x})\| = g_{j,d}(\|\mathbf{x}\|)$ for all $\mathbf{x} \in \mathbb{R}^d$. Note that $g_{j,d}$ is supported on $(-\infty, 1]$. Then

$$\begin{aligned}
\|D^j\psi_\delta\|_{L^p(\mathbb{R}^d)}^p &= \int_{\mathbb{R}^d} \|D^j\psi_\delta(\mathbf{x})\|^p d\mathbf{x} \\
&= \int_{\mathbb{R}^d} \delta^{-jp}\|D^j\psi_1\left(\tfrac{\mathbf{x}}{\delta}\right)\|^p d\mathbf{x} \\
&= \int_{\mathbb{R}^d} \delta^{-jp}g_{j,d}\left(\tfrac{\|\mathbf{x}\|}{\delta}\right)^p d\mathbf{x} \\
&= \int_0^\infty \int_{\partial B_r(\mathbf{0}_d)} \delta^{-jp}g_{j,d}\left(\tfrac{r}{\delta}\right)^p dS(\mathbf{x})dr \\
&\lesssim_d \delta^{-jp} \int_0^\infty r^{d-1}g_{j,d}\left(\tfrac{r}{\delta}\right)^p dr.
\end{aligned}$$

By a change of variables, the last line above is equal to

$$\delta^{-jp+1} \int_0^\infty (\delta r)^{d-1}g_{j,d}(r)^p dr.$$

Since $g_{j,d}$ is supported on $(-\infty, 1]$, the above expression is equal to

$$\delta^{-jp+1} \int_0^1 (\delta r)^{d-1} g_{j,d}(r)^p dr = \delta^{d-jp} \int_0^1 r^{d-1} g_{j,d}(r)^p dr$$
$$\lesssim_{k,d,p} \delta^{d-jp}.$$

Hence,

$$\|D^j \psi_\delta\|_{L^p(\mathbb{R}^d)} \le c_{k,d,p} \delta^{(d-jp)/p} \le c_{k,d,p} \max\left(1, \delta^{(d-jp)/p}\right).$$

Summing over $j$, we get

$$\|\psi_\delta\|_{W^{k,p}} \lesssim_{k,d,p} \sum_{j=0}^k \|D^j \psi_\delta\|_{L^p(\mathbb{R}^d)}$$
$$\lesssim_{k,d,p} \sum_{j=0}^k \max\left(1, \delta^{(d-jp)/p}\right)$$
$$\lesssim_{k,d,p} \sum_{j=0}^k \max\left(1, \delta^{(d-kp)/p}\right)$$
$$\lesssim_{k,d,p} \max\left(1, \delta^{(d-kp)/p}\right)$$
$$\lesssim_{k,d,p} 1 + \delta^{(d-kp)/p}.$$

So $\psi_\delta$ satisfies the desired properties. $\qquad\square$

As mentioned in Section 4.1, we need Lemma 4.2 to ensure that the supports of the bump functions are disjoint.

*Proof of Lemma 4.2.* Let $i, j \in [n]$ with $i \ne j$. Without loss of generality, suppose that $\delta_i \le \delta_j$. Suppose that there exists $\mathbf{x} \in B(\mathbf{x}_i, \delta_i/2) \cap B(\mathbf{x}_j, \delta_j/2)$. Then by the triangle inequality, $\|\mathbf{x}_i - \mathbf{x}_j\| < \frac{\delta_i}{2} + \frac{\delta_j}{2} \le \delta_j$. This contradicts the definition of $\delta_j$. So $B(\mathbf{x}_i, \delta_i/2) \cap B(\mathbf{x}_j, \delta_j/2) = \emptyset$. $\qquad\square$

Finally, we can use our interpolant to partially bound the min norm solution.

**Lemma D.6.** *Let $f^*$ be a solution to (1). Then*

$$\|f^*\|_{W^{k,p}}^p \lesssim_{k,d,p} \sum_{i=1}^n \left(1 + |y_i|^p \delta_i^{d-kp}\right).$$

*Proof of Lemma D.6.* By Lemma 4.1, for each $i \in [n]$, let $\psi_i \in C^\infty(\mathbb{R}^d)$ be a function supported on $B(\mathbf{x}_i, \delta_i/2)$ with $\psi_i(\mathbf{x}_i) = 1$ and $\|\psi_i\|_{W^{k,p}} \lesssim_{k,d,p} 1 + \delta_i^{(d-kp)/p}$. Then by the definition of $\delta_i$, $\psi_i(\mathbf{x}_\ell) = 0$ for all $\ell \ne i$.

Let us define $f \in W^{k,p}(\mathbb{R}^d)$ by

$$f = \sum_{i=1}^n y_i \psi_i.$$

Then $f(\mathbf{x}_i) = y_i$ for all $i \in [n]$. By Lemma 4.2, the sets $B(\mathbf{x}_i, \delta_i/2)$ are disjoint, so for all $j \in \{0, \cdots, k\}$,

$$\|D^j f\|_{L^p(\mathbb{R}^d)}^p \lesssim_{k,d,p} \left\|\sum_{i=1}^n |y_i| D^j \psi_i\right\|_{L^p(\mathbb{R}^d)}^p$$
$$= \sum_{i=1}^n |y_i|^p \|D^j \psi_i\|_{L^p(B(\mathbf{x}_i, \delta_i/2))}^p$$
$$\lesssim_{k,d,p,\Omega} \sum_{i=1}^n (1 + |y_i|^p \delta_i^{d-kp}).$$

Summing over all $j$, we get

$$\|f\|_{W^{k,p}}^p \lesssim_{k,d,p} 1 + \sum_{i=1}^n (1 + |y_i|^p \delta_i^{d-kp}).$$

Finally, by the definition of $f^*$,

$$\|f^*\|_{W^{k,p}(\mathbb{R}^d)}^p \leq \|f\|_{W^{k,p}(\mathbb{R}^d)}^p$$

$$\lesssim_{k,d,p} \sum_{i=1}^n \left(1 + |y_i|^p \delta_i^{d-kp}\right).$$

$\square$

### D.3. Properties of the nearest neighbor balls

The previous section bounds minimum-norm solutions using the nearest neighbor radii, however, the bound in Corollary 4.3 relies on concentration inequalities to control the $\delta_i$. We would like to apply a law of large numbers, but the $\delta_i$s are not independent, since they each depend on the entire dataset. We instead show that changing a single data point does not alter too many of the $\delta_i$s. We can then apply McDiarmid's inequality to bound their sum concentration around the mean.

We will need the following version of McDiarmid's inequality for high-probability subsets (Combes, 2024, Proposition 2).

**Theorem D.7.** *Let $\mathcal{X}_1, \cdots, \mathcal{X}_n$ be measurable spaces, and let $X_1, \cdots, X_n$ be independent random variables with $X_i$ taking values in $\mathcal{X}_i$. Let $\mathcal{Y} \subset \mathcal{X}_1 \times \cdots \times \mathcal{X}_n$ be measurable, and let us define*

$$p = \mathbb{P}((X_1, \cdots, X_n) \notin \mathcal{Y}).$$

*Let $C_1, \cdots, C_n > 0$. Let $g : \mathcal{X}_1 \times \cdots \times \mathcal{X}_n \to \mathbb{R}$ be a measurable function such that for all $i \in [n]$ and all $x_1 \in \mathcal{X}_1, \cdots, x_n \in \mathcal{X}_n, x_i' \in \mathcal{X}_i$ with $(x_1, \cdots, x_n), (x_1, \cdots, x_{i-1}, x_i', x_{i+1}, \cdots, x_n) \in \mathcal{Y}$,*

$$|g(x_1, \cdots, x_{i-1}, x_i, x_{i+1}, \cdots, x_n) - g(x_1, \cdots, x_{i-1}, x_i', x_{i+1}, \cdots, x_n)| \leq C_i.$$

*Then for all $t > 0$,*

$$\mathbb{P}\left(g(X_1, \cdots, X_n) - \mathbb{E}[g(X_1, \cdots, X_n)|(X_1, \cdots, X_n) \in \mathcal{Y}] \geq t + p \sum_{i=1}^n C_i\right) \leq p + \exp\left(-\frac{2t^2}{\sum_{i=1}^n C_i^2}\right).$$

For a dataset $\mathbf{X} \in \mathbb{R}^{n \times d}$ with distinct points, we define its *nearest neighbor graph* $NN(\mathbf{X})$ to be the directed graph with vertex set $\mathbf{x}_1, \cdots, \mathbf{x}_n$, with a directed edge from $\mathbf{x}_i$ to $\mathbf{x}_j$ if $\|\mathbf{x}_j - \mathbf{x}_i\| \leq \|\mathbf{x}_\ell - \mathbf{x}_i\|$ for all $\ell \neq i$.

**Lemma D.8.** *Let $\mathbf{X} \in \mathbb{R}^{n \times d}$ be a dataset with distinct points. Then for all $\ell \in [n]$, the in-degree $\deg^-(\mathbf{x}_\ell)$ satisfies*

$$\deg^-(\mathbf{x}_\ell) \lesssim_d 1.$$

*Proof of Lemma D.8.* Let $\mathcal{N}$ denote the set all $i \in [n]$ such that $(i, \ell)$ is an edge of $NN(\mathbf{X})$. We follow the argument by (Eppstein et al., 1997) relating the cardinality of $\mathcal{N}$ to the *kissing number* in dimension $d$. We define the kissing number $\tau(d)$ to be the maximum value $M$ such that there exist points $\mathbf{u}_1, \cdots, \mathbf{u}_M$ with $\|\mathbf{u}_i\| = 1$ for all $i$, and $\|\mathbf{u}_i - \mathbf{u}_j\| \geq 1$ for all $i \neq j$. The kissing number is finite for all $d \in \mathbb{N}$. To see this, observe that the sets $B(\mathbf{u}_i, 1)$ are disjoint and contained in $B(\mathbf{0}_d, 2)$ which has finite measure.

Now for $i \in \mathcal{N}$ let us define $\mathbf{w}_i = \mathbf{x}_i - \mathbf{x}_\ell$ and $\mathbf{u}_i = \frac{\mathbf{w}_i}{\|\mathbf{w}_i\|}$. Suppose that $i, j \in \mathcal{N}$ with $\|\mathbf{x}_i - \mathbf{x}_\ell\|^2 \leq \|\mathbf{x}_j - \mathbf{x}_\ell\|^2$. Since $i, j \in \mathcal{N}$, we have $\|\mathbf{x}_j - \mathbf{x}_\ell\|^2 \leq \|\mathbf{x}_i - \mathbf{x}_j\|^2$. Then

$$2\langle \mathbf{x}_i - \mathbf{x}_\ell, \mathbf{x}_j - \mathbf{x}_\ell\rangle = \|\mathbf{x}_i - \mathbf{x}_\ell\|^2 + \|\mathbf{x}_j - \mathbf{x}_\ell\|^2 - \|\mathbf{x}_i - \mathbf{x}_j\|^2$$

$$\leq \|\mathbf{x}_i - \mathbf{x}_\ell\|^2.$$

In other words,

$$2\langle \mathbf{w}_i, \mathbf{w}_j \rangle \leq \|\mathbf{w}_i\|^2.$$

Dividing both sides by $2\|\mathbf{w}_i\|\|\mathbf{w}_j\|$, we get

$$\langle \mathbf{u}_i, \mathbf{u}_j \rangle \leq \frac{1}{2} \frac{\|\mathbf{w}_i\|}{\|\mathbf{w}_j\|} \leq \frac{1}{2}.$$

Then

$$\begin{aligned}
\|\mathbf{u}_i - \mathbf{u}_j\|^2 &= \|\mathbf{u}_i\|^2 + \|\mathbf{u}_j\|^2 - 2\langle \mathbf{u}_i, \mathbf{u}_j \rangle \\
&= 1 + 1 - 2\langle \mathbf{u}_i, \mathbf{u}_j \rangle \\
&\geq 1.
\end{aligned}$$

So the set $\mathcal{N}' = \{\mathbf{u}_i : i \in \mathcal{N}\}$ consists of unit vectors with $\|\mathbf{u}_i - \mathbf{u}_j\| \geq 1$ for all $i \neq j$, and therefore has cardinality at most $\tau(d)$.

We claim that for all $i, j \in \mathcal{N}$ with $i \neq j$, $\mathbf{u}_i \neq \mathbf{u}_j$. To see this, suppose otherwise. Then there exist $i, j \in \mathcal{N}$ with $i \neq j$ and $\mathbf{x}_i - \mathbf{x}_\ell = \lambda(\mathbf{x}_j - \mathbf{x}_\ell)$ for some $\lambda > 1$. This implies that

$$\begin{aligned}
\|\mathbf{x}_i - \mathbf{x}_j\| &= \|(\mathbf{x}_i - \mathbf{x}_\ell) - (\mathbf{x}_j - \mathbf{x}_\ell)\| \\
&= \left\|\left(1 - \frac{1}{\lambda}\right)(\mathbf{x}_i - \mathbf{x}_\ell)\right\| \\
&< \|\mathbf{x}_i - \mathbf{x}_\ell\|.
\end{aligned}$$

This contradicts that $\mathbf{x}_\ell$ is the closest point in the dataset to $\mathbf{x}_i$. So for all $i \neq j$, $\mathbf{u}_i \neq \mathbf{u}_j$. This implies that $|\mathcal{N}'| = |\mathcal{N}|$. Putting everything together, we have

$$\deg^-(\mathbf{x}_\ell) = |\mathcal{N}| = |\mathcal{N}'| \leq \tau(d).$$

$\square$

In order to use McDiarmid's inequality, we must show that the functions of the $\delta_i$s satisfy a bounded difference property. To do so, we show that changing any single data point cannot change too many of the nearest neighbor radii.

**Lemma D.9.** *Let $\mathbf{X} \in \mathbb{R}^{n \times d}$ be a dataset with distinct points and fix $\ell \in [n]$. Let $\mathbf{X}' \in \mathbb{R}^{n \times d}$ be a dataset with distinct points such that $\mathbf{x}_i = \mathbf{x}'_i$ for all $i \neq \ell$. Then*

$$|\{i \in [n] : \delta_i(\mathbf{X}) \neq \delta_i(\mathbf{X}')\}| \lesssim_d 1.$$

*Proof of Lemma D.9.* Suppose that $\delta_i(\mathbf{X}) \neq \delta_i(\mathbf{X}')$ for $i \neq \ell$. Then the closest point to $\mathbf{x}_i$ changes as we modify $\mathbf{x}_\ell$ to $\mathbf{x}'_\ell$. This can only happen if $\mathbf{x}_\ell$ is the nearest neighbor of $\mathbf{x}_i$ in the dataset $\mathbf{X}$, or if $\mathbf{x}'_\ell$ is the nearest neighbor of $\mathbf{x}_i$ in the dataset $\mathbf{X}'$. In other words, either $(\mathbf{x}_i, \mathbf{x}_\ell)$ is an edge in $NN(\mathbf{X})$, or $(\mathbf{x}_i, \mathbf{x}'_\ell)$ is an edge in $NN(\mathbf{X}')$. By Lemma D.8, the number of such $i \in [n]$ is at most a constant $C_d$. So

$$\begin{aligned}
|\{i \in [n] : \delta_i(\mathbf{X}) \neq \delta_i(\mathbf{X}')| &\leq 1 + |\{i \in [n] \setminus \{\ell\} : \delta_i(\mathbf{X}) \neq \delta_i(\mathbf{X}')\}| \\
&\leq 1 + 2C_d \\
&\lesssim_d 1.
\end{aligned}$$

$\square$

We can then apply the previous lemma to bound the sum in Lemma D.6.

**Lemma D.10.** *Let $\beta \in (0, d/2)$ and let $\epsilon \in (0, 1)$. If $n \geq \text{Poly}_{\beta,d}(\epsilon^{-1})$, then with probability at least $1 - \epsilon$,*

$$\sum_{i=1}^{n} |y_i|^p \delta_i^{-\beta} \lesssim_{d,\beta} n^{\beta/d+1}.$$

*Proof of Lemma D.10.* Before we bound the sum of the terms, we bound the maximum.

For all $i \in [n]$ and $t > 0$,

$$
\begin{aligned}
\mathbb{P}(\delta_i^{-\beta} \geq t) &= \mathbb{P}(\delta_i \leq t^{-1/\beta}) \\
&\leq \sum_{j \neq i} \mathbb{P}(\|\mathbf{x}_i - \mathbf{x}_j\| \leq t^{-1/\beta}) \\
&\leq \sum_{j \neq i} C_1 t^{-d/\beta} \qquad \text{(By Assumption 3.3)} \\
&\leq C_1 n t^{-d/\beta},
\end{aligned}
\tag{18}
$$

where $C_1$ is a constant depending on $d$ and $\mu$. By a union bound,

$$
\mathbb{P}(\max(\delta_1^{-\beta}, \cdots, \delta_n^{-\beta}) \geq t) \leq C_1 n^2 t^{-d/\beta}.
$$

Setting $t = \left(\frac{4C_1 n^2}{\epsilon}\right)^{\beta/d}$, we get that with probability at least $1 - \frac{\epsilon}{4}$,

$$
\max(\delta_1^{-\beta}, \cdots, \delta_n^{-\beta}) < \left(\frac{4C_1 n^2}{\epsilon}\right)^{\beta/d}.
\tag{19}
$$

By Assumption 3.4, there exists a constant $C_2 > 0$ such that for all $i \in [n]$ and $t \geq 0$,

$$
\mathbb{P}(|y_i| \geq t) \leq 2 \exp\left(-\frac{t^2}{C_2^2}\right).
$$

By a union bound,

$$
\mathbb{P}(\max(|y_1|, \cdots, |y_n|) \geq t) \leq 2n \exp\left(-\frac{t^2}{C_2^2}\right).
$$

Setting $t = C_2 \sqrt{\log \frac{8n}{\epsilon}}$ yields that with probability at least $1 - \frac{\epsilon}{4}$,

$$
\max(|y_1|, \cdots, |y_n|) < C_2 \sqrt{\log \frac{8n}{\epsilon}}.
\tag{20}
$$

Now we apply the high probability McDiarmid's inequality (Theorem D.7). Let $\mathcal{X} = \Omega \times \mathbb{R}$. Let $g : \mathcal{X}^n \to \mathbb{R}$ be defined by

$$
g((\mathbf{x}_1, y_1), \cdots, (\mathbf{x}_n, y_n)) = \sum_{i=1}^{n} |y_i|^p \delta_i(\mathbf{x}_1, \cdots, \mathbf{x}_n)^{-\beta}
$$

if the $\mathbf{x}_i$ are all distinct. If $\mathbf{x}_i = \mathbf{x}_j$ for some $i, j \in [n]$ with $i \neq j$, we arbitrarily define $g((\mathbf{x}_1, y_1), \cdots, (\mathbf{x}_n, y_n)) = 0$. We similarly define a thresholded function $\tilde{g} : \mathcal{X}^n \to \mathbb{R}$ by

$$
\tilde{g}((\mathbf{x}_1, y_1), \cdots, (\mathbf{x}_n, y_n)) = \sum_{i=1}^{n} \min\left(|y_i|, C_2 \sqrt{\log \frac{8n}{\epsilon}}\right)^p \min\left(\delta_i(\mathbf{x}_1, \cdots, \mathbf{x}_n)^{-\beta}, \left(\frac{4C_1 n^2}{\epsilon}\right)^{\beta/d}\right).
$$

By (19) and (20), with probability at least $1 - \frac{\epsilon}{2}$ we have

$$
g((\mathbf{x}_1, y_1), \cdots, (\mathbf{x}_n, y_n)) = \tilde{g}((\mathbf{x}_1, y_1), \cdots, (\mathbf{x}_n, y_n)).
$$

We denote the event that this occurs by $\omega_1$.

Let $\mathcal{Y} \subset \mathcal{X}^n$ be the set of all $((\mathbf{x}_1, y_1), \cdots, (\mathbf{x}_n, y_n)) \in (\Omega \times \mathbb{R})^n$ such that the points $\mathbf{x}_1, \cdots, \mathbf{x}_n$ are distinct, so that

$$
\mathbb{P}(((\mathbf{x}_1, y_1), \cdots, (\mathbf{x}_n, y_n)) \notin \mathcal{Y}) = 0.
$$

To apply McDiarmid's inequality to $\tilde{g}$, we need to check that the bounded difference property holds for $\tilde{g}$. Suppose that $i \in [n]$, and $(\mathbf{x}_1, y_1), \cdots, (\mathbf{x}_n, y_n), (\mathbf{x}'_i, y'_i) \in \mathcal{X}$ with

$$(\mathbf{X}, \mathbf{y}) := ((\mathbf{x}_1, y_1), \cdots, (\mathbf{x}_n, y_n)) \in \mathcal{Y}$$

and

$$(\mathbf{X}', \mathbf{y}') := ((\mathbf{x}_1, y_1), \cdots, (\mathbf{x}_{i-1}, y_{i-1}), (\mathbf{x}'_i, y'_i), (\mathbf{x}_{i+1}, y_{i+1}), \cdots, (\mathbf{x}_n, y_n)) \in \mathcal{Y}.$$

Then

$$
\begin{aligned}
&|\tilde{g}(\mathbf{X}, \mathbf{y}) - \tilde{g}(\mathbf{X}', \mathbf{y}')| \\
&\leq \sum_{j \neq i} \left| \min\left(|y_j|, C_2\sqrt{\log\frac{8n}{\epsilon}}\right)^p \min\left(\delta_j(\mathbf{X})^{-\beta}, \left(\frac{4C_1 n^2}{\epsilon}\right)^{\beta/d}\right) \right. \\
&\qquad - \min\left(|y_j|, C_2\sqrt{\log\frac{8n}{\epsilon}}\right)^p \min\left(\delta_j(\mathbf{X}')^{-\beta}, \left(\frac{4C_1 n^2}{\epsilon}\right)^{\beta/d}\right) \Bigg| \\
&\quad + \left| \min\left(|y_i|, C_2\sqrt{\log\frac{8n}{\epsilon}}\right)^p \min\left(\delta_i(\mathbf{X})^{-\beta}, \left(\frac{4C_1 n^2}{\epsilon}\right)^{\beta/d}\right) \right. \\
&\qquad - \min\left(|y'_i|, C_2\sqrt{\log\frac{8n}{\epsilon}}\right)^p \min\left(\delta_i(\mathbf{X}')^{-\beta}, \left(\frac{4C_1 n^2}{\epsilon}\right)^{\beta/d}\right) \Bigg|.
\end{aligned}
\tag{21}
$$

The above expression consists of two quantities: a sum over indices $j \neq i$, and the $i$ term. We bound these two quantities separately. Let $\mathcal{S} \subset [n]$ denote the subset of indices $j \neq i$ for which $\delta_j(\mathbf{X}) \neq \delta_j(\mathbf{X}')$. By Lemma D.9, there exists a constant $C_3 > 1$ depending only on $d$ such that $|\mathcal{S}| \leq C_3$. Then

$$
\begin{aligned}
&\sum_{j \neq i} \left| \min\left(|y_j|, C_2\sqrt{\log\frac{8n}{\epsilon}}\right)^p \min\left(\delta_j(\mathbf{X})^{-\beta}, \left(\frac{4C_1 n^2}{\epsilon}\right)^{\beta/d}\right) \right. \\
&\qquad - \min\left(|y_j|, C_2\sqrt{\log\frac{8n}{\epsilon}}\right)^p \min\left(\delta_j(\mathbf{X}')^{-\beta}, \left(\frac{4C_1 n^2}{\epsilon}\right)^{\beta/d}\right) \Bigg| \\
&= \sum_{j \in \mathcal{S}} \min\left(|y_j|, C_2\sqrt{\log\frac{8n}{\epsilon}}\right)^p \left| \min\left(\delta_j(\mathbf{X})^{-\beta}, \left(\frac{4C_1 n^2}{\epsilon}\right)^{\beta/d}\right) \right. \\
&\qquad - \min\left(\delta_j(\mathbf{X}')^{-\beta}, \left(\frac{4C_1 n^2}{\epsilon}\right)^{\beta/d}\right) \Bigg| \\
&\leq 2\sum_{j \in \mathcal{S}} \min\left(|y_j|, C_2\sqrt{\log\frac{8n}{\epsilon}}\right)^p \left(\frac{4C_1 n^2}{\epsilon}\right)^{\beta/d} \\
&\leq 2\sum_{j \in \mathcal{S}} \left(C_2\sqrt{\log\frac{8n}{\epsilon}}\right)^p \left(\frac{4C_1 n^2}{\epsilon}\right)^{\beta/d} \\
&\leq 2C_3 \left(C_2\sqrt{\log\frac{8n}{\epsilon}}\right)^p \left(\frac{4C_1 n^2}{\epsilon}\right)^{\beta/d}.
\end{aligned}
\tag{22}
$$

To bound the second term, we observe that it is a difference of bounded expressions:

$$
\begin{aligned}
&\left| \min\left( |y_i|, C_2\sqrt{\log\frac{8n}{\epsilon}} \right)^p \min\left( \delta_i(\mathbf{X})^{-\beta}, \left(\frac{4C_1n^2}{\epsilon}\right)^{\beta/d} \right) \right. \\
&\left. - \min\left( |y_i'|, C_2\sqrt{\log\frac{8n}{\epsilon}} \right)^p \min\left( \delta_i(\mathbf{X}')^{-\beta}, \left(\frac{4C_1n^2}{\epsilon}\right)^{\beta/d} \right) \right| \\
&\leq 2\left( C_2\sqrt{\log\frac{8n}{\epsilon}} \right)^p \left(\frac{4C_1n^2}{\epsilon}\right)^{\beta/d} \\
&\leq 2C_3\left( C_2\sqrt{\log\frac{8n}{\epsilon}} \right)^p \left(\frac{4C_1n^2}{\epsilon}\right)^{\beta/d}.
\end{aligned}
\tag{23}
$$

Substituting (22) and (23) into (21), we get

$$
\begin{aligned}
|\tilde{g}(\mathbf{X}, \mathbf{y}) - \tilde{g}(\mathbf{X}', \mathbf{y}')| &\leq 4C_3\left( C_2\sqrt{\log\frac{8n}{\epsilon}} \right)^p \left(\frac{4C_1n^2}{\epsilon}\right)^{\beta/d} \\
&= 4C_3 C_2^p \left(\frac{\log 8n}{\epsilon}\right)^p \left(\frac{4C_1n^2}{\epsilon}\right)^{\beta/d}.
\end{aligned}
$$

So $\tilde{g}$ satisfies the bounded difference property. We denote

$$
\alpha = 4C_3 C_2^p \left(\frac{4C_1n^2}{\epsilon}\right)^{\beta/d} \left(\log\frac{8n}{\epsilon}\right)^{p/2}.
$$

By Theorem D.7, for all $t > 0$,

$$
\mathbb{P}\left( \tilde{g}(\mathbf{X}, \mathbf{y}) - \mathbb{E}[\tilde{g}(\mathbf{X}, \mathbf{y}) \mid (\mathbf{X}, \mathbf{y}) \in \mathcal{Y}] \geq t \right) \leq \exp\left( -\frac{2t^2}{n\alpha^2} \right).
$$

Note that since $\mathbb{P}((\mathbf{X}, \mathbf{y}) \in \mathcal{Y}) = 1$,

$$
\mathbb{E}[\tilde{g}(\mathbf{X}, \mathbf{y}) \mid (\mathbf{X}, \mathbf{y}) \in \mathcal{Y}] = \mathbb{E}[\tilde{g}(\mathbf{X}, \mathbf{y})].
$$

Setting $t = \alpha \left(\frac{n}{2}\log\frac{2}{\epsilon}\right)^{1/2}$ yields that with probability at least $1 - \frac{\epsilon}{2}$,

$$
\tilde{g}(\mathbf{X}, \mathbf{y}) \leq \mathbb{E}[\tilde{g}(\mathbf{X}, \mathbf{y})] + \alpha\left(\frac{n}{2}\log\frac{2}{\epsilon}\right)^{1/2}.
\tag{24}
$$

We denote the event that this occurs by $\omega_2$, and will further bound the quantity $\mathbb{E}[\tilde{g}(\mathbf{X}, \mathbf{y})]$ in (24). By definition, $\tilde{g}(\mathbf{X}, \mathbf{y}) \leq g(\mathbf{X}, \mathbf{y})$ almost surely, so

$$
\begin{aligned}
\mathbb{E}[\tilde{g}(\mathbf{X}, \mathbf{y})] &\leq \mathbb{E}[g(\mathbf{X}, \mathbf{y})] \\
&= \sum_{i=1}^n \mathbb{E}[|y_i|^p \delta_i(\mathbf{X})^{-\beta}] \\
&= n\mathbb{E}[|y_1|^p \delta_1(\mathbf{X})^{-\beta}] \\
&= n\mathbb{E}[\mathbb{E}[|y_1|^p \delta_1(\mathbf{X})^{-\beta} \mid \mathbf{X}]] \\
&= n\mathbb{E}[\delta_1(\mathbf{X})^{-\beta}\mathbb{E}[|y_1|^p \mid \mathbf{X}]].
\end{aligned}
$$

Since $y_1$ is independent of $\mathbf{X}$ conditional on $\mathbf{x}_1$,

$$
n\mathbb{E}[\delta_1(\mathbf{X})^{-\beta}\mathbb{E}[|y_1|^p \mid \mathbf{X}]] = n\mathbb{E}[\delta_1(\mathbf{X})^{-\beta}\mathbb{E}[|y_1|^p \mid \mathbf{x}_1]].
$$

Since $y_1$ is sub-Gaussian conditional on $\mathbf{x}_1$ (Assumption 3.4), $\mathbb{E}[|y_1|^p \mid \mathbf{x}_1] \lesssim p^{p/2} \lesssim_p 1$ almost surely (see Proposition 2.6.1 of (Vershynin, 2018)), so

$$n\mathbb{E}[\delta_1(\mathbf{X})^{-\beta}\mathbb{E}[|y_1|^p \mid \mathbf{x}_1]] \lesssim_p n\mathbb{E}[\delta_1(\mathbf{X})^{-\beta}]$$

$$= n\int_0^\infty \mathbb{P}(\delta_1^{-\beta} > t)dt$$

$$\leq n\int_0^{n^{\beta/d}} 1 dt + n\int_{n^{\beta/d}}^\infty \mathbb{P}(\delta_1^{-\beta} > t)dt$$

$$= n^{\beta/d+1} + n\int_{n^{\beta/d}}^\infty \mathbb{P}(\delta_i < t^{-1/\beta})dt$$

$$\lesssim_{d,\beta} n^{\beta/d+1} + n\int_{n^{\beta/d}}^\infty nt^{-d/\beta}dt \qquad \text{(By (18))}$$

$$\lesssim_{d,\beta} n^{\beta/d+1} + n^2\left(n^{\beta/d}\right)^{-d/\beta+1}$$

$$\lesssim n^{\beta/d+1}.$$

Hence, there exists a constant $C_4 > 0$ depending on $d, p, \beta$ such that

$$\mathbb{E}[\tilde{g}(\mathbf{X},\mathbf{y})] \leq C_4 n^{\beta/d+1}.$$

If both $\omega_1$ and $\omega_2$ occur (which happens with probability at least $1 - \epsilon$), then substituting the above inequality into 24 yields

$$\sum_{i=1}^n |y_i|^p \delta_i(\mathbf{X})^{-\beta} = g(\mathbf{X},\mathbf{y})$$

$$\leq \tilde{g}(\mathbf{X},\mathbf{y})$$

$$\leq C_4 n^{\beta/d+1} + \alpha\left(\frac{n}{2}\log\frac{2}{\epsilon}\right)^{1/2}.$$

Now if

$$n \geq (\epsilon^{-1})^{(3/4+\beta/(2d))/(1/4-\beta/(2d))},$$

then

$$\alpha\left(\frac{n}{2}\log\frac{2}{\epsilon}\right)^{1/2} \lesssim_{\beta,d,p} n^{2\beta/d+1/2}\left(\log\frac{n}{\epsilon}\right)^{p/2}\left(\frac{1}{\epsilon}\right)^{\beta/d}\left(\log\frac{1}{\epsilon}\right)^{1/2}$$

$$\lesssim_{\beta,d,p} n^{2\beta/d+1/2}\left(\frac{n}{\epsilon}\right)^{1/4-\beta/(2d)}\left(\frac{1}{\epsilon}\right)^{\beta/d}\left(\frac{1}{\epsilon}\right)^{1/2} \qquad \text{(Since } \beta < d/2)$$

$$= n^{(3\beta)/(2d)+3/4}(\epsilon^{-1})^{3/4+\beta/(2d)}$$

$$\leq n^{(3\beta)/(2d)+3/4}n^{1/4-\beta/(2d)}$$

$$= n^{\beta/d+1}$$

and therefore

$$\sum_{i=1}^n |y_i|^p \delta_i(\mathbf{X})^{-\beta} \lesssim_{d,p,\beta} n^{\beta/d+1}.$$

This happens with probability at least $1 - \epsilon$, which establishes the result. $\qquad \square$

Corollary 4.3 then follows trivially.

*Proof of Corollary 4.3.* As $k \in (d/p, 1.5d/p)$ by assumption, we have that $d - kp > 0$. The corollary then follows from Lemma D.10 and Lemma D.6 with $\beta = d - kp$. $\qquad \square$

To finish this section, we prove Lemma 4.4, as it uses similar techniques to the proof of Lemma D.10.

*Proof of Lemma 4.4.* We follow the proof technique of Lemma D.10, using that the $\delta_i$s are stable under changes of a single data point. Since the density $\rho_{\mathbf{x}}$ is bounded above and below by constants (Assumption 3.3), there exists a constant $C_1 \in (0, \infty)$ depending only on $d$ such that

$$\mathbb{P}(\|\mathbf{x} - \mathbf{x}'\| < \delta) \leq C_1 \delta^d \tag{25}$$

for all $\delta > 0$ and all $\mathbf{x}, \mathbf{x}'$ drawn i.i.d. from the data distribution $\mu_{\mathbf{x}}$. For $i \in [n]$, we define the following Bernoulli random variables. Let $Z_i$ be take the value 1 if $\delta_i \geq \left(\frac{1}{2C_1 n}\right)^{1/d}$ and the value 0 otherwise. Let $Y_i$ take the value 1 if

$$|y_i| \leq C_y \sqrt{\log \frac{4}{\rho}}$$

(where $C_y$ is as defined in Asssumption 3.4), and the value 0 otherwise. Let $W_i$ take the value 1 if

$$\mathcal{L}(y_i; \mathbf{x}_i) \geq \sigma + \inf_{\hat{y} \in \mathbb{R}} \mathcal{L}(\hat{y}; \mathbf{x}_i),$$

and the value 0 otherwise. Let $\mathcal{B} \subset [n]$ be the subset of all indices such that $Z_i Y_i W_i = 1$. We show that $\mathcal{B}$ satisfies the desired conditions with high probability. Since $Z_i = 1$ for all $i \in \mathcal{B}$, $\mathcal{B}$ satisfies condition 2. Since $Y_i = 1$ for all $i \in \mathcal{B}$, $\mathcal{B}$ satisfies condition 3. Since $W_i = 1$ for all $i \in \mathcal{B}$,

$$\mathcal{L}(y_i; \mathbf{x}_i) \geq \sigma + \inf_{\hat{y} \in \mathbb{R}} \mathcal{L}(\hat{y}; \mathbf{x}_i) = \sigma + \mathcal{L}(f_{\text{Bayes}}(\mathbf{x}_i); \mathbf{x}_i),$$

where the equality is an application of Lemma D.3. So $\mathcal{B}$ satisfies condition 4. It remains to show that $\mathcal{B}$ satisfies condition 1 with high probability. To this end, we show that the sum

$$|\mathcal{B}| = \sum_{i=1}^{n} Z_i Y_i W_i$$

concentrates, and start by analyzing the expectation of the terms. By Assumption 3.4,

$$\begin{aligned}
\mathbb{E}[Y_i \mid \mathbf{X}] &= \mathbb{P}\left(|y_i| \leq 2C_y \sqrt{\log \frac{1}{\rho}} \,\middle|\, \mathbf{X}\right) \\
&= \mathbb{P}\left(|y_i| \leq 2C_y \sqrt{\log \frac{1}{\rho}} \,\middle|\, \mathbf{x}_i\right) \\
&\geq 1 - \frac{\rho}{2}
\end{aligned} \tag{26}$$

almost surely. By Assumption 3.5,

$$\begin{aligned}
\mathbb{E}[W_i \mid \mathbf{X}] &= \mathbb{P}\left(\mathcal{L}(y_i; \mathbf{x}_i) \geq \sigma + \inf_{\hat{y} \in \mathbb{R}} \mathcal{L}(\hat{y}; \mathbf{x}_i) \,\middle|\, \mathbf{X}\right) \\
&= \mathbb{P}\left(\mathcal{L}(y_i; \mathbf{x}_i) \geq \sigma + \inf_{\hat{y} \in \mathbb{R}} \mathcal{L}(\hat{y}; \mathbf{x}_i) \,\middle|\, \mathbf{x}_i\right) \\
&\geq \rho
\end{aligned} \tag{27}$$

almost surely. Combining (26) and (27), we get

$$\begin{aligned}
\mathbb{E}[Y_i W_i \mid \mathbf{X}] &= \mathbb{P}(Y_i = W_i = 1 \mid \mathbf{X}) \\
&\geq 1 - \mathbb{P}(Y_i = 0 \mid \mathbf{X}) - \mathbb{P}(W_i = 0 \mid \mathbf{X}) \\
&= \mathbb{E}[Y_i \mid \mathbf{X}] + \mathbb{E}[W_i \mid \mathbf{X}] - 1 \\
&\geq \frac{\rho}{2}
\end{aligned} \tag{28}$$

almost surely. Then

$$
\begin{aligned}
\mathbb{E}[Z_i Y_i W_i] &= \mathbb{E}[\mathbb{E}[Z_i Y_i W_i \mid \mathbf{X}]] \\
&= \mathbb{E}[Z_i \mathbb{E}[Y_i W_i \mid \mathbf{X}]] && \text{(Since } Z_i \text{ is } \mathbf{X}\text{-measurable)} \\
&\geq \frac{\rho}{2} \mathbb{E}[Z_i] && \text{(By 28)} \\
&= \frac{\rho}{2} \mathbb{P}(\delta_i \geq (2C_1 n)^{-1/d}) \\
&= \frac{\rho}{2} \left( 1 - \mathbb{P}(\exists j \neq i : \|\mathbf{x}_i - \mathbf{x}_j\| < (2C_1 n)^{-1/d}) \right) \\
&\geq \frac{\rho}{2} \left( 1 - \sum_{j \neq i} \mathbb{P}(\|\mathbf{x}_i - \mathbf{x}_j\| < (2C_1 n)^{-1/d}) \right) \\
&\geq \frac{\rho}{2} \left( 1 - \sum_{j \neq i} C_1 (2C_1 n)^{-d/d} \right) && \text{(By (25) )} \\
&\geq \frac{\rho}{4}.
\end{aligned}
$$

The sum

$$
\sum_{i=1}^{n} Z_i Y_i W_i
$$

is a function of the dataset $(\mathbf{X}, \mathbf{y})$. If we change one of the points $(\mathbf{x}_j, y_j)$ to a different value $(\mathbf{x}_j', y_j')$, then at most $C_2$ of the values $Z_i$ will change in value by Lemma D.9 (where $C_2$ is a constant depending only on $d$), and only one of the values $Y_i$ and $W_i$ (the $j$-th value) will change. So changing one of the points can alter the value of the sum by at most $C_2 + 1$. By McDiarmid's inequality, for all $t \geq 0$,

$$
\mathbb{P}\left( \sum_{i=1}^{n} Z_i Y_i W_i \leq \frac{\rho n}{4} - t \right) \leq \mathbb{P}\left( \sum_{i=1}^{n} Z_i Y_i W_i \leq \sum_{i=1}^{n} \mathbb{E}[Z_i Y_i W_i] - t \right)
$$

$$
\leq \exp\left( -\frac{2t^2}{n(C_2 + 1)^2} \right).
$$

Setting $t = \frac{\rho n}{8}$ with $n \geq \frac{32(C_2+1)^2}{\rho^2} \log \frac{1}{\epsilon}$, we obtain with probability at least $1 - \epsilon$ that

$$
|\mathcal{B}| = \sum_{i=1}^{n} Z_i Y_i \geq \frac{\rho n}{8}.
$$

So $\mathcal{B}$ also satisfies condition 1 with probability at least $1 - \epsilon$. $\qquad\square$

## D.4. Proofs for Sobolev inequalities

To prove Corollary 4.5, we first establish some useful Sobolev inequalities for balls in $\mathbb{R}^d$ (see Section 5.6.2 in Evans (2022)).

**Lemma D.11** (Morrey's inequality). *Let $\delta > 0$ and suppose that $u \in C^1(\mathbb{R}^d)$. Then for all $\mathbf{x}_0 \in \mathbb{R}^d$ and all $\mathbf{x}_1 \in B(\mathbf{x}_0, \delta)$,*

$$
|u(\mathbf{x}_1) - u(\mathbf{x}_0)| \lesssim_{d,p} \delta^{1-d/p} \left( \int_{B(\mathbf{x}_0, 2\delta)} \|Du(\mathbf{x})\|^p d\mathbf{x} \right)^{1/p}.
$$

**Corollary D.12.** *Let $\delta > 0$ and $d < kp$, and suppose that $u \in C^\infty(\mathbb{R}^d)$ with $\|u\|_{W^{k,p}(\mathbb{R}^d)} < \infty$. Then for all $j \in \{0, 1, \cdots, k-1\}$, $\mathbf{x}_0 \in \mathbb{R}^d$ and all $\mathbf{x}_1 \in B(\mathbf{x}_0, \delta)$,*

$$
\|D^j u(\mathbf{x}_1) - D^j u(\mathbf{x}_0)\| \lesssim_{k,d,p} \delta^{k-j-d/p} \left( \int_{B(\mathbf{x}_0, 2\delta)} \|D^k u(\mathbf{x})\|^p d\mathbf{x} \right)^{1/p}.
$$

*Proof of Corollary D.12.* We prove by reverse induction on $j$. The base case $j = k - 1$ follows from Lemma D.11. Now suppose that the statement holds for some $j \in \{1, \cdots, k - 1\}$. Then for some constant $C_{k,d,p} > 0$,

$$\|D^j u(\mathbf{x}_1) - D^j u(\mathbf{x}_0)\| \leq C_{k,d,p} \delta^{k-j-d/p} \left( \int_{B(\mathbf{x}_0, 2\delta)} \|D^k u(\mathbf{x})\|^p d\mathbf{x} \right)^{1/p}$$

for all $\mathbf{x}_0 \in \mathbb{R}^d$ and $\mathbf{x}_1 \in B(\mathbf{x}_0, \delta)$. Now suppose that $\mathbf{x}_0, \mathbf{x}_1 \in \mathbb{R}^d$ with $\|\mathbf{x}_0 - \mathbf{x}_1\| < \delta$. By the fundamental theorem of calculus,

$$\|D^{j-1}u(\mathbf{x}_1) - D^{j-1}u(\mathbf{x}_0)\|$$

$$= \left\| \left[ D^{j-1}u(\mathbf{x}_1) - D^{j-1}u\left(\frac{1}{2}(\mathbf{x}_0 + \mathbf{x}_1)\right) \right] - \left[ D^{j-1}u(\mathbf{x}_0) - D^{j-1}u\left(\frac{1}{2}(\mathbf{x}_0 + \mathbf{x}_1)\right) \right] \right\|$$

$$= \left\| \int_0^1 2 \frac{d}{dt} \left[ D^{j-1}u\left(\mathbf{x}_0 + \frac{t+1}{2}(\mathbf{x}_1 - \mathbf{x}_0)\right) - D^{j-1}u\left(\mathbf{x}_1 + \frac{t+1}{2}(\mathbf{x}_0 - \mathbf{x}_1)\right) \right] dt \right\|$$

$$\leq 2 \int_0^1 \left\| D^j u\left(\mathbf{x}_0 + \frac{t+1}{2}(\mathbf{x}_1 - \mathbf{x}_0)\right) - D^j u\left(\mathbf{x}_1 + \frac{t+1}{2}(\mathbf{x}_0 - \mathbf{x}_1)\right) \right\| \|\mathbf{x}_1 - \mathbf{x}_0\| dt$$

$$\leq 2\delta \sup_{t \in [0,1]} \left\| D^j u\left(\mathbf{x}_0 + \frac{t+1}{2}(\mathbf{x}_1 - \mathbf{x}_0)\right) - D^j u\left(\mathbf{x}_1 + \frac{t+1}{2}(\mathbf{x}_0 - \mathbf{x}_1)\right) \right\|$$

$$\leq C_{k,d,p} \delta^{1+k-j-d/p} \left( \int_{B(\mathbf{x}_0, 2\delta)} \|D^k u(\mathbf{x})\|^p d\mathbf{x} \right)^{1/p},$$

where in the last line we used the inductive hypothesis and that

$$\left\| \left(\mathbf{x}_0 + \frac{t+1}{2}(\mathbf{x}_1 - \mathbf{x}_0)\right) - \left(\mathbf{x}_1 + \frac{t+1}{2}\mathbf{x}_0 - \mathbf{x}_1\right) \right\| = \|t(\mathbf{x}_1 - \mathbf{x}_0)\| < \delta \text{ for } t \in [0, 1].$$

So we have shown that the result holds for $j - 1$, and therefore the result holds for all $j \in \{0, 1, \cdots, k-1\}$ by induction. $\qquad \square$

Corollary 4.5 then follows using some additional results from Sobolev space theory.

*Proof of Corollary 4.5.* Let $U = B(\mathbf{x}_0, 4\delta)$ and let $V = B(\mathbf{x}_0, 2\delta)$. By the Sobolev embedding theorem (Theorem A.1), pointwise evaluation is well-defined and continuous on $W^{k,p}(U)$. Now by Theorem 5.3.3 in (Evans, 2022), there exist functions $u_m \in C^\infty(\overline{U})$ converging to $u$ in $W^{k,p}(U)$. For each $m$, let $v_m \in C^\infty(\mathbb{R}^d)$ be a smooth function which is equal to $u_m$ on $V$. Since $u_m \to u$ in $W^{k,p}(U)$, we also have $u_m \to u$ in $W^{k,p}(V)$, and so $v_m \to u$ in $W^{k,p}(V)$. Then by Corollary D.12 we have

$$|u(x_1) - u(x_0)| = \lim_{m \to \infty} |v_m(x_1) - v_m(x_0)|$$

$$\lesssim_{k,d,p} \lim_{m \to \infty} \delta^{k-d/p} \left( \int_{B(\mathbf{x}_0, 2\delta)} \|D^k v_m(\mathbf{x})\|^p d\mathbf{x} \right)^{1/p}$$

$$= \delta^{k-d/p} \left( \int_{B(\mathbf{x}_0, 2\delta)} \|D^k u(\mathbf{x})\|^p d\mathbf{x} \right)^{1/p}$$

$$\leq \delta^{k-d/p} \|u\|_{W^{k,p}(\mathbb{B}(\mathbf{x}_0, 2\delta))},$$

where the second to last line follows as $v_m \to u$ in $W^{k,p}(V)$ implies that $D^k v_m \to D^k u$ in $L^p(V)$. The result follows by raising both sides to the $p$th power. $\qquad \square$

The above corollary shows that for an interpolating function $f^*$ and a corrupt point $i \in \mathcal{B}$, we can find a ball around $\mathbf{x}_i$ within which the function does not vary too much. In order for us to use this, a nontrivial fraction of this ball should be contained in $\Omega$. The following two theorems establish this.

Let $\Omega$ be a bounded open subset of $\mathbb{R}^d$. We say that $\Omega$ is a $W^{k,p}$-*extension domain* if there exists a bounded linear operator $E : W^{k,p}(\Omega) \to W^{k,p}(\mathbb{R}^d)$ such that for all $f \in W^{k,p}(\Omega)$, $Ef|_\Omega = f$.

**Theorem D.13** (Extension theorem). *If $\Omega$ is a bounded open subset of $\mathbb{R}^d$ with $C^1$ boundary, then $\Omega$ is a $W^{1,p}$-extension domain for all $p \in [1, \infty]$.*

For a proof of the above theorem, see Evans (2022, Section 5.4).

**Theorem D.14** (Koskela 1990, Theorem 6.5). *Let $\Omega \subset \mathbb{R}^d$ be a $W^{1,p}$-extension domain for some $p > d - 1$. Then*

$$|\Omega \cap B(\mathbf{x}_0, \delta)| \gtrsim_{d,\Omega} |B(\mathbf{x}_0, \delta)|$$

*for all $\mathbf{x}_0 \in \overline{\Omega}$ and $\delta \in [0, \text{diam}(\Omega)]$.*

## E. Benign and harmful overfitting when $k > 2d/p$

In this section, we prove Theorems 6.2 and 6.3. We start with a few general lemmas about the regularity of the loss function and data distribution.

**Lemma E.1.** *For every $M, \epsilon > 0$, there exists $R > 0$ such that*

$$\sup_{\mathbf{x} \in \overline{\Omega}} \sup_{|\hat{y}| \leq M} \int_{\mathbb{R}} \ell(\hat{y}, y) \mathbf{1}_{|y|>R} d\nu_{\mathbf{x}}(y) < \epsilon.$$

*Proof.* By Assumption 3.1, for $|\hat{y}| \leq M$,

$$\ell(\hat{y}, y) \leq C_\ell \exp(\tau_\ell(1 + M)(1 + |y|)) \leq C_1 \exp(C_2 |y|)$$

for some $C_1, C_2 > 0$ depending on $\ell$ and $M$. Then

$$\sup_{\mathbf{x} \in \overline{\Omega}} \sup_{|\hat{y}| \leq M} \int_{\mathbb{R}} \ell(\hat{y}, y) \mathbf{1}_{|y|>R} d\nu_{\mathbf{x}}(y) \leq \sup_{\mathbf{x} \in \overline{\Omega}} \int_{\mathbb{R}} C_1 \exp(C_2 |y|) \mathbf{1}_{|y|>R} d\nu_{\mathbf{x}}(y)$$

$$\leq \sup_{\mathbf{x} \in \overline{\Omega}} \int_{\mathbb{R}} C_1 \exp(C_2 |y|) \exp(|y| - R) \mathbf{1}_{|y|>R} d\nu_{\mathbf{x}}(y)$$

$$\leq e^{-R} \sup_{\mathbf{x} \in \overline{\Omega}} \int_{\mathbb{R}} C_1 \exp((C_2 + 1)|y|) d\nu_{\mathbf{x}}(y).$$

By Lemma D.1, the supremum in last term is finite, so we can choose $R$ sufficiently large such that the above expression is at most $\epsilon$. $\square$

**Lemma E.2.** *There exist constants $C_1 > 0$, $C_2 \in (0, 1]$ depending on $\ell$ and $\mu$ such that for all $\mathbf{x} \in \overline{\Omega}$ and $y, y' \sim \nu_{\mathbf{x}}$ drawn independently,*

$$\mathbb{P}(\ell(y, y') \geq C_1) \geq C_2.$$

*Proof.* Let $\rho$ and $\sigma$ be the constants in Assumption 3.5. By Assumption 3.4,

$$\mathbb{P}\left(|y| \geq C_y \sqrt{\log \frac{4}{\rho}}\right) \leq \frac{\rho}{2}.$$

Let $K_1 = C_y \sqrt{\log \frac{4}{\rho}}$, so that $\mathbb{P}(|y| \leq K_1) \geq 1 - \rho/2$. Let $C_\ell$ and $\tau_\ell$ be the constants from Assumption 3.1. By Lemma E.1, there exists $K_2 > 0$ depending on $\mu$ and $\ell$ such that for all $y \in \mathbb{R}$ with $|y| \leq K_1$,

$$\mathbb{E}[\ell(y, y') \mathbf{1}_{|y'|>K_2} \mid y] = \int_{\mathbb{R}} \ell(y, y') \mathbf{1}_{|y'|>K_2} d\nu_{\mathbf{x}}(y') \leq \frac{\sigma}{2}.$$

By Assumption 3.5,

$$\mathbb{P}(\mathbb{E}[\ell(y, y') \mid y] \geq \sigma) = \mathbb{P}(\mathcal{L}(y; \mathbf{x}) \geq \sigma) \geq \rho.$$

Let $M = 1 + \sup_{|y| \leq K_1} \sup_{|y'| \leq K_2} \ell(y, y')$, which is finite since $\ell$ is continuous. For $|y| \leq K_1$,

$$\mathbb{E}[\ell(y, y')1_{|y'| \leq K_2} \mid y] \leq M\mathbb{P}\left(\ell(y, y')1_{|y'| \leq K_2} \geq \frac{\sigma}{4}\Big|y\right) + \frac{\sigma}{4},$$

so if $|y| \leq K_1$ and $\mathbb{E}[\ell(y, y') \mid y] \geq \sigma$, then

$$
\begin{aligned}
\mathbb{P}\left(\ell(y, y') \geq \frac{\sigma}{4}\Big|y\right) &\geq \mathbb{P}\left(\ell(y, y')1_{|y'| \leq K_2} \geq \frac{\sigma}{4}\Big|y\right) \\
&\geq \frac{1}{M}\mathbb{E}[\ell(y, y')1_{|y'| \leq K_2} \mid y] - \frac{\sigma}{4M} \\
&= \frac{1}{M}\mathbb{E}[\ell(y, y') \mid y] - \frac{1}{M}\mathbb{E}[\ell(y, y')\mathbf{1}_{|y'| > K_2} \mid y] - \frac{\sigma}{4M} \\
&\geq \frac{\sigma}{M} - \frac{\sigma}{2M} - \frac{\sigma}{4M} \\
&= \frac{\sigma}{4M}.
\end{aligned}
$$

Therefore,

$$
\begin{aligned}
\mathbb{P}\left(\ell(y, y') \geq \frac{\sigma}{4}\right) &\geq \mathbb{P}\left(|y| \leq K_1, \mathbb{E}[\ell(y, y') \mid y] \geq \sigma\right)\mathbb{P}\left(\ell(y, y') \geq \frac{\sigma}{4}\Big||y| \leq K_1, \mathbb{E}[\ell(y, y') \mid y] \geq \sigma\right) \\
&\geq \mathbb{P}\left(|y| \leq K_1, \mathbb{E}[\ell(y, y') \mid y] \geq \sigma\right)\frac{\sigma}{4M} \\
&\geq \frac{\sigma}{4M}\left(1 - \mathbb{P}(|y| > K_1) - \mathbb{P}(\mathbb{E}[\ell(y, y') \mid y] < \sigma)\right) \\
&\geq \frac{\sigma}{4M}\left(1 - \frac{\rho}{2} - (1 - \rho)\right) \\
&= \frac{\sigma\rho}{8M}.
\end{aligned}
$$

$\square$

**Lemma E.3.** *There exist constants $C_1, C_2 > 0$, $C_3 \in (0, 1]$ depending on $\ell$ and $\mu$ such that for all $\mathbf{x}, \mathbf{x}' \in \overline{\Omega}$ with $\|\mathbf{x} - \mathbf{x}'\| \leq C_1$ and $y \sim \nu_{\mathbf{x}}, y' \sim \nu_{\mathbf{x}'}$ drawn independently,*

$$\mathbb{P}(|y - y'| \geq C_2) \geq C_3.$$

*Proof.* First, we consider the case $\mathbf{x} = \mathbf{x}'$. By Lemma E.2, there exist constants $C_4 > 0$, $C_5 \in (0, 1]$ depending on $\ell$ and $\mu$ such that for all $\mathbf{x} \in \overline{\Omega}$ and $y, y' \sim \nu_{\mathbf{x}}$ drawn independently,

$$\mathbb{P}(\ell(y, y') \geq C_4) \geq C_5.$$

By Assumption 3.4, there exists a constant $C_6 > 0$ depending on $\mu$ such that

$$\mathbb{P}\left(\max(|y|, |y'|) \geq C_6\right) \leq \frac{C_5}{2}.$$

Let $\omega$ be the event that $\max(|y|, |y'|) \leq C_6$ and $\ell(y, y') \geq C_4$, so that $\mathbb{P}(\omega) \geq C_5/2$. Since the set $[-C_6, C_6]^2$ is compact and $\ell$ is continuous and equal to zero on the diagonal, there exists $\delta > 0$ depending on $\ell$ and $C_6$ such that for all $y, y' \in [-C_6, C_6]$ with $|y - y'| < \delta$, $\ell(y, y') < C_4$. Therefore, if $\omega$ occurs, then $|y - y'| \geq \delta$. This establishes the result for the case $\mathbf{x} = \mathbf{x}'$.

Now we consider the general case. Let $\psi : [0, \infty) \to [0, 1]$ be a function which takes value 0 on $[0, \delta/2]$, value 1 on $[\delta, \infty)$, and is linear on $[\delta/2, \delta]$. Let us define $\varphi : \overline{\Omega} \times \overline{\Omega} \to \mathbb{R}$ by

$$\varphi(\mathbf{x}, \mathbf{x}') = \int_{-\infty}^{\infty} \int_{-\infty}^{\infty} \psi(|y - y'|)d\nu_{\mathbf{x}}(y)d\nu_{\mathbf{x}'}(y').$$

We claim that $\varphi$ is continuous. To see this, let $(\mathbf{x}_n, \mathbf{x}'_n)$ be a sequence in $\overline{\Omega} \times \overline{\Omega}$ converging to some $(\mathbf{x}, \mathbf{x}')$. By Assumption 3.2, $\nu$ is weakly continuous, so $\nu_{\mathbf{x}_n} \to \nu_{\mathbf{x}}$ and $\nu_{\mathbf{x}'_n} \to \nu_{\mathbf{x}'}$ weakly. This implies that $\nu_{\mathbf{x}_n} \otimes \nu_{\mathbf{x}'_n} \to \nu_{\mathbf{x}} \otimes \nu_{\mathbf{x}'}$

weakly (see Bogachev (2018, Proposition 2.7.7)). Since $(y, y') \mapsto \psi(|y - y'|)$ is bounded and continuous, it follows that $\varphi(\mathbf{x}_n, \mathbf{x}'_n) \to \varphi(\mathbf{x}, \mathbf{x}')$, establishing the continuity of $\varphi$. Next, observe that for $y, y' \sim \nu_{\mathbf{x}}$ drawn independently,

$$
\begin{aligned}
\varphi(\mathbf{x}, \mathbf{x}) &= \int_{-\infty}^{\infty} \int_{-\infty}^{\infty} \psi(|y - y'|) d\nu_{\mathbf{x}}(y) d\nu_{\mathbf{x}}(y') \\
&\geq \int_{-\infty}^{\infty} \int_{-\infty}^{\infty} \mathbf{1}_{|y-y'| \geq \delta} d\nu_{\mathbf{x}}(y) d\nu_{\mathbf{x}}(y') \\
&= \mathbb{P}(|y - y'| \geq \delta) \\
&\geq \frac{C_5}{2}.
\end{aligned}
$$

Since $\varphi$ is continuous, there exists $C_1 > 0$ depending on $\ell$ and $\mu$ such that for all $\mathbf{x}, \mathbf{x}' \in \overline{\Omega}$ with $\|\mathbf{x} - \mathbf{x}'\| \leq C_1$, $\varphi(\mathbf{x}, \mathbf{x}') \geq \frac{C_5}{4}$. For $y \sim \nu_{\mathbf{x}}$ and $y' \sim \nu_{\mathbf{x}'}$ drawn independently,

$$
\begin{aligned}
\mathbb{P}(|y - y'| \geq \delta/2) &= \int_{-\infty}^{\infty} \int_{-\infty}^{\infty} \mathbf{1}_{|y-y'| \geq \delta/2} d\nu_{\mathbf{x}}(y) d\nu_{\mathbf{x}'}(y') \\
&\geq \int_{-\infty}^{\infty} \int_{-\infty}^{\infty} \psi(|y - y'|) d\nu_{\mathbf{x}}(y) d\nu_{\mathbf{x}'}(y') \\
&= \varphi(\mathbf{x}, \mathbf{x}') \\
&\geq \frac{C_5}{4}.
\end{aligned}
$$

So the result holds with our chosen $C_1$, with $C_2 = \frac{\delta}{2}$, and $C_3 = \frac{C_5}{4}$.

$\square$

In contrast to the $k < 1.5d/p$ case where the representational cost is primarily determined by the average nearest-neighbor distance $\delta_i$, the representational cost in the $k > 2d/p$ regime is determined by the points which are closest together. To this end, we bound their distance along with their associated label differences.

**Lemma E.4.** *Let* $\epsilon \in (0, 1)$*, and let* $n \gtrsim_{d,\Omega,\mu,\ell} \epsilon^{-1}$*. Then with probability at least* $1 - \epsilon$*, there exist* $i, j \in [n]$ *with* $i \neq j$ *such that* $\|\mathbf{x}_i - \mathbf{x}_j\| \lesssim_{d,\Omega,\mu,\ell} \epsilon^{-1/d} n^{-2/d}$ *and* $|y_i - y_j| \gtrsim_{d,\Omega,\mu,\ell} 1$*.*

*Proof.* By Lemma E.3, there exist constants $C_1, C_2 > 0$, $C_3 \in (0, 1]$ depending on $\ell$ and $\mu$ such that for all $\mathbf{x}, \mathbf{x}' \in \overline{\Omega}$ with $\|\mathbf{x} - \mathbf{x}'\| \leq C_1$ and $y \sim \nu_{\mathbf{x}}, y' \sim \nu_{\mathbf{x}'}$ drawn independently,

$$
\mathbb{P}(|y - y'| \geq C_2) \geq C_3.
$$

Let $t \in (0, C_1)$. For $\{i, j\} \in \binom{[n]}{2}$, let $Z_{i,j}$ be the Bernoulli random variable which takes the value 1 if $\|\mathbf{x}_i - \mathbf{x}_j\| < t$ and $|y_i - y_j| \geq C_2$. By Theorem D.14 and Assumption 3.3, for all $\mathbf{x} \in \overline{\Omega}$,

$$
\mu_x(B(\mathbf{x}, t) \cap \overline{\Omega}) \gtrsim_{d,\Omega} |B(\mathbf{x}, t) \cap \overline{\Omega}| \gtrsim_{d,\Omega} |B(\mathbf{x}, t)| \gtrsim_d t^d,
$$

so

$$
\begin{aligned}
\mathbb{E}[Z_{i,j}] &= \mathbb{P}(\|\mathbf{x}_i - \mathbf{x}_j\| < t) \mathbb{P}(|y_i - y_j| \geq C_2 \mid \|\mathbf{x}_i - \mathbf{x}_j\| < t) \\
&\geq \mathbb{P}(\|\mathbf{x}_i - \mathbf{x}_j\| < t) C_3 \\
&\geq C_4 t^d,
\end{aligned}
$$

and

$$
\begin{aligned}
\mathbb{E}[Z_{i,j}] &= \mathbb{P}(\|\mathbf{x}_i - \mathbf{x}_j\| < t) \mathbb{P}(|y_i - y_j| \geq C_2 \mid \|\mathbf{x}_i - \mathbf{x}_j\| < t) \\
&\leq \mathbb{P}(\|\mathbf{x}_i - \mathbf{x}_j\| < t) \\
&\leq C_5 t^d, \quad\quad\quad\quad\quad\quad\quad\quad\quad\quad\quad\quad\quad\quad\quad\quad \text{(by Assumption 3.3)}
\end{aligned}
$$

where $C_4, C_5 > 0$ are constants depending on $d$, $\Omega$, $\mu$, and $\ell$.

Suppose that $\{i,j\}, \{i',j'\} \in \binom{[n]}{2}$. If $|\{i,j\} \cap \{i',j'\}| = 0$, then $Z_{i,j}$ and $Z_{i',j'}$ are independent and so $\mathrm{Cov}(Z_{i,j}, Z_{i',j'}) = 0$. If $|\{i,j\} \cap \{i',j'\}| = 1$, then without loss of generality we can assume that $i = i'$ and $j \neq j'$. Then

$$
\begin{aligned}
\mathrm{Cov}(Z_{i,j}, Z_{i',j'}) &\leq \mathbb{E}[Z_{i,j} Z_{i',j'}] \\
&= \mathbb{P}(\|\mathbf{x}_i - \mathbf{x}_j\| < t, |y_i - y_j| \geq C_2, \|\mathbf{x}_i - \mathbf{x}_{j'}\| < t, |y_i - y_{j'}| \geq C_2) \\
&\leq \mathbb{P}(\|\mathbf{x}_i - \mathbf{x}_j\| < t, \|\mathbf{x}_i - \mathbf{x}_{j'}\| < t) \\
&= \mathbb{E}[\mathbb{P}(\|\mathbf{x}_i - \mathbf{x}_j\| < t, \|\mathbf{x}_i - \mathbf{x}_{j'}\| < t \mid \mathbf{x}_i)] \\
&= \mathbb{E}[\mathbb{P}(\|\mathbf{x}_i - \mathbf{x}_j\| < t \mid \mathbf{x}_i)\mathbb{P}(\|\mathbf{x}_i - \mathbf{x}_{j'}\| < t \mid \mathbf{x}_i)] \\
&\leq C_6 t^{2d}, &\text{(by Assumption 3.3)}
\end{aligned}
$$

where $C_6 > 0$ is a constant depending on $d$, $\Omega$, and $\mu$. If $|\{i,j\} \cap \{i',j'\}| = 2$, then $\{i,j\} = \{i',j'\}$ and

$$
\mathrm{Cov}(Z_{i,j}, Z_{i',j'}) = \mathrm{Var}(Z_{i,j}) \leq \mathbb{E}[Z_{i,j}^2] = \mathbb{E}[Z_{i,j}] \leq C_5 t^d.
$$

Combining the second-moment terms, we get

$$
\begin{aligned}
\mathrm{Var}\left(\sum_{\mathcal{S} \in \binom{[n]}{2}} Z_{\mathcal{S}}\right) &= \sum_{\mathcal{S}_1, \mathcal{S}_2 \in \binom{[n]}{2}} \mathrm{Cov}(Z_{\mathcal{S}_1}, Z_{\mathcal{S}_2}) \\
&= \sum_{\substack{\mathcal{S}_1, \mathcal{S}_2 \in \binom{[n]}{2} \\ |\mathcal{S}_1 \cap \mathcal{S}_2| = 0}} \mathrm{Cov}(Z_{\mathcal{S}_1}, Z_{\mathcal{S}_2}) + \sum_{\substack{\mathcal{S}_1, \mathcal{S}_2 \in \binom{[n]}{2} \\ |\mathcal{S}_1 \cap \mathcal{S}_2| = 1}} \mathrm{Cov}(Z_{\mathcal{S}_1}, Z_{\mathcal{S}_2}) + \sum_{\substack{\mathcal{S}_1, \mathcal{S}_2 \in \binom{[n]}{2} \\ |\mathcal{S}_1 \cap \mathcal{S}_2| = 2}} \mathrm{Cov}(Z_{\mathcal{S}_1}, Z_{\mathcal{S}_2}) \\
&\leq 0 + n(n-1)(n-2)C_6 t^{2d} + \frac{n(n-1)}{2} C_5 t^d \\
&\leq C_7 (n^3 t^{2d} + n^2 t^d),
\end{aligned}
$$

where $C_7 > 0$ is a constant depending on $d$, $\Omega$, $\mu$, and $\ell$.

Let $s = \frac{C_4 n(n-1)}{4} t^d$. Let $Z = \sum_{\mathcal{S} \in \binom{[n]}{2}} Z_{\mathcal{S}}$. Then by Chebyshev's inequality,

$$
\begin{aligned}
\mathbb{P}(|Z - \mathbb{E}[Z]| > s) &\leq \frac{\mathrm{Var}(Z)}{s^2} \\
&\leq C_8 (n^{-1} + t^{-d} n^{-2}),
\end{aligned}
$$

where $C_8 > 0$ is a constant depending on $d$, $\Omega$, $\mu$, and $\ell$. Let us define

$$
A := \max\left(1, \left(\frac{8}{C_4}\right)^{1/d}, \left(\frac{2C_8}{\epsilon}\right)^{1/d}\right).
$$

Setting

$$
n \geq \max\left(2, \frac{2C_8}{\epsilon}, \left(\frac{A}{C_1}\right)^{d/2}\right)
$$

and $t = An^{-2/d}$, so that

$$
t = An^{-2/d} \leq A\left(\frac{A}{C_1}\right)^{-1} = C_1,
$$

we have

$$\mathbb{P}(|Z - \mathbb{E}[Z]| > s) \leq C_8 n^{-1} + C_8 t^{-d} n^{-2}$$

$$\leq C_8 \left(\frac{2C_8}{\epsilon}\right)^{-1} + C_8 \left(\left(\frac{2C_8}{\epsilon}\right)^{1/d} n^{-2/d}\right)^{-d} n^{-2}$$

$$= \frac{\epsilon}{2} + \frac{\epsilon}{2}$$

$$= \epsilon.$$

In this case,

$$Z \geq \mathbb{E}[Z] - s$$

$$\geq \frac{n(n-1)}{2} C_4 t^d - \frac{C_4 n(n-1)}{4} t^d$$

$$= \frac{C_4 n(n-1)}{4} A^d n^{-2}$$

$$\geq \frac{C_4 n^2}{8} \left(\frac{8}{C_4}\right) n^{-2}$$

$$= 1.$$

So with probability at least $1 - \epsilon$, $Z \geq 1$, which means that there exist $i, j \in [n]$ with $i \neq j$ such that $\|\mathbf{x}_i - \mathbf{x}_j\| < t \lesssim_{d,\Omega,\mu,\ell} \epsilon^{-1/d} n^{-2/d}$ and $|y_i - y_j| \geq C_2 \gtrsim_{d,\Omega,\mu,\ell} 1$, as desired. $\qquad \square$

The minimum norm interpolator can be sharply bounded using the minimum distance points.

**Lemma E.5.** *Let $\epsilon \in (0,1)$, let $n \gtrsim_{d,\Omega,\mu,\ell} \epsilon^{-1}$. Then with probability at least $1 - \epsilon$, all min-norm interpolators $f^* \in W^{k,p}(\mathbb{R}^d)$ satisfy*

$$\|f^*\|_{W^{k,p}(\mathbb{R}^d)}^p \gtrsim_{k,d,p,\Omega,\mu,\ell} \epsilon^{kp/d-1} n^{2(kp/d-1)}.$$

*Proof.* Let $f^* \in W^{k,p}(\mathbb{R}^d)$ be a min-norm interpolator. By Lemma E.4, there exist constants $C_1, C_2 > 0$ depending on $d$, $\Omega$, $\mu$, and $\ell$ such that with probability at least $1 - \epsilon$, there exist $i, j \in [n]$ with $i \neq j$ such that $\|\mathbf{x}_i - \mathbf{x}_j\| \leq C_1 \epsilon^{-1/d} n^{-2/d}$ and $|y_i - y_j| \geq C_2$. The ball $B(\mathbf{x}_i, 2C_1 \epsilon^{-1/d} n^{-2/d})$ contains $\mathbf{x}_j$, so by Corollary 4.5,

$$|f^*(\mathbf{x}_i) - f^*(\mathbf{x}_j)|^p \lesssim_{k,d,p} (2C_1 \epsilon^{-1/d} n^{-2/d})^{kp-d} \|f^*\|_{W^{k,p}(\mathbb{R}^d)}^p$$

$$\lesssim_{k,d,p,\Omega,\mu,\ell} \epsilon^{-kp/d+1} n^{-2kp/d+2} \|f^*\|_{W^{k,p}(\mathbb{R}^d)}^p.$$

Rearranging and using that $f^*$ interpolates the data, we get

$$\|f^*\|_{W^{k,p}(\mathbb{R}^d)}^p \gtrsim_{k,d,p,\Omega,\mu,\ell} \epsilon^{kp/d-1} n^{2(kp/d-1)} |y_i - y_j|^p$$

$$\gtrsim_{k,d,p,\Omega,\mu,\ell} \epsilon^{kp/d-1} n^{2(kp/d-1)}.$$

$$\square$$

The following lemma bounds an expression which appears in the asymptotic representational cost for our benign and harmful interpolator constructions.

**Lemma E.6.** *Let $\beta = kp - d$, let $\alpha \in (1/\beta, 1/d)$, let $\epsilon \in (0,1)$, and assume $\beta > d$. Then with probability at least $1 - \epsilon$,*

$$\sum_{i=1}^{n} (1 + |y_i|)^p \min\left(n^{-2/d+\alpha}, \frac{\delta_i}{2}\right)^{-\beta} \lesssim_{k,p,d,\Omega,\mu,\ell,\alpha} \epsilon^{-\beta/d} n^{2\beta/d}.$$

*Proof.* For $i \in [n]$, let us define

$$a_i := (1 + |y_i|)^p$$

and

$$w_i := \min\left(n^{-2/d+\alpha}, \frac{\delta_i}{2}\right)^{-\beta}.$$

By Assumption 3.3, for $i \in [n]$ and $t > 0$, we have

$$\mathbb{P}(\delta_i \leq t) \leq \sum_{j \neq i} \mathbb{P}(\|\mathbf{x}_i - \mathbf{x}_j\| \leq t) \leq C_1 n t^d,$$

where $C_1 > 0$ is a constant depending on $d$ and $\mu$, and in particular

$$\mathbb{P}\left(\min_{i \in [n]} \delta_i \leq \left(\frac{\epsilon}{2C_1}\right)^{1/d} n^{-2/d}\right) \leq \sum_{i=1}^{n} \mathbb{P}\left(\delta_i \leq \left(\frac{\epsilon}{2C_1}\right)^{1/d} n^{-2/d}\right) \leq \frac{\epsilon}{2}.$$

Let $\omega$ denote the event that $\min_{i \in [n]} \delta_i/2 > h$, where $h = \frac{1}{2}\left(\frac{\epsilon}{2C_1}\right)^{1/d} n^{-2/d}$. We have

$$\mathbb{E}[w_i \mathbf{1}_\omega] = \int_0^\infty \mathbb{P}\left(\min\left(n^{-2/d+\alpha}, \frac{\delta_i}{2}\right)^{-\beta} \mathbf{1}_\omega > t\right) dt$$

$$= n^{2\beta/d-\alpha\beta} + \int_{n^{2\beta/d-\alpha\beta}}^\infty \mathbb{P}\left(\frac{\delta_i}{2} < t^{-1/\beta}, \omega\right) dt$$

$$\leq n^{2\beta/d-\alpha\beta} + \int_{n^{2\beta/d-\alpha\beta}}^\infty \mathbb{P}\left(h < \frac{\delta_i}{2} < t^{-1/\beta}\right) dt$$

$$\leq n^{2\beta/d-\alpha\beta} + \int_{n^{2\beta/d-\alpha\beta}}^{h^{-\beta}} \mathbb{P}\left(\frac{\delta_i}{2} < t^{-1/\beta}\right) dt$$

$$\leq n^{2\beta/d-\alpha\beta} + \int_0^{h^{-\beta}} 2^d C_1 n t^{-d/\beta} dt$$

$$= n^{2\beta/d-\alpha\beta} + \frac{2^d C_1 n}{1 - d/\beta} h^{d-\beta}$$

$$\lesssim_{d,\Omega,\mu,\ell,\alpha} n^{2\beta/d-\alpha\beta} + \epsilon^{1-\beta/d} n^{2\beta/d-1}$$

$$\lesssim_{d,\Omega,\mu,\ell,\alpha} \epsilon^{1-\beta/d} n^{2\beta/d-1}, \qquad\qquad (\text{since } \alpha\beta > 1)$$

so we can write

$$\mathbb{E}[w_i \mathbf{1}_\omega] \leq C_2 \epsilon^{1-\beta/d} n^{2\beta/d-1},$$

where $C_2 > 0$ is a constant depending on $d$, $\Omega$, $\mu$, $\ell$, and $\alpha$. Since the $y_i$ are conditionally sub-Gaussian (Assumption 3.4), there exists a constant $C_3 > 0$ depending on $\mu$ such that $\mathbb{E}[a_i \mid \mathbf{X}] \leq C_3$. Let $Z = \sum_{i=1}^{n} a_i w_i \mathbf{1}_\omega$. Then

$$\mathbb{E}[Z] = \sum_{i=1}^{n} \mathbb{E}[a_i w_i \mathbf{1}_\omega]$$

$$= \sum_{i=1}^{n} \mathbb{E}[w_i \mathbf{1}_\omega \mathbb{E}[a_i \mid \mathbf{X}]]$$

$$\leq C_3 \sum_{i=1}^{n} \mathbb{E}[w_i \mathbf{1}_\omega]$$

$$\leq C_2 C_3 \epsilon^{1-\beta/d} n^{2\beta/d},$$

By Markov's inequality,

$$\mathbb{P}(Z > 2C_2 C_3 \epsilon^{-\beta/d} n^{2\beta/d}) \leq \frac{\mathbb{E}[Z]}{2C_2 C_3 \epsilon^{-\beta/d} n^{2\beta/d}}$$

$$\leq \frac{\epsilon}{2}.$$

Let us denote the event that $Z \leq 2C_2 C_3 \epsilon^{-\beta/d} n^{2\beta/d}$ by $\omega'$. We have $\mathbb{P}(\omega') \geq 1 - \epsilon/2$. If $\omega$ and $\omega'$ both occur, then

$$\sum_{i=1}^{n} a_i w_i = Z \leq 2C_2 C_3 \epsilon^{-\beta/d} n^{2\beta/d}.$$

$\square$

*Proof of Theorem 6.2.* In the regime $kp > 2d$, the representational cost of the interpolator will be dominated by the two closest points. We take advantage of this property by constructing a function which interpolates with narrow bump functions around every point. Let $\beta = kp - d$, so that $\beta > d$. Let $\alpha' = \frac{1}{2}\left(\frac{1}{\beta} + \alpha\right)$, so that $\alpha' \in \left(\frac{1}{\beta}, \frac{1}{d}\right)$. For $i \in [n]$, let $r_i = \min\left(n^{-2/d + \alpha'}, \frac{\delta_i}{2}\right)$. By Lemma 4.1, there exist functions $\psi_i \in W^{k,p}(\mathbb{R}^d)$ satisfying the following properties:

1. For all $\mathbf{x} \in \mathbb{R}^d$, $0 \leq \psi_i(\mathbf{x}) \leq 1$.

2. For all $\mathbf{x} \in B(\mathbf{x}_i, r_i/2)$, $\psi_i(\mathbf{x}) = 1$.

3. For all $\mathbf{x} \notin B(\mathbf{x}_i, r_i)$, $\psi_i(\mathbf{x}) = 0$.

4. $\|\psi_i\|_{W^{k,p}(\mathbb{R}^d)}^p \lesssim_{k,p,d} 1 + r_i^{-\beta}$.

Then the $\psi_i$ have disjoint supports, since $B(\mathbf{x}_i, r_i) \subset B(\mathbf{x}_i, \delta_i/2)$ and the sets $B(\mathbf{x}_i, \delta_i/2)$ are disjoint by Lemma 4.2. Let $f_{\text{good}} = f_{\text{bayes}} + \sum_{i=1}^{n}(y_i - f_{\text{bayes}}(\mathbf{x}_i))\psi_i$. Then $f_{\text{good}}$ interpolates the data, and

$$\|f_{\text{good}}\|_{W^{k,p}(\mathbb{R}^d)}^p \lesssim_{\mu,p} 1 + \sum_{i=1}^{n}(1 + |y_i|)^p \|\psi_i\|_{W^{k,p}(\mathbb{R}^d)}^p$$

$$\lesssim_{k,p,d,\mu} 1 + \sum_{i=1}^{n}(1 + |y_i|)^p r_i^{-\beta}.$$

By Lemma E.6, with probability at least $1 - \epsilon/3$,

$$\sum_{i=1}^{n}(1 + |y_i|)^p r_i^{-\beta} \lesssim_{k,p,d,\Omega,\mu,\ell,\alpha} \epsilon^{-\beta/d} n^{2\beta/d},$$

so in this case,

$$\|f_{\text{good}}\|_{W^{k,p}(\mathbb{R}^d)}^p \leq C_1 \epsilon^{-\beta/d} n^{2\beta/d},$$

where $C_1 > 0$ is a constant depending on $k$, $p$, $d$, $\Omega$, $\mu$, $\ell$, and $\alpha$. Let us denote the event that this occurs by $\omega_1$. By Lemma E.5, with probability at least $1 - \epsilon/3$, all min-norm interpolators $f^* \in W^{k,p}(\mathbb{R}^d)$ satisfy

$$\|f^*\|_{W^{k,p}(\mathbb{R}^d)}^p \geq C_2 \epsilon^{\beta/d} n^{2\beta/d},$$

where $C_2 > 0$ is a constant depending on $k$, $d$, $p$, $\Omega$, $\mu$, and $\ell$. Let us denote the event that this occurs by $\omega_2$. If $\omega_1$ and $\omega_2$ both occur, then $f_{\text{good}}$ is a $\gamma$-ANM interpolator for all $\gamma \geq (C_2^{-1} C_1 \epsilon^{-2\beta/d})^{1/p}$. It remains to show that $f_{\text{good}}$ is benign.

By Assumption 3.4, there exists a constant $C_3 > 0$ depending on $\mu$ such that for all $t > 0$ and $i \in [n]$,

$$\mathbb{P}(|y_i| > t) \leq 2\exp\left(-\frac{t^2}{C_3^2}\right).$$

By a union bound, with probability at least $1 - \epsilon/3$, we have

$$|y_i| \leq C_3\sqrt{\log(6n\epsilon^{-1})}$$

for all $i \in [n]$. Let us denote the event that this occurs by $\omega_3$. Let $B = \cup_{i=1}^n B(\mathbf{x}_i, r_i)$, so that $f_{\text{good}}(\mathbf{x}) = f_{\text{bayes}}(\mathbf{x})$ for all $\mathbf{x} \notin B$, and $|f_{\text{good}}(\mathbf{x})| \leq C_3 \sqrt{\log(6n\epsilon^{-1})} + 2\|f_{\text{bayes}}\|_{L^\infty(\mathbb{R}^d)}$ for all $\mathbf{x} \in B$.

If $\omega_3$ occurs, then conditional on $(\mathbf{X}, \mathbf{y})$, we have

$$\int_{\overline{\Omega} \times \mathbb{R}} \ell(f_{\text{good}}(\mathbf{x}), y) d\mu(\mathbf{x}, y) = \int_{(\overline{\Omega} \backslash B) \times \mathbb{R}} \ell(f_{\text{bayes}}(\mathbf{x}), y) d\mu(\mathbf{x}, y) + \int_{B \times \mathbb{R}} \ell(f_{\text{good}}(\mathbf{x}), y) d\mu(\mathbf{x}, y)$$

$$\leq \int_{\overline{\Omega} \times \mathbb{R}} \ell(f_{\text{bayes}}(\mathbf{x}), y) d\mu(\mathbf{x}, y) + \int_{B \times \mathbb{R}} \ell(f_{\text{good}}(\mathbf{x}), y) d\mu(\mathbf{x}, y).$$

We bound the second term. By Assumption 6.1,

$$\int_{B \times \mathbb{R}} \ell(f_{\text{good}}(\mathbf{x}), y) d\mu(\mathbf{x}, y) \lesssim_\ell \int_{B \times \mathbb{R}} \kappa_\ell (1 + |f_{\text{good}}(\mathbf{x})|^q)(1 + |y|^q) d\mu(\mathbf{x}, y).$$

Using the bound for $|f_{\text{good}}(\mathbf{x})|$ on $B$ and that $n \gtrsim_{d,\Omega,\mu,\ell} \epsilon^{-1}$, we can write

$$\kappa_\ell (1 + |f_{\text{good}}(\mathbf{x})|^q) \leq C_4 (\log n)^{q/2},$$

where $C_4 > 0$ is a constant depending on $d, \Omega, \mu, \ell$. Then

$$\int_{B \times \mathbb{R}} \kappa_\ell (1 + |f_{\text{good}}(\mathbf{x})|^q)(1 + |y|^q) d\mu(\mathbf{x}, y) \leq \int_{B \times \mathbb{R}} C_4 (\log n)^{q/2} (1 + |y|^q) d\mu(\mathbf{x}, y).$$

By Assumption 3.4, we have $\sup_{\mathbf{x} \in \overline{\Omega}} \mathbb{E}[|y|^q \mid \mathbf{x}] \lesssim_{\mu,\ell} 1$, and by Assumption 3.3, we have $\mu_x(B) \lesssim_{\mu,d} |B|$. Thus, the above expression is bounded above by

$$C_5 |B| (\log n)^{q/2},$$

where $C_5 > 0$ is a constant depending on $d, \Omega, \mu$, and $\ell$. By the definition of $B$ and that the $B(\mathbf{x}_i, r_i)$ are disjoint, we have

$$|B| = \sum_{i=1}^n |B(\mathbf{x}_i, r_i)| \lesssim_d \sum_{i=1}^n r_i^d \leq n(n^{-2/d+\alpha'})^d = n^{-1+\alpha'd}.$$

Thus, for a test point $(\mathbf{x}, y)$ drawn independently from $\mu$, we have

$$\mathbb{E}[\ell(f_{\text{good}}(\mathbf{x}), y) \mid \mathbf{X}, \mathbf{y}] - \mathbb{E}[\ell(f_{\text{bayes}}(\mathbf{x}), y) \mid \mathbf{X}, \mathbf{y}] = \int_{\overline{\Omega} \times \mathbb{R}} \ell(f_{\text{good}}(\mathbf{x}), y) d\mu(\mathbf{x}, y) - \int_{\overline{\Omega} \times \mathbb{R}} \ell(f_{\text{bayes}}(\mathbf{x}), y) d\mu(\mathbf{x}, y)$$

$$\leq \int_{B \times \mathbb{R}} \ell(f_{\text{good}}(\mathbf{x}), y) d\mu(\mathbf{x}, y)$$

$$\lesssim_{d,\Omega,\mu,\ell} n^{-1+\alpha'd} (\log n)^{q/2}$$

$$= n^{-1+\alpha d} n^{-0.5(\alpha - 1/\beta)} (\log n)^{q/2}$$

$$\lesssim_{k,p,d} n^{-1+\alpha d}.$$

$\square$

*Proof of Theorem 6.3.* Let $\beta = kp - d$, so that $\beta > d$. Let $\alpha = \frac{1}{2} \left( \frac{1}{\beta} + \frac{1}{d} \right)$, so that $\alpha \in \left( \frac{1}{\beta}, \frac{1}{d} \right)$. For $i \in [n]$, let $r_i = \delta_i/4$. By Lemma 4.1, there exist functions $\psi_i \in W^{k,p}(\mathbb{R}^d)$ satisfying the following properties:

1. For all $\mathbf{x} \in \mathbb{R}^d$, $0 \leq \psi_i(\mathbf{x}) \leq 1$.

2. For all $\mathbf{x} \in B(\mathbf{x}_i, r_i/2)$, $\psi_i(\mathbf{x}) = 1$.

3. For all $\mathbf{x} \notin B(\mathbf{x}_i, r_i)$, $\psi_i(\mathbf{x}) = 0$.

4. $\|\psi_i\|_{W^{k,p}(\mathbb{R}^d)}^p \lesssim_{k,p,d} 1 + r_i^{-\beta}$.

Then the $\psi_i$ have disjoint supports, since $\overline{B(\mathbf{x}_i, r_i)} \subset B(\mathbf{x}_i, \delta_i/2)$ and the sets $B(\mathbf{x}_i, \delta_i/2)$ are disjoint by Lemma 4.2. Let $f_{\text{bad}} = \sum_{i=1}^{n} y_i \psi_i$. Then $f_{\text{bad}}$ interpolates the data, and

$$\|f_{\text{bad}}\|_{W^{k,p}(\mathbb{R}^d)}^p \lesssim_{\mu,p} 1 + \sum_{i=1}^{n} (1 + |y_i|)^p \|\psi_i\|_{W^{k,p}(\mathbb{R}^d)}^p$$

$$\lesssim_{k,p,d,\mu} 1 + \sum_{i=1}^{n} (1 + |y_i|)^p r_i^{-\beta}.$$

By Lemma E.6, with probability at least $1 - \epsilon/3$,

$$\sum_{i=1}^{n} (1 + |y_i|)^p r_i^{-\beta} \lesssim_{k,p,d,\Omega,\mu,\ell} \epsilon^{-\beta/d} n^{2\beta/d},$$

so in this case,

$$\|f_{\text{bad}}\|_{W^{k,p}(\mathbb{R}^d)}^p \leq C_1 \epsilon^{-\beta/d} n^{2\beta/d},$$

where $C_1 > 0$ is a constant depending on $k$, $p$, $d$, $\Omega$, $\mu$, $\ell$, and $\alpha$. Let us denote the event that this occurs by $\omega_1$. By Lemma E.5, with probability at least $1 - \epsilon/3$, all min-norm interpolators $f^* \in W^{k,p}(\mathbb{R}^d)$ satisfy

$$\|f^*\|_{W^{k,p}(\mathbb{R}^d)}^p \geq C_2 \epsilon^{\beta/d} n^{2\beta/d},$$

where $C_2 > 0$ is a constant depending on $k$, $d$, $p$, $\Omega$, $\mu$, and $\ell$. Let us denote the event that this occurs by $\omega_2$. If $\omega_1$ and $\omega_2$ both occur, then $f_{\text{bad}}$ is a $\gamma$-ANM interpolator for all $\gamma \geq (C_2^{-1} C_1 \epsilon^{-2\beta/d})^{1/p}$. It remains to show that $f_{\text{bad}}$ is harmful.

By Lemma 4.4, there exists a constant $C_3 > 0$ depending on $\mu$ such that the following holds. With probability at least $1 - \epsilon/3$, there exists a subset $\mathcal{B} \subset [n]$ satisfying the following properties:

1. $|\mathcal{B}| \gtrsim_\mu n$.

2. For all $i \in \mathcal{B}$, $\delta_i \gtrsim_d n^{-1/d}$.

3. For all $i \in \mathcal{B}$, $|y_i| \leq C_3$.

4. For all $i \in \mathcal{B}$,
$$\mathcal{L}(y_i; \mathbf{x}_i) \geq \sigma + \mathcal{L}(f_{\text{Bayes}}(\mathbf{x}_i); \mathbf{x}_i).$$

Let us denote the event that this occurs by $\omega_3$. For the remainder of this proof, we consider the event that $\omega_1, \omega_2$, and $\omega_3$ occur simultaneously. By Sobolev embedding, $f_{\text{Bayes}}$ is continuous, so let $C_4 = \max\left(C_3, \max_{\mathbf{x} \in \overline{\Omega}} |f_{\text{Bayes}}(\mathbf{x})|\right)$. By Lemma 4.6 and the compactness of $\overline{\Omega} \times [-C_4, C_4]$, there exists a constant $C_5 > 0$ depending on $d$, $\Omega$, $\mu$, and $\ell$ such that for all $\mathbf{x}, \mathbf{x}' \in \overline{\Omega}$ with $\|\mathbf{x} - \mathbf{x}'\| \leq C_5$ and all $y \in [-C_4, C_4]$,

$$|\mathcal{L}(y; \mathbf{x}) - \mathcal{L}(y; \mathbf{x}')| \leq \frac{\sigma}{4}.$$

Then for all $i \in \mathcal{B}$ and $\mathbf{x} \in B(\mathbf{x}_i, \min(r_i/2, C_5))$,

$$\begin{aligned}
\mathcal{L}(f_{\text{bad}}(\mathbf{x}); \mathbf{x}) &= \mathcal{L}(y_i; \mathbf{x}) \\
&\geq \mathcal{L}(y_i; \mathbf{x}_i) - \frac{\sigma}{4} \\
&\geq \mathcal{L}(f_{\text{Bayes}}(\mathbf{x}_i); \mathbf{x}_i) + \sigma - \frac{\sigma}{4} \\
&\geq \mathcal{L}(f_{\text{Bayes}}(\mathbf{x}); \mathbf{x}) - \frac{\sigma}{4} + \sigma - \frac{\sigma}{4} \\
&\geq \mathcal{L}(f_{\text{Bayes}}(\mathbf{x}); \mathbf{x}) + \frac{\sigma}{2}.
\end{aligned}$$

Increasing the implicit lower bound on $n$ if necessary, the lower bound $r_i \gtrsim_d n^{-1/d}$ implies that $\min(r_i/2, C_5)^d \gtrsim_d n^{-1}$ for all $i \in \mathcal{B}$. Thus, the excess risk of $f_{\mathrm{bad}}$ can be bounded below as

$$
\mathbb{E}[\ell(f_{\mathrm{bad}}(\mathbf{x}), y) \mid \mathbf{X}, \mathbf{y}] - \mathbb{E}[\ell(f_{\mathrm{bayes}}(\mathbf{x}), y) \mid \mathbf{X}, \mathbf{y}]
$$

$$
= \int_{\overline{\Omega}} \left( \mathcal{L}(f_{\mathrm{bad}}(\mathbf{x}); \mathbf{x}) - \mathcal{L}(f_{\mathrm{Bayes}}(\mathbf{x}); \mathbf{x}) \right) d\mu_x(\mathbf{x})
$$

$$
\geq \sum_{i \in \mathcal{B}} \int_{\Omega \cap B(\mathbf{x}_i, \min(r_i/2, C_5))} \left( \mathcal{L}(f_{\mathrm{bad}}(\mathbf{x}); \mathbf{x}) - \mathcal{L}(f_{\mathrm{Bayes}}(\mathbf{x}); \mathbf{x}) \right) d\mu_x(\mathbf{x})
$$

$$
\geq \sum_{i \in \mathcal{B}} \int_{\Omega \cap B(\mathbf{x}_i, \min(r_i/2, C_5))} \frac{\sigma}{2} d\mu_x(\mathbf{x})
$$

$$
= \frac{\sigma}{2} \sum_{i \in \mathcal{B}} \mu_x(\Omega \cap B(\mathbf{x}_i, \min(r_i/2, C_5)))
$$

$$
\gtrsim_\mu \sum_{i \in \mathcal{B}} |\Omega \cap B(\mathbf{x}_i, \min(r_i/2, C_5))| \qquad\qquad \text{(by Assumption 3.3)}
$$

$$
\gtrsim_{d,\Omega} \sum_{i \in \mathcal{B}} |B(\mathbf{x}_i, \min(r_i/2, C_5))| \qquad\qquad \text{(by Theorem D.14)}
$$

$$
\gtrsim_d \sum_{i \in \mathcal{B}} \min(r_i/2, C_5)^d
$$

$$
\gtrsim_d \sum_{i \in \mathcal{B}} n^{-1}
$$

$$
\gtrsim_\mu 1.
$$

$\square$

