# OpenReview forum: "Harmful Overfitting in Sobolev Spaces"
_ICML.cc/2026/Conference — ICML 2026 regular_

### Official Review · Reviewer_sJVM · 2026-02-27

**Soundness:** 3
**Presentation:** 3
**Significance:** 2
**Originality:** 2
**Overall Recommendation:** 4
**Confidence:** 3

**Summary:**

This paper asks a simple but important question: if we fit noisy data perfectly using very smooth functions, will the model still generalize well? The authors show that in fixed, low-dimensional settings, the answer is no. Even if we choose the smoothest possible function (in a Sobolev sense) that exactly fits the training data, the test error does not go away as we collect more data. In other words, smoothness alone cannot prevent harmful overfitting when the dimension is fixed. The main contribution is extending this result beyond standard Hilbert space (kernel) settings to more general Sobolev spaces and showing, through a geometric argument, that noisy training points inevitably create regions of persistent error. The paper clarifies that benign overfitting is fundamentally a high-dimensional phenomenon and does not arise automatically from smoothness-based inductive bias.

**Compliance With Llm Reviewing Policy:**

Affirmed.

**Final Justification:**

The authors' rebuttal has fully addressed the concerns raised in the previous review. Consequently, I recommend Weak Acceptance and have raised my score from 2 to 4. The paper demonstrates that overfitting in nonparametric regression is not benign. While this finding aligns with intuition, the authors have successfully made the analysis rigorous.

**Key Questions For Authors:**

(1) minor issue: take d=p=1, from Theorem3.7 we require 1<k<1.5. For Lemma 10, if we take beta=1/4, by line 1234 in appendix, k=3/4 contradict with k>1. This suggests there may be a notation inconsistency. Could the authors clarify and resolve this issue?

(2) major issue: at line 328, inequality does not hold because we have a lots of nontrivial constant in Corollary 4.3. After accounting for the leading constants carefully, does the proof in the subsequent section still go through as written?

(3) minor issue: Could the authors comment on the connection between this work and the phenomenon that deep neural networks often generalize well despite perfectly fitting the training data?

**Limitations:**

yes

**Strengths And Weaknesses:**

Strengths:

The paper is generally easy to follow and makes a clear contribution to the understanding of overfitting in nonparametric regression. It clearly distinguishes itself from prior literature. It provides valuable theoretical explanations for why $\gamma$-approximately norm-minimizing interpolators fail to generalize well in fixed dimensions.

Weaknesses:

There appears to be a typo in the proof of the main text. This is a critical issue as it may affect subsequent arguments and requires careful checking and correction(see key question below). There are notation inconsistencies throughout the proofs that need clarification. Although it is a theoretical paper, the absence of experimental results weakens the motivation. Including some experiments would provide empirical verification of the phenomenon. Compared to the surprising results of benign overfitting in linear regression, the findings here (explaining poor generalization in fixed dimensions) are considered less unexpected, even though they are theoretically valuable.

---

> ### Author Rebuttal · Authors · 2026-03-30
>
> We thank the reviewer for the careful reading of the paper and detailed review. We address the concerns here and hope that in light of this the reviewer might consider raising their score.
>
> > There appears to be a typo in the proof of the main text. This is a critical issue as it may affect subsequent arguments and requires careful checking and correction... at line 328, inequality does not hold because we have a lots of nontrivial constant in Corollary 4.3. After accounting for the leading constants carefully, does the proof in the subsequent section still go through as written?
>
> We thank the reviewer for pointing this out and apologize for this error. Thankfully, it is easily fixed: First, we can explicitly identify the constant in 4.3 by changing line 280 to
>
>  $$\lVert f^*\rVert^p\_{W^{k, p}(\mathbb{R}^d)} \leq {C\_\*} n^{kp/d}$$
>
> where $C_*$ depends on $k,p,d$. Below, we update the definition of $C_5$ to include this constant as
>     $$C_5 = \min\left(\frac{C_2}{2}, \left(\frac{C_1 \delta^p \rho }
>         {2C_4 {C_*}}\right)^{1/(kp - d) } \right)$$
>
> in lines 327-328 (column 1). This change to $C_5$  propagates through to cancel out the extra $C_*$ which is in line 328 (column 2) when invoking Corollary 4.3, as desired. This should address the error. We will fix this in the camera-ready version.
>
> > There are notation inconsistencies throughout the proofs that need clarification.
>
> We are updating notation according to the suggestions of the other reviewers. We are happy to incorporate additional suggestions if needed.
>
> > Although it is a theoretical paper, the absence of experimental results weakens the motivation. Including some experiments would provide empirical verification of the phenomenon.
>
> This is a good idea, though it is difficult for the nature of this work. Compared to prior work, one of the main benefits of our approach is that it applies generally to Sobolev norm minimization rather than specific parameterized function classes (ReLU networks, etc., as in the introduction). Our results are based on a particular type of inductive bias rather than our choice of function class. As experimental results would require us to restrict to a particular parameterized function class, it is difficult to distinguish between the effects of the choice of inductive bias and choice of function class when examining the effects of harmful overfitting numerically.
>
> > minor issue: take d=p=1, from Theorem3.7 we require 1<k<1.5. For Lemma 10, if we take beta=1/4, by line 1234 in appendix, k=3/4 contradict with k>1. This suggests there may be a notation inconsistency. Could the authors clarify and resolve this issue?
>
> We note that in our results, $k$ needs to be an integer. We will make this more explicit in the final version. In line 1234, we are *defining* $\beta$ to equal $d - kp$ given fixed values of $d, k, p$ and then applying the lemmas with that particular value of $\beta$. We are not making a claim over all $\beta$ in that line. We will write a few more sentences to make this clear in the final version.
>
> > Could the authors comment on the connection between this work and the phenomenon that deep neural networks often generalize well despite perfectly fitting the training data?
>
> Our work is motivated by this phenomenon, which is often taken as the definition of benign overfitting. This effect is usually seen in the very high-dimensional regime $d \gg n$, and we complement it by studying generalization behavior in the fixed-dimension case. We expand upon this in the introduction and related work sections.

---

> > ### Author Rebuttal · Reviewer_sJVM · 2026-04-01
> >
> > Thanks for your clarification, please rigorously fix this issue to see why it hold or not: ''in lines 327-328 (column 1). This change to $C_5$ propagates through to cancel out the extra $C_*$ which is in line 328 (column 2) when invoking Corollary 4.3, as desired. This **should** address the error. We will fix this in the camera-ready version.''
> >
> > Author may paste the corrected version as response for this message. Specifically, on the last part of page 6, current version is not correct
> > $$
> > \delta^p\left\|\mathcal{B} \backslash \mathcal{B}^{\prime}\right\|  \leq \frac{1}{2} C_1 \delta^p \gamma^{-p} \rho n^{1-k p / d}\left\\|f_\gamma\right\\|_{W^{k, p}\left(\mathbb{R}^d\right)}^p \\
> >  \leq \frac{1}{2} C\_1 \delta^p \rho n^{1-k p / d} \left\\| f^* \right\\|\_{W^{k, p}\left(\mathbb{R}^d\right)}^p \\
> >  \leq \frac{1}{2} C\_1 \delta^p \rho n \quad \text { (By (2)). }
> > $$
> >
> > and consequently, the following are incorrect:
> >
> > Rearranging, we get
> > $
> > \left|\mathcal{B} \backslash \mathcal{B}^{\prime}\right| \leq \frac{1}{2} C_1 \rho n,
> > $
> >
> > so by property 1 of $\mathcal{B}$,
> >
> > $$
> > \left|\mathcal{B}^{\prime}\right|=|\mathcal{B}|-\left|\mathcal{B} \backslash \mathcal{B}^{\prime}\right| \geq C_1 \rho n-\frac{1}{2} C_1 \rho n=\frac{1}{2} C_1 \rho n .
> > $$

---

> > > ### Author Response · Authors · 2026-04-04
> > >
> > > Here is a description of the changes to the proof. As we mentioned in the previous response, we explicitly identify the constant in Corollary 4.3 by changing line 280 to
> > > $$
> > > \|f^\*\|^p\_{W^{k, p}(\mathbb{R}^d)} \leq C_\* n^{kp/d},
> > > $$
> > > where $C_*$ depends on $k, p, d$. We also update $C_5$ to have the additional factor of $C_*$ as
> > > $$
> > > C_5 = \min\left(\frac{C_2}{2}, \left(\frac{C_1 \delta^p \rho}{2C_4 C_*} \right)^{1/(kp - d)} \right).
> > > $$
> > > With these changes, beginning in line 288 (column 2), we now have
> > > $$
> > > \begin{aligned}
> > >         \delta^p &\leq |f\_{\gamma}(\mathbf{x}) - f\_{\gamma}(\mathbf{x}\_i)|^p\\\\
> > >         &\leq C\_4\|\mathbf{x} - \mathbf{x}\_i\|^{kp-d}\|f_\gamma\|^p\_{W^{k,p}(B(\mathbf{x}\_i, \delta\_i/2))}\quad (\text{By (3)})\\\\
> > >         &\leq C_4 \left(C_5 \gamma^{-p/(kp-d)}n^{-1/d} \right)^{kp - d} \|f_\gamma\|^p_{W^{k,p}(B(\mathbf{x}\_i, \delta\_i/2))}\\\\
> > >         &\leq C\_4 \left(\left(\frac{C_1 \delta^p \rho}{2C\_4C\_\* \gamma^p }\right)^{1 / (kp - d)} n^{-1/d} \right)^{kp - d} \|f\_\gamma\|^p\_{W^{k,p}(B(\mathbf{x}\_i, \delta\_i/2))}\\\\
> > >         &= \frac{1}{2} \frac{C\_1}{C\_\*} \delta^p \gamma^{-p} \rho n^{1 - kp/d} \|f_\gamma\|^p_{W^{k,p}(B(\mathbf{x}_i, \delta\_i/2))}.
> > > \end{aligned}
> > > $$
> > >
> > > Summing the above over $\mathcal{B}\setminus\mathcal{B}'$, we get
> > > $$
> > > \delta^p|\mathcal{B}\setminus\mathcal{B}'|\leq\frac{1}{2}\frac{C\_1}{C\_\*} \delta^p \gamma^{-p} \rho n^{1 - kp/d}\sum\_{i \in \mathcal{B}\setminus\mathcal{B}'}\|f\_{\gamma}\|^p\_{W^{k,p}(B(\mathbf{x}\_i, \delta\_i/2))}.
> > > $$
> > > The inequalities from lines 308-320 (right) are unchanged (except for fixing a typo $f \to f_{\gamma}$ on the last line), which gives us
> > > $$
> > > \sum\_{i \in \mathcal{B}\setminus\mathcal{B}'}\|f\_{\gamma}\|^p\_{W^{k,p}(B(\mathbf{x}\_i, \delta\_i/2))}\leq \|f_{\gamma}\|^p\_{W^{k, p}(\mathbb{R}^d)}.
> > > $$
> > > Then
> > > $$
> > > \begin{aligned}
> > > \delta^p |\mathcal{B}\setminus\mathcal{B}'| &\leq \frac{1}{2}\frac{C\_1}{C\_\*} \delta^p \gamma^{-p} \rho n^{1 - kp/d}\sum\_{i \in \mathcal{B}\setminus\mathcal{B}'}\|f\_{\gamma}\|^p\_{W^{k,p}(B(\mathbf{x}\_i, \delta\_i/2))}\\\\
> > > &\leq \frac{1}{2}\frac{C\_1}{C\_\*} \delta^p \gamma^{-p} \rho n^{1 - kp/d} \|f\_{\gamma}\|\_{W^{k, p}(\mathbb{R}^d)}^p\\\\
> > > &\leq \frac{1}{2}\frac{C\_1}{C\_\*} \delta^p\rho n^{1 - kp/d} \|f^\*\|\_{W^{k, p}(\mathbb{R}^d)}^p\\\\
> > > &\leq \frac{1}{2}\frac{C\_1}{C\_\*} \delta^p \rho n^{1 - kp/d}(C\_\* n^{kp/d} ) \\\\
> > > &\leq \frac{1}{2} C\_1 \delta^p  \rho n
> > > \end{aligned}
> > > $$
> > > where the third line follows from the $\gamma$-ANM property and the fourth line follows from the definition of $C_*$. Finally, rearranging, we get $\left|\mathcal{B} \setminus \mathcal{B}^{\prime}\right| \leq \frac{1}{2} C_1 \rho n$, so by property 1 of $\mathcal{B}$,
> > >
> > > $$\left|\mathcal{B}^{\prime}\right|=|\mathcal{B}|-\left|\mathcal{B} \setminus \mathcal{B}^{\prime}\right| \geq C_1 \rho n-\frac{1}{2} C_1 \rho n=\frac{1}{2} C_1 \rho n$$
> > >
> > > as desired.

---

### Official Review · Reviewer_zvRW · 2026-03-03

**Soundness:** 4
**Presentation:** 3
**Significance:** 3
**Originality:** 4
**Overall Recommendation:** 5
**Confidence:** 4

**Summary:**

The paper investigates the generalization properties of functions that perfectly interpolate noisy training data within Sobolev spaces $W^{k, p}(\mathbb{R}^d)$. Motivated by the recent literature on benign overfitting, the authors prove that approximately norm-minimizing interpolators in Sobolev spaces exhibit harmful overfitting. More precisely, under several assumptions on label noise, loss function, data distribution, and regularity of the Bayes risk function, they show that the expected excess risk of these interpolators remains bounded below by a positive constant, even as the sample size $n \to \infty$. This work extends previous results from limited combinations of $k$ and $p$ to arbitrary $k$ and $p \in [1, \infty)$.

**Compliance With Llm Reviewing Policy:**

Affirmed.

**Final Justification:**

As the problem of harmful overfitting with label noises warrants some attentions for practical consideration in statistical learning, I believe contributes some insight in this aspect. As the discussion with the authors has cleared up all of my concerns, I would like to maintain my positive evaluation of the paper.

**Key Questions For Authors:**

1. Regarding the main result, do we assume that $y$ is continuous? Does the main result hold for discrete $\mu_y$ as well?

2. From what I read, the result of Buchholz only holds in RKHS, that is, only when $p=2$ (though we may use Sobolev embedding to extend the condition $d/2 <k <3d/4$ to other values of $k$ and $p\ge 1$).

2. In Assumption 3.3, is it possible to replace the lower bound by assuming that $p(x)$ is sub-exponential?

3. If $\mu_y$ can be discrete, regarding Assumption 3.5, can the authors make some comments on whether this assumption is equivalent to $\min_y P(y | x) > \epsilon$ for some $\epsilon >0$?

4. The statement of Corollary 3.8 should mention that the result is strictly for the squared loss.

**Limitations:**

The authors have already mentioned some of the important limitations. In addition, if not entirely addressed in the main paper, the authors should add the setting where $k \not\in (d/p, 1.5d/p)$ as a possible direction.

**Strengths And Weaknesses:**

### Strengths

1. Extending the harmful overfitting results from limited combinations of $k$ and $p$ to arbitrary $k$ and $p \in [1, \infty)$ is a substantial theoretical contribution.

2. The paper introduces the notion of $\gamma$-approximate norm minimizers that allows the main bound to be quantified based on $\gamma$. In addition, the bound applies to a broad class of loss functions.

3. Personally, I appreciate the detailed proofs of the lemmas, for example, the proof of the continuity of the conditional loss (Lemma 4.6).

### Weaknesses

1. The main theorem relies on the condition $k \in (d/p, 1.5d/p)$, which matches the result of Buchholz (2022). There should be a discussion of why such condition is required. If there are some challenges when $k$ and $p$ are outside of this regime, the authors must mention them. If harmful overfitting outside of this regime is not possible, the authors must provide some counterexamples.

2. Though the authors have provided extensive list of references in the Related Work section, there is a lack of references in the Setup and Proof sections, in order to show how the proof is related to prior work. Specifically, the authors should discuss whether any of the assumptions are also used in previous work, whether the minimum-norm solution is defined in the previous work, and whether the proof technique is completely new, or some of it is motivated by previous work. This is to help the reader understand the paper’s place within the benign overfitting landscape

3. The authors could use a simple simulation experiment to further support the main result.

---

> ### Author Rebuttal · Authors · 2026-03-30
>
> We thank the reviewer for their time and for the thoughtful review of the paper. We address your comments and questions here.
>
> > The main theorem relies on the condition $k\in(d/p,1.5d/p)$, which matches the result of Buchholz (2022). There should be a discussion of why such condition is required.
>
> We expect that for differentiability indices $k\gg d/p$, there will exist solutions which overfit both benignly and harmfully due to the norm becoming increasingly localized. As $p\to\infty$, the representational cost is dominated by the "worst region", which suggests that there is enough flexibility in the "average region" to accommodate both benign and harmful overfitting. We will provide intuition for this regime in the final version.
>
> > the authors should discuss whether any of the assumptions are also used in previous work, whether the minimum-norm solution is defined in the previous work, and whether the proof technique is completely new, or some of it is motivated by previous work.
>
> Our proof technique is different from existing work in that it uses the local geometry of interpolators rather than kernel methods specific to the case $p=2$. The minimum-norm interpolating solution is a standard setup used in the benign overfitting literature, e.g. in [1]. Our other assumptions are mostly generalizations of the assumptions in the closely related work [2] and [3]. Assumption 3.3 is the same assumption on the density of the input distribution used by [2] and [3]. Assumption 3.5 is a generalization of the additive noise $y=f^*(x)+\xi$ for label noise $\xi$, which is the setting of [2] and [3]. Assumption 3.6 is essentially stating that there exists a sufficiently smooth ground truth function generating the data and generalizes the smooth ground truth assumed by [2] and [3].
>
> > The authors could use a simple simulation experiment to further support the main result.
>
> This is a good idea, though it is difficult for the nature of this work. Compared to prior work, one of the main benefits of our approach is that it applies generally to Sobolev norm minimization rather than specific parameterized function classes (ReLU networks, etc., as in the introduction). Our results are based on a particular type of inductive bias rather than our choice of function class. As experimental results would require us to restrict to a particular parameterized function class, it is difficult to distinguish between the effects of the choice of inductive bias and choice of function class when examining the effects of harmful overfitting numerically.
>
> > Regarding the main result, do we assume that $y$ is continuous? Does the main result hold for discrete $\mu_y$ as well?
>
> $y$ does not need to be a continuous random variable, though its conditional distribution needs to be sufficiently well-behaved as stated in Assumption 3.2. An example of a non-continuous distribution on $y$ that satisfies this condition is where $y$ takes values in $\{-1,1\}$ with $P(y=1\mid x)=g(x)$ for a continuous bounded function $g$. We will add a discussion of this example in the final version.
>
> > From what I read, the result of Buchholz only holds in RKHS, that is, only when $p=2$
>
> This is a good point, and is one of the motivations for our work. We will clarify this in the introduction.
>
> > In Assumption 3.3, is it possible to replace the lower bound by assuming that $p(x)$ is sub-exponential?
>
> Since the density $p(x)$ is supported on a compact set $\overline{\Omega}$, a sub-exponential tail would not provide additional information, unless you are referring to a different definition of sub-exponential. We use the upper and lower bound to relate the probability measure for $x$ to the Lebesgue measure of the set where the interpolator generalizes poorly. The two-sided bound lets us exchange between these two notions in our proofs.
>
> > If  $\mu_y$ can be discrete, regarding Assumption 3.5, can the authors make some comments on whether this assumption is equivalent to $\min_y P(y\mid x)\geq\epsilon$ for some $\epsilon>0$?
>
> In the case where $\mu_y$ is discrete and bounded and the loss function $\ell$ is strongly convex in $\hat{y}$, the condition that $\min_y P(y\mid x)\geq\epsilon$ for a fixed $\epsilon>0$ does indeed imply Assumption 3.5. We will expand upon this example in the final version.
>
> > The statement of Corollary 3.8 should mention that the result is strictly for the squared loss.
>
> This is a good catch; we will clarify this in the final version.
>
> [1] Bartlett, P. L., Long, P. M., Lugosi, G., and Tsigler, A. Benign overfitting in linear regression. Proceedings of the National Academy of Sciences, 117(48):30063–30070, 2020.
>
> [2] Buchholz, S. Kernel interpolation in sobolev spaces is not consistent in low dimensions. In Conference on Learning Theory, pp. 3410–3440. PMLR, 2022.
>
> [3] Rakhlin, A. and Zhai, X. Consistency of interpolation with Laplace kernels is a high-dimensional phenomenon. In Conference on Learning Theory, pp. 2595–2623. PMLR, 2019.

---

> > ### Author Rebuttal · Reviewer_zvRW · 2026-04-01
> >
> > Thanks. I'm satisfied with the answers and will keep my high score. Please include the additional discussions as they will be helpful for the readers.

---

### Official Review · Reviewer_C8Xb · 2026-03-11

**Soundness:** 4
**Presentation:** 4
**Significance:** 2
**Originality:** 2
**Overall Recommendation:** 4
**Confidence:** 4

**Summary:**

In this article, the authors study the properties of interpolating predictors that are near-minimizing a given Sobolev norm. They give conditions for this interpolation to be harmful, and detached from the context of benign overfitting. In particular, in the regime where $k\in (d/p, 3d/2p)$, they lower bound the generalization error, showing that it does not vanish as $n\to \infty$.

**Compliance With Llm Reviewing Policy:**

Affirmed.

**Final Justification:**

I believe results on benign overfitting to be of interest in the ML community. Here the authors take a negative stance, and derive examples of overfitting for the Sobolev norm. However, this is not a strong accept from my side because the results are not surprising, the mathematical tools are not very involved, the assumptions are restrictive, the link with neural networks is unclear, and this harmful overfitting can be easily bypassed by penalizing by the kernel norm.

**Key Questions For Authors:**

1) There seems to me that there is a typo in the Proof of Corollary 4.3. Should it not be $\beta = kp -d > 0$?

2) Thoughout the article, is $k$ an integer, or do you consider fractional Sobolev spaces?

3) In the introduction, the authors clearly state that similar results were derived for $p=2$. However, this appears to me as the most interesting settings, since in this case the problem is a kernel methods, and there are some known algorithms to solve it (like using the Matérn kernels). What algorithms or practical case falls into the case $p \neq 2$?

4) though upper bounding the probability density is common to establish meaningful convergence rate (the extreme case being that if $X$ is a Dirac, then the convergence rate falls back to the parametric $1/n$ rate), lower bounding the probability density as done in Assumption  3.3 is quite restrictive. Can the authors justify this need?

5) In the conclusion, the authors state that in the *case d ≫ n, one sees benign overfitting* for interpolating predictors of minimum Sobolev norm. Could they provide a reference?

**Limitations:**

The authored offers some ideas of improvements in the conclusion.

**Strengths And Weaknesses:**

The paper is easy to read and looks technically sound. It adresses an interesting problem, though focusing on the case  $p\neq 2$ and with the very specific condition that $k\in (d/p, 3d/2p)$, which does not adress practical algorithms as far as I know. The proofs rely on well-known probability (concentration inequalities) and functional analysis results (sobolev embeddings...).

---

> ### Author Rebuttal · Authors · 2026-03-30
>
> We thank the reviewer for their helpful feedback and questions which we address here.
> > There seems to me that there is a typo in the Proof of Corollary 4.3. Should it not be $\beta = kp -d > 0$?
>
> In the statement of Corollary 4.3, we assume that $k \in (d/p, 1.5d/p)$, which implies that $\beta = kp- d \in (0, d/2)$, as is necessary for Corollary C.10. We will clarify this in the camera-ready version.
>
> > Thoughout the article, is $k$ an integer, or do you consider fractional Sobolev spaces?
>
> Yes, our results assume that $k$ is an integer. The case where $k$ is not an integer and $p \neq 2$ is challenging since the norm is no longer defined by pointwise derivatives and we cannot fall back on kernel methods. Our proof techniques use local information about interpolants which becomes more involved to use in the fractional case. We present our results in the integer case to illustrate the effect over a range of values of $k$ while keeping our arguments simple.
>
> > In the introduction, the authors clearly state that similar results were derived for $p=2$. However, this appears to me as the most interesting settings, since in this case the problem is a kernel methods, and there are some known algorithms to solve it (like using the Matérn kernels). What algorithms or practical case falls into the case $p\neq 2$?
>
> The case $p = 2$ is indeed an important special case. Our motivation for generalizing beyond this case is to better understand how approximate norm minimization as a general phenomenon impacts generalization. There are several works studying this for the $p = 2$ case, but it is not obvious how much of this is an artifact of the specific choice of norm. For instance, neural networks trained by gradient descent are not easily captured under this umbrella. On the other hand, the weight norm of a neural network can be bounded in terms of a Sobolev norm $W^{d, d + 1}(\mathbb{R}^d)$ (see Corollary 1 of [1]). We will add motivation for our setting in the camera-ready version.
>
> > though upper bounding the probability density is common to establish meaningful convergence rate (the extreme case being that if $X$ is a Dirac, then the convergence rate falls back to the parametric $1/n$ rate), lower bounding the probability density as done in Assumption 3.3 is quite restrictive. Can the authors justify this need?
>
> We impose both upper and lower bounds on the density so we can translate between the probability measure of a subset of $\Omega$ and the Lebesgue measure and keep the geometric argument simple. It is possible that this can be generalized with a condition that the probability density not be spread out "too much". One can interpret this condition as "filtering out outliers" which  consist of infrequent parts of the input distribution. We also remark that this is the same condition used in [3] and [4].
>
> > In the conclusion, the authors state that in the case d ≫ n, one sees benign overfitting for interpolating predictors of minimum Sobolev norm. Could they provide a reference?
>
> This is a good catch. We should clarify that we are referring broadly to benign overfitting results in the $d \gg n$ case, such as [2] (as well as the several other works on benign overfitting in linear regression and neural networks which we cite in the introduction), which are not for the more general Sobolev norm minimization case. We would like to convey that the regime $d \gg n$ is the typical setting of benign overfitting, and outside of this scaling, we are able to prove that the generalization behavior is qualitatively different. We will clarify this in camera-ready version.
>
>
>
> [1] Ongie, G., Willett, R., Soudry, D., and Srebro, N. A function space view of bounded norm infinite width relu
> nets: The multivariate case. In International Conference on Learning Representations, 2020.
>
> [2] Kornowski, G., Yehudai, G., and Shamir, O. From tempered
> to benign overfitting in ReLU neural networks. In Thirty-seventh Conference on Neural Information Processing
> Systems, 2023.
>
> [3] Buchholz, S. Kernel interpolation in sobolev spaces is not consistent in low dimensions. In Conference on Learning Theory, pp. 3410–3440. PMLR, 2022.
>
> [4] Rakhlin, A. and Zhai, X. Consistency of interpolation with Laplace kernels is a high-dimensional phenomenon. In Conference on Learning Theory, pp. 2595–2623. PMLR, 2019.

---

> > ### Author Rebuttal · Reviewer_C8Xb · 2026-04-02
> >
> > I thank the authors for their rebuttal. I maintain my opinion that the paper reads well, that the topic is interesting, and that the results are well analyzed. I appreciate that the authors were honnest and accurate in their response.
> >
> > Please make sure to precise that you only tackle the case $s \in \mathbb N$, maybe here on page 1 when you write *We consider interpolators in spaces $W^{k,p}(\mathbb R^d)$ for $p \in [1, \infty)$*, and maybe also in the conclusion as future works.  Similarly, please flag clearly that your assumptions are somewhat restrictive, because of the framework you have developped, and that future works could focus on relaxing those assumptions.
> >
> > I have read this article exclusively from the point of view of kernel theory, and already found the result noteworthy in this context. Maybe it should be clearly stated however that adding the kernel norm results in the consistency of the kernel method, and in the Sobolev minimax rate, so that your work might not be wrongly interpreted as a failure of kernel method in high dimension.
> >
> > However, reading the authors' rebuttal and looking again at the paper, I see that they intend to establish a link with neural networks. Please, either frame the motivation as a general study of benign overfitting accross different algorithms, or if you wish to say that your results can somehow be applied to neural networks, please add a paragraph and cite some papers linking neural networks' optimization with the Sobolev norm minimization setting. Whatever you chose, be clearer about it.
> >
> > Given the restrictive assumptions, the unclear link with neural networks, and the fact that this harmful overfitting can be easily bypassed by penalizing by the kernel norm, I chose to keep my score as it is.

---

### Official Review · Reviewer_Mo8k · 2026-03-12

**Soundness:** 4
**Presentation:** 4
**Significance:** 3
**Originality:** 3
**Overall Recommendation:** 5
**Confidence:** 4

**Summary:**

The paper is concerned with harmful overfitting in Sobolev spaces. This means that the authors prove a lower bound for the generalization error of a hypothesis which has zero training error and at the same time (almost) minimizes the $W^{k,p}(\mathbb R^d)$ Sobolev norm, where the differentiability index $k$ is in a certain range to, in particular, ensure existence of continuous representatives. In particular, the result implies that double descent does not occur in this setting. To prove the result the authors uses geometric methods combined with concentration inequalities to (1) construct an explicit minimizer of the empirical risk with controlled Sobolev norm, (2) show that any data set contains points with labels that are sufficiently noisy, and (3) prove that the smoothness of the explicit interpolator implies a large generalization error around these points.

**Compliance With Llm Reviewing Policy:**

Affirmed.

**Final Justification:**

I am happy with the discussion and maintain my score to accept the paper.

**Key Questions For Authors:**

See above for some more conceptual questions. Here are a few minor ones:

- In Assumption 3.2, what exactly do you mean by ``regular conditional probability''? What do you mean by ``version''. This assumption should be more mathematically precise.
- In Assumption 3.3 misses a quantor for $\mathbf x$. Is it that $\forall \mathbf x$ there exist constants $c_\mathcal{D},D_\mathcal{D}$ or that there exist such constants such that $\forall\mathbf x$ we have the inequality?
- In the same assumption, is $\mu_{\mathbf x}$ related to the conditional probability from Assumption 3.2 and, if yes, how?
- In the same assumption, was $\mathcal D$ introduced somewhere?
- After Theorem 3.7 you write $\mathcal D_{\mathbf x}$. Do you mean $\mu_{\mathbf x}$?
- In inequality (3), please explain how you get $\|\mathbf x-\mathbf x_i\|$ on the right hand side and not just $\delta_i$.
- In the first inequality after (4), do you mean $|f_\gamma(\mathbf x) - f_\gamma(\mathbf x_i)|\geq\delta$ instead of $\leq$?
- In the second to last displayed inequality on page 6 you are missing a subscript $\gamma$ at the very end.
- In the proof of Lemma C.1 do you have a reference for the claim that every subset of an open set with full measure is dense? Couldn't one think of the fat Cantor set as possible counter example?
- In Appendix C.2 please check the correct domains and codomains for the functions $\psi$ and $\varphi$, especially in Lemma C.4. The proof of this result also seems incomplete.
- In the proof of Lemma 4.1 why not say that the support of $g_{j,d}$ is $[0,1]$ since it just takes non-negative inputs?

**Limitations:**

Not applicable.

**Strengths And Weaknesses:**

**Strengths**
- The paper is very well written, the proofs are correct (I didn't check all carefully), and the structure of the paper makes it easy to follow the ideas.
- The result seems novel and interesting, in particular, since it contributes to de-mystifying the generalization abilities of certain hypothesis classes.
- The result significantly generalizes previously known results in similar settings.

**Weaknesses**
- The paper is a bit scarce in comments on how sharp the assumptions are. E.g. what happens for large differentiability indices $k\gg \frac{d}{p}$? Shouldn't the solution be even smoother and generalize even worse?
- To support the significance of the result, it would be nice to make some examples for commonly used hypothesis classes and how the result applies to them. Can one, e.g., relate the result to Barron spaces which possess embeddings in the Sobolev spaces under consideration? Even though this problem might be non-trivial to tackle, some guidance would be appreciated. Related to the previous point, does the result apply to any parametrized hypothesis class?

---

> ### Author Rebuttal · Authors · 2026-03-30
>
> We thank the reviewer for the thoughtful review and careful reading of the paper. We address each of your points below.
>
> > E.g. what happens for large differentiability indices $k\gg\frac{d}{p}$? Shouldn't the solution be even smoother and generalize even worse?
>
> We expect that for differentiability indices $k \gg d/p$, there will exist solutions which overfit both benignly and harmfully due to the norm becoming increasingly localized. As $p \to \infty$, the representational cost is dominated by the "worst region", which suggests that there is enough flexibility in the "average region" to accommodate both benign and harmful overfitting.
> We will provide intuition for this regime in the final version.
> > To support the significance of the result, it would be nice to make some examples for commonly used hypothesis classes and how the result applies to them.
>
> For spaces which embed into Sobolev spaces, our results apply to them in the sense that interpolators in those spaces *which approximately minimize the Sobolev norm* will exhibit harmful overfitting. That is, our results are making conclusions based off of a type of inductive bias rather than off of the function class.
>
> > In Assumption 3.2, what exactly do you mean by "regular conditional probability"?
>
> We mean a mapping $\nu: \overline{\Omega} \times \mathcal{B}(\mathbb{R})\to[0,1]$ satisfying the following two conditions.
> - For all $x\_0\in\overline{\Omega}$, the map $A\mapsto\nu(x_0,A)$ is a probability measure.
> - For all $A\in\mathcal{B}(\mathbb{R})$, the map $x\mapsto\nu(x,A)$ is a version of the conditional probability $\mathbb{P}(y\in A \mid x)$.
>
> Here "version" is a random variable which is equal to the conditional probability almost surely. See Chapter 4.1.3 of [1] for more details. We will provide this background in the final version.
>
> > In Assumption 3.3 misses a quantor for $x$.
>
> This should read: "There exist constants $c\_{\mathcal{D}}, C\_{\mathcal{D}}$ such that for all $x,x\_0\in\overline{\Omega}$, the marginal distribution $\mu\_x$ has a density $p\_x: \overline{\Omega} \to [0,\infty)$ satisfying $c\_{\mathcal{D}}\leq p\_x(x\_0) \leq C_{\mathcal{D}}$." We will update this in the final version.
> > In the same assumption, is $\mu\_x$ related to the conditional probability from Assumption 3.2 and, if yes, how?
>
> The marginal distribution $\mu_{x}$ in Assumption 3.3 is the distribution of $x$ when $(x,y)$ is sampled from $\mu$. We introduce this notation in the first paragraph of Section 3. The regular conditional distribution $\nu_{x_0}$ is distinct; it essentially measures the distribution of $y$ conditioned on $x=x_0$.
> > In the same assumption, was $\mathcal{D}$ introduced somewhere?
>
> The subscript $\mathcal{D}$ was intended to indicate that these constants are for the input distribution, but in the final version we will use the notation $c_{\mu},C_{\mu}$ instead for clarity.
> > After Theorem 3.7 you write $\mathcal{D}_x$. Do you mean $\mu_x$?
>
> Thanks for pointing this out; we will correct this.
> > In inequality (3), please explain how you get $\|x-x_i\|$ on the right hand side and not just $\delta_i$.
>
> We apply Corollary 4.5 using the ball centered at $x_i$ with radius $\|x-x_i\|$ rather than $\delta_i$. This is the minimal possible radius according to the corollary, which may be smaller than $\delta_i$ alone.
> > In the first inequality after (4), do you mean $\|f\_{\gamma}(x)-f_{\gamma}(x_i)\|\geq\delta$ instead of $\leq$?
>
> This should be correct as written. In line 281 (column 2), note that we are assuming that $i\notin\mathcal{B}'$, so line 288 (column 2) is reversed from the first inequality after (4).
> > In the second to last displayed inequality on page 6 you are missing a subscript $\gamma$ at the very end.
>
> Thanks for catching this; we will fix this in the final version.
> > In the proof of Lemma C.1 do you have a reference for the claim that every subset of an open set with full measure is dense?
> Couldn't one think of the fat Cantor set as possible counter example?
>
> To see why this is true, suppose that $x$ is not in the closure of the set. Then there exists a ball around $x$ which is not in the set, so the set doesn't have full measure. The fat Cantor set has positive but not full measure.
> > In Appendix C.2 please check the correct domains and codomains for the functions $\psi$ and $\varphi$, especially in Lemma
> C.4. The proof of this result also seems incomplete.
>
> Thanks for catching this. It should read "such that for all $x\in\mathbb{R}^d$, we have $\psi(x)=\phi(\|x\|)$". Aside from this the proof should be correct. We will fix this in the revision.
> > In the proof of Lemma 4.1 why not say that the support of  $g_{j,d}$ is $[0,1]$ since it just takes non-negative inputs?
>
> By the way we have defined $g_{j,d}$, it could technically take any values for inputs less than zero, but as you point out this does not have any effect for our purposes.
>
> [1] Durrett, R. Probability: theory and examples, volume 49. Cambridge university press, 2019.

---

> > ### Author Rebuttal · Reviewer_Mo8k · 2026-04-01
> >
> > The authors satisfactorly answered my questions and addressed my concerns. I will maintain my rating to accept the paper.

---

### Decision · Program_Chairs · 2026-04-30

**Decision:**

Accept (regular)

**Comment:**

The paper studies harmful overfitting for approximately norm-minimizing interpolators in Sobolev spaces, showing that exact interpolation with smooth functions can still fail to generalize. The paper is clearly written, technically solid, and extends prior harmful-overfitting results beyond narrower special cases. The main concern is that the result is developed under somewhat restrictive assumptions and its connection to practical algorithms, especially neural networks, remains indirect. Even so, the discussion ended up clearly positive: after the rebuttal, all four reviewers remained on the acceptance side with technical concerns about the proof details and assumptions addressed. Although concerns remain regarding the restrictiveness of the assumptions and the limited connection to practical algorithms, the main result is novel, technically solid, and sufficiently valuable to justify acceptance.